# A multi-wavelength numerical model in support to quantitative retrievals of aerosol properties from automated-lidar-ceilometers and test applications for AOT and PM10 estimation

Davide Dionisi[1,2], Francesca Barnaba[1], Henri Diémoz[3], Luca Di Liberto[1], Gian Paolo Gobbi[1]

[1]Istituto di Scienze dell'Atmosfera e del Clima, Consiglio Nazionale delle Ricerche (ISAC-CNR), Roma, Italy
[2]Istituto di Scienze Marine, Consiglio Nazionale delle Ricerche (ISMAR-CNR), Roma, Italy
[3]Aosta Valley Regional Environmental Protection Agency (ARPA Valle d'Aosta), Saint-Christophe (Aosta), Italy

*Correspondence to*: Davide Dionisi (d.dionisi@isac.cnr.it)

**Abstract.** Knowledge of the vertical distribution of aerosol particles is a key factor in the study of climate, air pollution, and meteorological processes. The use of automated lidar-ceilometers (ALC) systems for the aerosol vertically-resolved characterization has increased in the recent years thanks to their low construction and operation costs, and to their capability at providing continuous, unattended measurements. At the same time there is a need to convert the ALC signals into usable geophysical quantities. In fact, the quantitative assessment of the aerosol properties from ALC measurements and the relevant assimilation in meteorological forecast models is amongst the main objectives of the EU COST Action TOPROF (Towards Operational ground-based PROFiling with ALCs, doppler lidars and microwave radiometers). Concurrently, the E-PROFILE program of the European Meteorological Services Network (EUMETNET) focuses on the harmonization of ALC measurements and data provision across Europe. Within these frameworks, we implemented a model-assisted methodology to retrieve key aerosol properties (extinction coefficient, surface area and volume) from elastic lidar and/or ALC measurements. The method is based on results from a large set of aerosol scattering simulations (Mie-theory) performed at UV, visible and near IR wavelengths using a "Monte-Carlo" approach to select the input aerosol microphysical properties. An average 'continental aerosol type' (i.e. clean-to-moderately polluted continental aerosol conditions) is addressed in this study. Based on the simulation results, we derive mean functional relationships linking the aerosol backscatter coefficients to the above-mentioned variables. Applied in the data inversion of single wavelength lidars and/or ALCs, these relationships allow quantitative determination of the vertically-resolved aerosols backscatter, extinction, volume and surface area and, in turn, of the extinction-to-backscatter ratio (i.e., the lidar-ratio, LR) and of extinction-to-volume conversion factor ($c_v$) at 355, 532, 1064 nm. These variables provide valuable information for visibility, radiative transfer and air quality applications. This study also includes 1) validation of the model simulations with real measurements and 2) test applications of the proposed model-based ALC inversion methodology. In particular, our model simulations were compared to backscatter and extinction coefficients independently retrieved by Raman lidar systems operating at different continental sites within the European Aerosol Research LIdar NETwork (EARLINET). This comparison shows good model-measurements agreement, with LR discrepancies below 20%. The model-assisted quantitative retrieval of both aerosol extinction and volume was then tested using raw data from three different ALCs systems (CHM15k-Nimbus), operating within the Italian Automated Lidar-ceilometer Network (ALICENET). To this purpose, a one-year-record of the ALCs-derived aerosol optical thickness (AOT) at each site was compared to direct AOT measurements performed by co-located sun-sky photometers. This comparison shows an overall AOT agreement within 30% at all sites. At one site, the model-assisted ALC estimation of the aerosol volume and mass (i.e., PM10) in the lowermost levels was compared to values

measured at the surface-level by co-located in situ instrumentation. Within this exercise, the ALC-derived daily-mean
mass concentration was found to well reproduce the corresponding (EU regulated) PM10 values measured by the local
Air Quality agency in terms of both temporal variability and absolute values. Although limited in space and time, the
good performances of the proposed approach in these preliminary tests suggest it could possibly represent a valid option
to extend the capabilities of ALCs at providing quantitative information for operational air quality and meteorological
monitoring.
**1    Introduction**
Due to the impact of atmospheric aerosols on both air quality and climate, substantial efforts have been made to expand
our knowledge of their sources, properties and fate. Aerosol particles affect the Earth's radiation budget mainly by two
different processes: 1) by scattering and absorbing both solar and terrestrial radiation (aerosol direct effect, Haywood
and Boucher, 2000 and aerosol semi-direct effect, Johnson et al., 2004) and 2) by serving as cloud and ice condensation
nuclei (aerosol indirect effect, Lohmann and Feichter, 2005, Stevens and Feingold, 2009 and Feingold et al., 2016). The
complexity of these processes and the extreme spatial and temporal variability of the aerosol sources, physical and
chemical properties and atmospheric processing make the quantification of their impacts very difficult. Aerosols have
also proven detrimental effects on human health (e.g., D'Amato et al., 2013, World Health Organization, 2013,
Lelieveld et al., 2015). In fact, their concentration (often evaluated in terms of particulate matter mass, or PM) is
regulated by specific air quality legislation worldwide. In Europe, the Air Quality Directive 2008/50 defines the
'objectives for ambient air quality designed to avoid, prevent or reduce harmful effects on human health and the
environment as a whole' (EC, 2008).
Among the aerosol observational systems, the LIDAR technique has been proved to be the optimal tool to provide
range-resolved, accurate aerosol data necessary in radiative transfer computations (e.g. Koetz et al., 2006, Tosca et al.,
2017) and is often usefully employed in supporting air quality studies (e.g. Menut et al., 1997, He et al., 2012). With a
spectrum of different system types (elastic backscatter, Raman, High Spectral Resolution, and multi-wavelength lidars),
each with specific pro and cons (Lolli et al., 2018), this technique allows retrievals of aerosol and cloud optical
properties and relevant distribution within the atmospheric column at several ground-based observational sites (Fernald
et al., 1972; Klett, 1981; Shipley et al., 1983, Kovalev and Eichinger, 2004, Heese and Wiegner, 2008; Ansmann et al.,
2012). Since 2006, the Cloud Aerosol Lidar and Infrared Pathfinder Satellite Observation (CALIPSO) platform (Winker
et al. 2003) also provides a unique, global view of aerosol and cloud vertical distributions through space-based
observations (at the operating wavelengths of 532 and 1064 nm). Recently, within the Cloud-Aerosol Transport System
(CATS) mission, a lidar was also installed at the International Space Station (ISS, McGill et al., 2015 and York et al.,
2016). Space-borne lidar observations are however affected by some drawbacks, as: 1) limited temporal resolution and
spatial coverage (the CALIPSO spatial distance between two consecutive ground tracks is about 1000 kilometers and
each track has a footprint of 70 m), 2) the contamination of unscreened clouds, and 3) difficulties in quantitatively
characterizing the aerosol properties in the lowermost troposphere (Pappalardo et al., 2010). Ground-based lidar
networks thus still represent key tools in integrating space-borne observations to study aerosol properties and their 4D
distribution. An example of these networks is the European Aerosol Research LIdar NETwork (EARLINET,
http://www.earlinet.org/), which, since 2000, provides an extensive collection of ground-based data for the aerosol
vertical distribution over Europe (Bösenberg et al., 2003, Pappalardo et al., 2014). The advanced multi-wavelength
elastic and Raman lidars employed in this network allows independent retrieval of aerosol extinction ($\alpha_a$) and
backscattering coefficient ($\text{\ss}_a$) profiles. Yet, despite their unsurpassed potential in data accuracy, advanced lidar

networks such as EARLINET have the unsolved problems of the sparse spatial and temporal sampling and of the complexity of operations. In fact, the typical distance between the EARLINET stations is of the order of several hundreds of kilometers and regular measurements of EARLINET are only performed on selected days of the week (Mondays and Thursdays) and for a few hours (mainly at nighttime, due to low signal-to-noise ratio of the Raman signal in daylight). Furthermore, these systems are complicated to be operated, require specific expertise and are therefore unsuitable for operational applications.

Nowadays, hundreds of single channel Automated Lidar Ceilometers (ALCs) are in operation over Europe and worldwide. Although such simple lidar-type instruments were originally designed for cloud base detection only, the recent technological advancements make now these systems reliable and affordable for aerosol measurements, increasing the interest in using this technology in different aerosol-related sectors (e.g. air quality, aviation security, meteorology, etc.). Recent studies showed that the ALC technology is now mature enough to be used for a quantitative evaluation of the aerosol physical properties in the lower atmosphere (Wiegner, M. and A. Geiß, 2012, Wiegner et al., 2014) and the exploitation of the full potential of ALCs in the aerosol remote sensing is a current matter of discussion in the lidar community (e.g. Madonna et al., 2015, 2018). The evaluation of ALC capabilities at providing quantitative aerosol information is among the main objectives of the EU COST Action ES1303, TOPROF (Towards Operational ground-based PROFiling with ALCs, doppler lidars and microwave radiometers). An effort in this direction is also underway in the framework of E-PROFILE, one of the observation programs of the EUropean METorological services NETwork (EUMETNET). In fact, several ALC stations are progressively joining E-PROFILE to develop an operational network to produce and exchange ALC-derived profiles of attenuated backscatter. A recent project funded by the EU LIFE+ program (DIAPASON, Desert-dust Impact on Air quality through model-Predictions and Advanced Sensors ObservatioNs, LIFE+2010 ENV/IT/391) also prototyped and tested an ALC system with an additional depolarization channel, capable of discriminating non spherical aerosol types, such as desert dust (Gobbi et al., 2018). Such upgraded ALC systems could further improve the capabilities of the operational aerosol profiling in a near future.

Given the necessity to couple advancement in instrumental technology with tools capable of translating raw data into a robust, quantitative and usable information, we propose and characterize here a methodology to be applied to elastic backscatter lidars and/or ALC measurements to retrieve, in a quasi-automatic way, vertically-resolved profiles of some key aerosol optical and microphysical properties. This effort is intended to contribute better exploit these systems potential in integrating data collected by more advanced lidar systems/networks. In particular, the ALC-derived aerosol properties addressed in this study are: backscatter ($\beta_a$, km$^{-1}$ sr$^{-1}$), extinction ($\alpha_a$, km$^{-1}$), surface area ($S_a$, cm$^2$ cm$^{-3}$) and volume ($V_a$, cm$^3$ cm$^{-3}$), the latter being convertible into aerosol mass concentration ($\mu$g m$^{-3}$) via assumption on particle density. To this purpose, we developed a numerical aerosol model to perform a large set of aerosol scattering simulations. Based on results from this numerical model, we derive mean functional relationships linking $\beta_a$ to $\alpha_a$, $S_a$ and $V_a$, respectively. These relationships are then applied in the ALC data inversion and analysis. A similar approach was applied in past studies for lidar-based investigations of stratospheric (Gobbi, 1995) and tropospheric aerosols (maritime, desert dust and continental type) at visible and UV lidar wavelengths, (Barnaba and Gobbi, 2001, Barnaba and Gobbi, 2004a, hereafter BG01, BG04a, respectively, Barnaba et al., 2004). Here we extend this approach to all the Nd:YAG laser harmonics commonly used by both advanced lidars and ALC systems (i.e. 355, 532, 1064 nm wavelengths) and specifically address an 'average-continental' aerosol type, intended to represent clean-to-moderately polluted continental aerosol conditions (see Section 2.1). In fact, despite the known differences that can be encountered

across the continent both in the short and the long-term (e.g., Putaud et al. 2010), this aerosol type is expected to
climatologically dominate over most of Europe.
Overall, this investigation is organized as follows: in Section 2 we describe the aerosol model set up to reproduce clean
to moderately polluted continental conditions, and the Monte Carlo methodology followed to compute the
corresponding bulk optical and physical properties. Section 3 shows and discusses the results of the numerical model,
and presents the model-based, mean functional relationships linking the different variables at 355, 532 and 1064 nm. In
Section 4 we evaluate both the model simulations capability to reproduce real measurements in continental aerosol
conditions, and the capability of the model-based ALC inversion approach to derive quantitative geophysical
information. The EARLINET database was used for the first task while tests on the accuracy of the model-based ALC
inversion were performed evaluating both the ALC-derived aerosol volume and optical thickness (AOT, i.e. the
vertically integrated aerosol extinction). To this purpose we applied the proposed methodology to three ALC systems
operating within the Italian Automated LIdar-CEilometer NETwork (ALICE-NET, www.alice-net.eu). In particular, the
ALC-derived AOT and aerosol volume (plus mass) were compared, respectively, to reference measurements performed
by ground-based sun photometers and in situ aerosol instruments (optical counters and PM10 samplers).
Section 5 summarizes the developed approach and main results, critically examining strengths and weaknesses. It also
includes discussion on the perspectives of the application of this (or similar) methodology in operational ALC
networks.

## 2    The aerosol model

A numerical aerosol model was set up to calculate mean functional relationships between the aerosol backscatter ($\beta_a$)
and some relevant aerosol properties, as $\alpha_a$, $S_a$ and $V_a$. This is done in a two-step procedure (Figure 1), following an
approach similar to that developed by BG01 and BG04a:
1) Generation of a large set (here 20000) of aerosol optical and physical properties by randomly varying, within
appropriate ranges, the microphysical parameters describing the aerosol size distribution and composition (blue box in
Fig. 1);
2) Based on results at point 1), determination of mean functional relationships linking such key variables (grey box in
Fig. 1).
The following Section describes rationale and set-up of the first step, the second one being thoroughly discussed in
Section 3.

### 2.1    Selection of the aerosol microphysical parameters

As anticipated, an average 'continental' aerosol type (i.e. describing clean to moderately polluted continental
conditions, e.g. Hess et al., 1998) was targeted in this study, this being the aerosol type expected to dominate over
Europe. Based on a scheme originally proposed by d'Almeida et al. (1991), and on a large set of following
observational evidences (e.g. Van Dingenen et al., 2004), in this work its size distribution is described as an external
mixture of three size modes. These are (in order of increasing size range): 1) a first ultrafine mode; 2) a second fine
mode, mainly composed of water-soluble particles; 3) a third mode of coarse particles.
A three-mode lognormal size distribution described by Eq. (1) is employed to this purpose:
$$n(r) = \frac{dN}{d \log r} = \sum_{i=1}^{3} \frac{N_i}{\sqrt{2\pi} \log \sigma_i} exp \left[ -\frac{(\log r - \log r_i)}{2(\log \sigma_i)^2} \right]$$ (1).
In Eq. (1), $r_i$, $\sigma_i$ and $N_i$ are respectively the modal radius, the width and the particle number density of the $i^{th}$ aerosol
mode (i = 1, 2, 3). At each computation, $r_i$ and $\sigma_i$ are randomly chosen within a relevant variability range. Values of $N_i$
are conversely obtained by firstly randomly choosing the total number of particles, $N_{tot}$, to be included in the whole size
distribution ($N_{tot} = N_1 + N_2 + N_3$), and then by applying specific rules for the number mixing ratio, $x_i = N_i/N_{tot}$, of each
component to this total. To reproduce clean to moderately polluted continental conditions, the value of $N_{tot}$ is made
variable between $10^3$ and $3\times10^4$ cm$^{-3}$ (e.g. Hess et al., 1998; Van Dingenen et al., 2004). Being the result of different
sources/processes, the three modes are also assumed to have a different composition, this impacting the optical
computations through the relevant particle refractive index ($m_i$), with both its real and imaginary component ($m_i = m_{r\_i}$
$- i \times m_{im\_i}$). The Mie theory for spherical particles of radius $r_i$ and refractive index $m_i$ is then used to compute the
extinction and backscatter coefficients (see below).
A description of the assumptions made for each mode and relevant parameter, mostly based on literature data (Table 1),
is given hereafter, the summary of the relevant variability chosen for each parameter being provided in Table 2.
1) First Mode
This ultrafine mode is the one more directly simulating fresh, anthropogenic emissions. The number mixing ratio $x_{i=1}$
($N_{i=1}/N_{tot}$) of this mode is let variable between 10% (rural conditions, Van Dingenen et al., 2004) and 60% (more
polluted conditions, Hess et al., 1998). The variability of its modal radius ($r_1 = 0.005 - 0.03$ µm) is chosen to include
from nucleation mode particles to Aitken mode particles. To take into account the wide variability of species within this
ultrafine mode, from non-absorbing (e.g., inorganic particles) to highly absorbing materials (e.g. black carbon), wide
ranges of variability has been set for its refractive indexes (at $\lambda$=355 nm: $m_{r\_1}$ in the range 1.40 - 1.8, and $m_{im\_1}$ in the
range 0.01 – 0.47, see Table 2 for the corresponding values at $\lambda$=532 and 1064 nm).
2) Second mode
The second aerosol mode accounts for 40-90% of $N_{tot}$, with (dry) $r_2$ between 0.03 and 0.1 µm. Its composition ($m_{r\_2}$,
and $m_{im\_2}$) is also made highly variable so to include water soluble inorganic and organic particles (Hess et al., 1998;
BG04a; Dinar et al. 2008). In this case, at $\lambda$=355 nm, $m_{r\_2}$ is in the range 1.40 - 1.7 and $m_{im\_2}$ is in the range 0.0001 –
0.01 (Table 2).
3) Third mode
This coarser aerosol mode (modal radius $r_3$ in the range 0.3 – 0.5 µm) is mainly intended to account for soil derived
(dust-like) particles that are a primary continental emission. A quite narrow variability is thus fixed for its $m_{r\_3}$ and
$m_{im\_3}$ values (1.5 – 1.6 and 0.0001 – 0.01, respectively at 355 nm). The relevant number mixing ratio $x_3$ ($N_3/N_{tot}$) is set
variable between 0.001% and 0.5 %, this mode contributing mostly to the total aerosol volume (thus mass) rather than
to the total number of particles.
As mentioned, refractive indexes were also made wavelength dependent, as this feature is also typically observed as
linked to the different particle composition. In particular, for the second mode (water-soluble particles) we include an
increase with the wavelength of the upper boundary values of $m_{im\_2}$ and a decrease of $m_{r\_2}$ at $\lambda$ =1064 nm (d'Almeida et
al., 1991). For the (dust-like) third-mode particles, the upper boundary values of $m_{im\_3}$ are set to decrease with
increasing wavelengths (Gasteiger et al., 2011, Wagner et al., 2012).
For convenience, the aerosol parameters boundaries summarized in Table 2 refer to dry particles and to ground level.
However, the effect of a variable RH, its variability with altitude as well as the generally observed decrease of particle
number with altitude is also considered in the model. More specifically, the number of particles in each mode, $N_i$, and
RH are both made altitude-dependent through the following equations (Patterson et al., 1980, BG01):
$$N_i(z) = N_i(0) \times \exp\left(\frac{-z}{H_i}\right),$$        (2)
$$RH(z) = 70 \times \exp\left(\frac{-z}{5.5\ km}\right) \times (1 + dRH),$$        (3)
the altitude z being variable here between 0 and 5 km. $N_i(0)$ and $H_i$ in eq.2 are the number of particles at the ground and
the scale height for each mode, respectively.
To describe the altitude effect, in eq. (2) an exponential decrease with height of the particle number density is assumed.
To rescale the particle number density of the different modes, $H_{i=1=2}$ is set equal to 5.5 km (Barnaba et al., 2007) while
$H_{i=3}$ (coarse particles) is set to 0.8 km (Barnaba et al., 2007). In eq. (3), the additional term (1+dRH) allows a further
variability with respect to the mean RH(z) profile assumed, here dRH is randomly chosen between -60 and +60). Values
of RH greater than 95% are discarded to avoid divergence.
Additionally, while first and third modes are assumed to be water insoluble, the second mode (i=2) is fully hygroscopic.
Aerosol humidification is thus considered to act on both particle size and refractive indices of the second aerosol mode
(e.g., BG01), as:
$$r_{2\_RH} = r_{2\_0} \sqrt{\frac{2 - 0.01RH}{2(1 - 0.01RH)}} \quad ,$$        (4)
$$m_{2\_RH} = m_w + (m_{2\_0} - m_w)\left(\frac{r_{2\_0}}{r_{2\_RH}}\right)^3,$$        (5).
In eq. 4 and 5, $r_{2\_RH}$ and $m_{2\_RH}$ are the RH-corrected modal radius and refractive index for the second mode,
respectively; $r_{2\_0}$ and $m_{2\_0}$ are the particle dry modal radius and refractive index, respectively; $m_w$ is the water refractive
index (assumed as equal to $1.34 - i7e^{-9}$, $1.33 - i1.3e^{-9}$ $1.33 - i2.9e^{-6}$ at 355, 532 and 1064 nm, respectively).
Finally, following Barnaba et al. (2007), an increase of the width of the size distribution with altitude (eq. 6) has been
introduced for the first and second aerosol mode:
$$\sigma_{1,2}(z) = \sigma_{1,2,z0} \times \exp\left(\frac{z}{30}\right).$$        (6)
In fact, Barnaba et al., (2007) showed that this was necessary to better reproduce the observed decrease of the Lidar
Ratio (LR) with altitude, and likely related to a broadening of the particle size distribution with aging.
Once the value of each microphysical parameter is randomly selected within its relevant variability range, and once
corrections are applied following eqs. (2) – (6), each resulting aerosol size and composition-resolved distribution is used
to compute the aerosol $S_a$ and $V_a$, as well as to feed a Mie code (assumption of spherical particles, Bohren and
Huffman, 1983) to compute $\beta_a$, and $\alpha_a$, (BG01, see also Fig. 1). Overall, the equations used are as follows:
$$\beta_a = \int Q_{bsc}(r, \lambda, m)\, \pi r^2\, \frac{dN}{d\log r} \frac{1}{r \ln 10}\, dr$$        (7)
$$\alpha_a = \int Q_{ext}(r, \lambda, m)\, \pi r^2\, \frac{dN}{d\log r} \frac{1}{r \ln 10}\, dr$$        (8)
$S_a = 4\pi \int r^2 \frac{dN}{d \log r} \frac{1}{r \ln 10} dr$ (9)
$V_a = \frac{4}{3}\pi \int r^3 \frac{dN}{d \log r} \frac{1}{r \ln 10} dr,$ (10)
where $Q_{bsc}$ ($r_i$, $\lambda$, $m_i$) and $Q_{ext}$ ($r_i$, $\lambda$, $m_i$) are, respectively, the backscatter and the extinction efficiencies. As mentioned,
the optical computations are made at the three different wavelengths: 355, 532, 1064 nm (i.e., those of Nd:YAG laser
harmonics, the most common wavelengths used by ground-based and space-borne aerosol lidars).
Since in our simulations the third aerosol mode is intended to represent dust-like particles, an empirical correction for
non-sphericity is also applied to the Mie-derived optical properties of this mode. This procedure is based on BG01,
which uses the results of Mishchenko et al. (1997) obtained for surface-equivalent mixtures of prolate and oblate
spheroids.
**2.2    Model simulation results**
In Figure 2 we show the results of 20000 simulations of continental aerosol optical and physical properties derived
randomly varying the relevant aerosol size distributions and compositions as described in the previous section. In
particular, the results for $\alpha_a$, $S_a$ and $V_a$ are shown as a function of $\beta_a$ in Figure 2a, b, c (blue crosses) referring to $\lambda$ =
1064 nm. For each variable (A), average value per bin of $\beta_a$ and relevant standard deviations (<A> ± dA) are shown as
red dots and vertical bars, respectively. Note that 10 equally spaced bins per decade of $\beta$ have been considered, and that
<A> ± dA are only shown for bins containing at least 1% of the total number of pairs. Corresponding relative errors
(dA/<A>) are depicted in Figure 2d, e, f. Some sensitivity tests of these model outputs to the variability of the input
microphysical parameters employed are provided in Appendix A.
Based on these results, at step-two of the procedure (see scheme in Figure 1), we derive aerosol-specific mean
relationships linking aerosol extinction, surface area and volume ($\alpha_a$, $S_a$ and $V_a$) to its backscatter ($\beta_a$). To this purpose,
we used a seventh-order polynomial fit in log-log coordinates. The choice of a seventh-order polynomial fit was made
for homogeneity with BG01 and BG04a. These relationships are shown as green lines in Figure 2a, b, c while the
relevant fit parameters are reported in Table 3 referring to $\lambda$ = 1064 nm (fit parameters related to computations at $\lambda$ =
355 and 532 nm, are given in Table A1 and Table A2, Appendix B).
The red vertical bars of Figure 2 also highlight the ranges of $\alpha_a$, $S_a$ and $V_a$ which are statistically significant, i.e. those in
which, at $\lambda$ = 1064 nm, the model provides at least 1% of the total points per corresponding bin of $\beta_a$. These are: $10^{-4}$ -
$10^{-1}$ km$^{-1}$, $10^{-7}$ - $10^{-5}$ cm$^2$/cm$^3$ and $10^{-13}$ - $10^{-10}$ cm$^3$/cm$^3$, for $\alpha_a$, $S_a$ and $V_a$ respectively, corresponding to the backscatter
range $9\times10^{-5} \leq \beta_a \leq 4\times10^{-3}$ km$^{-1}$ sr$^{-1}$. In terms of aerosol properties variability, the relative errors associated to $\alpha_a$ and $V_a$
show almost no dependence on $\beta_a$, with values between 30% and 40%. Conversely, the modeled aerosol surface area
exhibits a larger dispersion, with relative error values spanning the range 40% - 70%, and decreasing as $\beta_a$ increases.
A key parameter for the inversion of lidar signals is the so-called Lidar Ratio (*LR*), i.e. the ratio between $\alpha_a$ and $\beta_a$
(Ansmann et al., 1992). In Figure 3 we thus show the results of our simulations in terms of LR vs $\beta_a$ at the three $\lambda$ (355,
532 and 1064 nm, Figure 3a, b, c, respectively) and relevant dLR/LR values (Figure 3d, e, f, respectively). The color
code is the same of Fig. 2. Additional horizontal black lines have been inserted representing values (solid central lines)
of the 'weighted-LR' ± 1 standard deviation (dotted side lines), i.e. the LR weighted by the number of simulated points
in each considered backscatter bin. The 'weighted-LR' values derived at 355, 532 and 1064 nm, are 50.1 ± 17.9 sr, 49.6
± 16.0 sr and 37.7 ± 12.6 sr, respectively. Figure 3 also allows showing that the statistically significant regions of
simulated backscatter values shifts towards smaller values with increasing $\lambda$ (e.g. at $\lambda = 355$, the $\beta_a$ extending regions is
$4\times10^{-5} - 2\times10^{-2}$ km$^{-1}$ sr$^{-1}$, whereas, at 532 nm, it ranges between $2\times10^{-5} - 1\times10^{-2}$ km$^{-1}$ sr$^{-1}$). Furthermore, Figure 3
reveals a quite different shape of the LR vs $\beta_a$ functional relationships (green curves) at different wavelengths. At 355
and 532 nm the curve is concave, with quite similar LR maxima of the fitting curve (54.3 and 53.8 sr at approximately
$\beta_a = 4\times10^{-4}$ km$^{-1}$ sr$^{-1}$ and $2\times10^{-3}$ km$^{-1}$ sr$^{-1}$, respectively). At 1064 nm the curve is conversely monotonic, with a flex
point at $\beta_a = 3\text{-}4\times10^{-4}$ km$^{-1}$ sr$^{-1}$. A larger data dispersion also characterizes the results at $\lambda = 355$ and 532 nm (LR values
from 10 to 90 sr) in comparison to $\lambda = 1064$ nm (LR in the range $18 - 80$ sr, except for a minor number of outliers).
This translates into different LR relative errors at UV, VIS and infrared (IR) wavelengths. At 1064, dLR/LR slightly
decreases for increasing backscatter, with values around 35%. At the shorter wavelengths, it increases as a function of
$\beta_a$, with a large (>40%) relative error for values of $\beta_a > 2\times10^{-3}$ km$^{-1}$ sr$^{-1}$.
To insert our results into a more general context, we compared the derived, model-based weighted-LR values to some
LR data reported in the literature (Table 4). In particular, we selected some of the works using the aerosol model
developed to invert the Calipso lidar data (Omar et al., 2009). This latter considers six different aerosol sub-types: clean
continental (CC), clean marine (CM), dust (D), polluted continental (PC), polluted dust (PD), and smoke (S). Our
model-derived LR at 532 nm falls in the middle of the range (35-70 sr) fixed by the Calipso CC and PC aerosol classes.
The work by Papaggianopoulos et al. (2016), in which the LR values are adjusted accordingly to EARLINET
observations, reports a LR range at 532 nm of 47-62 sr. At the same wavelength, the aerosol range defined by the
LIVAS climatology (LIdar climatology of Vertical Aerosol Structure for space-based lidar simulation studies, Amiridis
et al., 2015), is 54-64 sr. In both cases, our model seems to be closer to the LR values of CC aerosol type, which is
compatible to our intention to simulate clean-to-moderately polluted continental aerosol type. At 532 nm, our LR value
is also reasonably in between the CC and PC LR values derived by Omar et al. (2009), but again closer to the CC LR
value. The very small decrease of LR values between 532 and 355 nm estimated by LIVAS for the CC aerosol is also
consistent with our results. Similarly, our model predicts a lower mean LR in the near IR with respect to the green, in
agreement with results of Amiridis et al. (2015) in CC conditions and not to those in polluted conditions. Table 4 also
includes the continental aerosol LR values estimated in the work of Düsing et al. (2018) through comparison between
airborne in situ and ground-based lidar measurements. Our model is in good agreement with their LR values at 355 and
532 nm. At 1064 nm, the algorithm developed by Düsing et al. (2018) provided a value of LR around 15 sr. On the
other hand, in the same study the authors found that, rather, a value of LR = 30 sr gives the better accord between their
Mie and lidar-based $\alpha_a$, this value being closer to our model-derived one at 1064 nm (LR =37.7). The difference
between these two values is explained by the authors to be probably due to the estimation of the aerosol particle number
size distribution, a critical parameter for a reliable modeling of aerosol particle backscattering.
As a last added value of the outcome from our model-based results, we derive here and provide in Table 5 extinction-to-
volume conversion factors, $c_v = V_a/\alpha_a$ (e.g., Ansmann et al., 2010) at three different wavelengths (355, 532 1064 nm),
and compare these to similar outcomes from other studies. To our knowledge, values of continental particles $c_v$ at three
wavelengths are only available in Mamouri and Ansmann (2017). Note that $c_v$, is also proportional, through the particle
density $\rho_a$, to the inverse of the so-called 'mass-to-extinctions efficiency' (MEE, i.e. $\alpha_a/(V_{a*}\rho_a)$) a parameter important in
several aerosol-related applications (e.g. the estimation of particulate matter mass from satellite AOT or in modules of
global circulation and chemical transport models to compute aerosol radiative forcing effects, Hand and Malm, 2007).
For convenience, model-derived MEE values are also included in Table 5.

## 3    Evaluation of the model performances and potential of its application

In this section, we evaluate the capability of the model results to reproduce 'real' aerosol conditions and explore the potential of the proposed model-based ALC inversion in producing quantitative geophysical information. In particular:

- In Section 3.1 we compare our simulations to real observations of independent backscatter and extinction coefficients made by different EARLINET Raman lidars (Bösenberg et al., 2001, Pappalardo et al., 2014).

- In Section 3.2, our model results are used to invert measurements acquired by some ALCs systems operating within ALICE-NET, which networks several ALCs systems (Nimbus CHM15k by Lufft) located across Italy and run by Italian research institutions and environmental agencies. Here we use data from some of these systems to derive the aerosol optical and physical properties (e.g. the aerosol optical thickness, AOT, and the aerosol volume and mass).

### 3.1    Comparison of the modelled aerosol optical properties to EARLINET measurements

As mentioned EARLINET Raman stations perform coordinated measurements two days per week following a schedule established in 2000 (Bösenberg et al., 2003). Overall, the EARLINET database includes the following categories: 'climatology', 'CALIPSO', 'Saharan dust', 'volcanic eruptions', 'diurnal cycles', 'cirrus', and 'others' (forest fires, photo smog, rural or urban, and stratosphere). To be comparable to our results, we used EARLNET $\beta_a$ and $\alpha_a$ coefficients at 355 nm and at 532 nm within the quality assured (QA) 'climatology' category (Pappalardo et al., 2014). However, note that additional data filtering was necessary to screen out residual, likely unreliable values within this QA-'climatology' category. In particular, we only selected those EARLINET QA data further satisfying the following criteria:

-    $\beta_a$ and $\alpha_a$ coefficients evaluated independently, i.e. only obtained using the Raman method (Ansmann et al., 1992);
-    $\beta_a$ and $\alpha_a > 0$;
-    LR < 100;
-    Relative errors on $\beta_a$ and $\alpha_a < 30\%$.

Then, we selected those sites in Europe expected to be mostly impacted by 'continental' aerosols and having the largest datasets (e.g., at least 100 points) at 355 and 532 nm. Overall, 5 sites satisfied these conditions (Table 6), and namely Madrid (Spain), Potenza and Lecce (Italy), Leipzig and Hamburg (Germany). Finally, being interested in continental conditions here, we filtered out those measurements dates affected by desert dust at the measuring sites, i.e. we removed from our 'model-measurement comparison data set' all the dates within the EARLINET 'climatology' category also belonging to the EARLINET 'Saharan dust' category.

Figure 4 depicts the results of the model-measurements comparison at the sites fulfilling our requirements in terms of LR vs $\beta_a$ at $\lambda$=355 nm (the corresponding results at $\lambda$=532 nm, including Madrid in place of Hamburg, are given in Appendix C, Figure C1). The colored area represents the model-simulated data range, while the color code indicates the absolute number of simulated values (i.e. counts) in each $\beta_a$ - LR pair. The EARLINET-measured values are reported as black open circles. Note that, being the model simulations performed over an altitude range 0-5 km (see Section 2.1) only those simulations corresponding to the altitude range ($\Delta z$) covered by the measurements at each EARLINET station was taken into account here. Figure 4 shows the model results to well encompass the measured LR vs $\beta_a$ data, with few measurements outside the modeled range (most of the exceptions are found for Potenza). Statistically, the

highest number density of simulated data well fits the observations, with the exception of Hamburg (Figure 4a), which
however has the lowest number of measured data (it is not an EARLINET station any longer, see Table 6).
In Figure 5 the previous results at $\lambda$=355 nm are converted in terms of 'mean' LR per bin of $\beta_a$ for both model (blue)
and observations (red, again, only $\beta_a$ bins containing at least 1% of the total modeled data were considered). This view
shows that there is a general good agreement between the modeled and the measured LR values, and in their variation
with $\beta_a$. Some major deviations are found for Potenza and are further discussed in the following. The model-
measurements accordance shown in Figure 5 was evaluated in quantitative terms by computing mean LR relative
differences at both $\lambda$ = 355 and 532 nm, i.e., we derived ([(LR$_{mod}$- LR$_{meas}$)/ LR$_{meas}$]*100) values, where LR$_{mod}$ and
LR$_{meas}$ are the lidar ratio values computed by model and derived by lidar measurements, respectively. These values are
reported in Table 7 for each considered EARLINET station, together with the measurements-based mean LR in each
observational site (computed weighting the number of observations per $\beta_a$-spaced bins).
Results in Figure 5 and Table 7 also give some hints on the capability of the aerosol type assumed (and its admitted
ranges of variability) to reproduce 'real' continental aerosol conditions in different sites across Europe. In fact, the four
continental sites selected with our criteria are still expected to be partially impacted by different aerosol types.
- A good agreement between the model and the observations in terms of LR mean values is found for Hamburg (Figure
5a), with mean LR differences of the order of 5% (Table 7). Still, the measured LR values have a high variability and
their distribution is positioned towards high values of $\beta_a$ ($1\times10^{-3}$ to $4\times10^{-3}$ km$^{-1}$ sr$^{-1}$). This could be due to the presence
of different aerosols types as slightly polluted marine and polluted aerosol (Matthias and Bösenberg, 2002).
- A good accord for Leipzig (Fig. 5c) also indicates that this site is mostly dominated by 'pure' continental particles. In
fact, the distribution of observed LR points in Fig. 4, which covers $\beta_a$ values ranging from $2\times10^{-4}$ to $3\times10^{-3}$ km$^{-1}$ sr$^{-1}$, is
well centered to the modeled simulations highest density (counts > 40). Table 7 shows that at both wavelengths mean
discrepancies with LR measurements keep well below 10%.
The highest differences in Fig. 5 are found in some southern Europe EARLINET sites:
- In Lecce (Fig. 5b), the best agreement between model and observations is found for the lowest values of $\beta_a$ (between
$9\times10^{-4}$ to $1\times10^{-3}$ km$^{-1}$ sr$^{-1}$, see Table 7). Also, the increase from 10% to 18% in the discrepancies at 355 and 532 nm
indicates some model problems in correctly reproducing the spectral variability of the optical properties, suggesting
some mismatch between modeled and real aerosol sizes in this site (see discussion below).
- In Potenza (Fig. 5d), a significant difference between the mean LR curves emerges for $\beta_a$ values > $6\times10^{-4}$ km$^{-1}$ sr$^{-1}$ ,
with observed LR values lower than those simulated here.
These discrepancies could be due to the influence of marine aerosols at both stations (De Tomasi et al., 2006, Mona et
al., 2006, Madonna et al., 2011), which is expected to produce lower LR values for high values of $\beta_a$ (e.g. BG01). In
fact, Madrid shows better performances, with dLR/LR comparable to those in Leipzig.
To provide some insight into the reasons of the model-measurements differences at LC and PO sites, some specific
model sensitivity tests have been performed and are reported in Appendix D. In particular, for Lecce, we found that
better agreement between the observed and simulated LR vs $\beta_a$ behavior at 355 nm is obtained by reducing the
variability range of N$_{tot}$ (from 500 - 10000 cm$^{-3}$ to 500 - 5000 cm$^{-3}$ at ground). This indicates that LC is likely affected
by cleaner continental aerosol type conditions. The sensitivity simulations done for understanding the mismatches with
Potenza measurements show that an extension of the variability range of the coarse mode radius is needed to reproduce
the observed decrease of LR for increasing backscatter (Figure 5d). This suggests the presence of coarse particles larger
than those assumed in such clean continental environment (Appendix D). This is compatible with the suspect of marine
air contamination, although at this stage we are not able to exclude additional contamination of coarser particle of soil
origin.
Overall, mean LR differences between our 'average-continental' model and data at selected continental sites in Europe
keep lower than 20% (Table 7), this indicating it reasonably well reproduces the clean-to-moderately polluted
continental aerosol conditions we intended to simulate.
**3.2     Model results application to Nimbus CHM15-k ALC measurements**
To test and validate the model-based inversion methodology, we used the derived functional relationships (Section 2.2)
to invert and analyze the measurements of some ALICENET ALCs (Lufft CHM15k systems). These instruments are
biaxial ceilometers that emit laser pulses at 1064 nm (Nd:YAG-laser, class M1) with a typical pulse energy of 8 μJ and
a pulse repetition rate of about 6500 Hz. The instruments have a specified range of 15 km and full overlap at around
1500 m (Heese et al., 2010). The manufacturer provides the overlap correction functions ($O(z)$) for each system. As
shown recently by Wiegner and Geiß (2012) and Wiegner et al. (2014), a promising strategy to retrieve the aerosol
backscatter coefficient from ALC measurement is adopting the forward solution of the Klett inversion algorithm (Klett
1985). This solution requires a known calibration constant of the system (i.e. absolute calibration, $c_L$) and an
assumption on the LR. The advantage with respect to the backward solution is that calibration is not affected by the low
SNR in the upper troposphere and it is needed occasionally. Furthermore, starting close to the surface, the data retrieval
allows resolving aerosol layers in the boundary layer even if their optical depth is high. The forward solution of the
Klett inversion algorithm is thus adopted here. For convenience, we report here the equations used within our procedure
to obtain $\beta_a$ from ALC measurements, which are also described in Wiegner and Geiß (2012, equations 1 – 3):
$$\beta_a(z) = \frac{Z(z)}{LR\,N(z)} - \beta_m(z) \tag{11}$$
with
$$Z(z) = LR\,z^2 P(z) exp\left[-2\int_0^z (LR\,\beta_m - \alpha_m)dz'\right] \tag{12}$$
and
$$N(z) = c_L - 2\int_0^z Z(z')dz'. \tag{13}$$
Here, $\beta_m$ and $\alpha_m$ are the molecular backscatter and extinction coefficients calculated from climatological, monthly air
density profiles and $z^2 P(z)$ is the ALC range ($z$) corrected signal ($P$) (also referred to as RCS), that is the raw data
obtained by the considered ALCs. As anticipated, knowledge of the calibration constant $c_L$ is needed to solve eq. 13
(and thus 11, forward solution). In our analysis of ALC daily records, the constant $c_L$ has been obtained by the
"backward approach" (Rayleigh calibration) applied to night-time, cloud-free ALC signal averaged over 1 or 2 hours at
75 m height resolution. This allows for using the best $c_L$ retrieval (that is the night-time, lowest noise one), in the
forward solution of the lidar equation, which guarantees operating over the best signal to noise range of the ALC signal.

### 3.2.1 Model-based retrieval of aerosol optical properties

Operatively, inversion of the aerosol properties, $\alpha_a(z)$ and $\beta_a(z)$, is performed using an iterative technique, since we need to correct the backscatter signal at each altitude z for extinction losses. The iterative procedure is stopped when convergence in the integrated aerosol backscatter (IAB=$\Sigma_0^{zcal}\beta_a(z)$) is reached (e.g. BG01). At each step, aerosol extinction is derived using the functional relationship $\alpha_a = \alpha_a (\beta_a)$ of Table 3.

An example of the outcome of this retrieval methodology is depicted in Figure 6. It shows the time-height (24h, 0 - 6 km) contour plot of $\alpha_a$ retrieved at 1064 nm during a whole day of measurements (June 26, 2016) performed by the ALICENET system of Aosta San Christophe (ASC, 45.8°N, 7.4°E 570 m a.s.l., Northern Italy, Figure 7a). Time and altitude resolutions are 1 min and 15 m, respectively. Note that ALC data are cloud-screened using the cloud mask of the Lufft firmware.

The aerosol optical thickness (AOT) is obtained vertically integrating the ALC-derived $\alpha_a(z)$ from the surface up to a fixed height $z_{AOT}$, above which the aerosol contribution is assumed to be negligible. In Figure 6, the ALC-derived AOT values at 1064 nm (pink curve, with a temporal resolution of 5 min) is superimposed to the extinction contour. Reference AOT values from a co-located sun-sky radiometer (a Prede POM-02 system) are shown by orange circles. These were extrapolated at 1064 nm from the instrument 1020 nm-channel using the Angström exponent derived fitting AOT values at all the radiometer wavelengths. This example illustrates the very good performances of our model-assisted inversion scheme, and the capability of this approach to extend to nighttime the (daylight-only) radiometer observations.

To evaluate the performances of our model-assisted retrieval of $\alpha_a(z)$ over a more statistically significant dataset, the same approach illustrated in Figure 6 was applied to a longer record in the ASC site, plus Nimbus CHM-15k ALC datasets from two additional ALICENET sites: San Pietro Capofiume (SPC, 44°39N, 11°37E, 10 m a.s.l.) and Rome Tor Vergata (RTV, 41.88°N, 12.68°E, 100 m a.s.l.). The location of the instruments is shown in Figure 7a (red circles), while some information on system types and site characteristics is given in Table 8. The data analyzed here were collected during the following periods: April 2015 – June 2017, June 2012 – June 2013 and February 2014 – September 2015, for ASC, SPC and RTV, respectively.

In those sites, reference AOTs were collected by three co-located sun-sky radiometer, and namely using two SKYNET Prede sun-sky radiometers at ASC and SPC (POM-02L and POM-02, respectively, www.euroskyrad.net) and an AERONET Cimel CE 318-2 instrument operational at RTV (https://aeronet.gsfc.nasa.gov, Rome Tor Vergata station, data level 2.0). Only AOT values between 0.01 and 0.2 at 1064 nm were considered. This range allows for excluding the data points with 1064 nm-AOT lower than the sunphotometer expected accuracy (dAOT=0.01) and those where we found aerosol extinction to cause significant deterioration of our ALC signal. Overall a total of 1237, 268, 850 AOT pairs were analyzed at ASC, SPC and RTV, respectively.

Also note that, although CHM-15k data are already corrected for the *O(z)* function provided by the manufacturer, the variation of the ALC internal temperature was shown to lead to *O(z)* differences up to 45% in the first 300 m above ground (Hervo et al., 2016). For this reason, in our analyses the lowest valid altitude of the CHM-15k for both the SPC and RTV systems was fixed to be about 400 m. A linear fit of the first two valid ALC points is then used to extrapolate $\alpha_a(z)$ down to the ground ($z_0$). Conversely, due to the optimal characterization down to the ground of *O(z)* provided by Lufft for the CHM-15k system installed at ASC, values at $z_0$ at this site are not those extrapolated but actually those

measured. The maximum altitude of aerosol extinction vertical integration to derive the AOT, $z_{AOT}$, was selected as the
first height above 4000 m where the range corrected signal (RCS) has a SNR < 1.
Results of the long-term AOT comparison are summarized in Figure 7 and Table 9. For each site under investigation,
Figure 7 shows the histograms of the AOT differences between the hourly-mean coincident AOTs as derived by the
ALCs and measured by the sun-photometers (red curve, corresponding AOT vs AOT scatter plots at the three
considered sites are given in Appendix E). To evaluate the advantage of our approach with respect to more standard
lidar inversions, we also computed AOT differences using two fixed-LR values. In particular, we used LR = 52 sr (i.e.
the value suggested by the E-Profile network, black lines) and LR = 38 sr (i.e. the weighted mean LR value derived
from our model, see Section 3, blue lines). Figure 7 shows that the best agreement is found at ASC. The distribution of
AOT difference has a maximum around 0 for each of the three inversions schemes, with very low dispersion. The full
width at half maximum, FWHM, is in fact around 0.015, and approximately 55% of the data are included in the interval
-0.01 – 0.01, which is even within the expected error of photometric measurement. For SPC and RTV, the red and blue
histograms are peaked around 0, whereas the black ones are shifted, with maxima around 0.01-0.02 and 0.02-0.03 for
SPC and RTV, respectively. These two sites have higher dispersion (FWHM = 0.03), and approximately 30% of the
data are included in the interval -0.01 – 0.01 for the red and blue histograms at both sites, which is probably due to the
different aerosol load affecting the different ALICENET stations. As pointed out by the low value of the average AOT
computed at ASC for the analyzed dataset (<AOT> = 0.027), low pollution levels generally characterize this site, with
some exceptions due to wind-driven aerosol transport from the nearby Po valley (Diémoz et al,2018a, 2018b this issue).
On the contrary, RTV (<AOT> = 0.044) and, especially, SPC in the Po Valley (<AOT> = 0.076) are characterized by
higher aerosol content and pollution levels, which explain the larger histogram dispersions. Note that the high frequency
of fog events in winter markedly reduces the number of analyzed AOT pairs at SPC site, while some desert dust
affected days at both SPC (e.g., Bucci et al., 2018) and RTV (e.g., Barnaba et al., 2017) were removed from our
datasets (no desert-dust affected dates in ASC).
Table 9 summarizes the long-term performances of the model-based procedure in deriving quantitative AOT from the
ALC systems at the three investigated sites. It includes values of the average differences between the ALC-derived and
sunphotometers-measured AOT (both bias, < dAOT >, and absolute difference <|dAOT|>, with associated standard
deviations) obtained using both the proposed model-based approach and the fixed-LR inversions. For SPC and RTV
sites, these numbers show that the best ALC–photometer accordance is reached when employing either the model-based
or the fixed LR=38 sr inversion scheme. In fact, these two approaches have similar performances in terms of mean
dAOT values (<|dAOT|> = 0.011, 0.013 and 0.013, 0.014 for SPC and RTV, respectively), mean percent error
(<|dAOT|> /<AOT> = 0.16, 0.19 and 0.31, 0.33) and a very low mean relative bias (<dAOT>/<AOT> = -0.043, 0.005
and 0.088, 0.11). On the other hand, the fixed LR=52 sr retrieval produces an overestimation of AOT in both SPC and
RTV (<dAOT>/<AOT> = 0.33 and 0.44) with larger discrepancies between retrieved and observed AOTs (<|dAOT|>
= 0.021 and 0.026, <|dAOT|>/<AOT> = 0.38 and 0.49). For the ASC site, due to the low aerosol content, the
differences among the inversion schemes are almost negligible.
Overall, for the three sites, the statistics over the long-term datasets employed showed good results of the model-based
approach with similar behavior of the retrievals with a fixed LR of 38 sr, while a fixed LR value of 52 sr produces an
overestimation of the AOT at SPC and RTV. As different sites have different (and not known a-priori) characteristic LR
values, these results highlight the potential of the model-based approach to derive quite accurate $\beta_a$ and $\alpha_a$ coefficients
without the need to choose and fix an arbitrary LR value.

### 3.2.2    Model-based retrieval of aerosol volume (and mass)

In this section we provide examples of the applicability of the proposed approach to derive air-quality relevant parameters. In particular, we use the ALC, $\beta_a$-retrieved data and the 7$^{th}$-order polynomial fit linking $\beta_a$ (at $\lambda$ = 1064 nm) to $V_a$  (see also Table 3 and Figure 2c) to derive the aerosol volume (and mass).

   The ALC-estimates were firstly compared to aerosol volume derived in situ at the ASC site by two different optical particle counters (OPCs) on 29$^{th}$ December 2016 and 5$^{th}$ September 2017. For the case of the 29$^{th}$ December 2016, a TSI Optical Particle Sizer (OPS) 3330 was employed. This instrument has 16 channels that can be programmed to provide the number concentration at different (and logarithmically spaced) diameter size ranges within the interval 0.3 - 10 µm. Further details can be found in the TSI manual (2011). For the case of the 5$^{th}$ September 2017, the Fidas®200s OPC was used. This spectrometer is able to retrieve high-resolution particle spectra (size measurements between 0.15 and 27 $\mu$ m, with 32 channels/decade, Pletscher et al., 2016). For both  dates, Figure 8 shows the time (x-axis, 24h) vs. height (left y-axis) contour plots of the ALC-based retrieval of the aerosol volume concentration (cm$^3$/cm$^3$). The OPC-derived aerosol volume concentration measured at ground-level is reported as a function of time (x-axis) on the right y-axis (grey curve). The corresponding ALC-derived volume concentration (integrating the ALC data between 0 and 75 m) is shown by a pink curve (same right y-axis). Daily mean volume concentration values derived by OPCs and by ALC are also plotted (grey cross and pink triangle, respectively). The horizontal bar in the upper part of the figure indicates the ranges of RH measured in-situ during the analyzed cases.

The OPC-to-ALC comparison is certainly affected by intrinsic factors, as differences on the atmospheric layer sampled (at ground and integrated between 0 and 75 m, for OPC and ALC, respectively) and on the probing methods (in-situ and remote sensing, dried air sampled by OPC and ambient conditions sampled by the ALC). Furthermore, as mentioned in Section 4.2.1, a major critical issue of ALC retrievals at low levels is the correction for the overlap function, which needs to be experimentally characterized and verified for each instrument.

These issues are visible in the given example of Figure 8. In fact, in the upper panel, the agreement between the ALC-derived and the TSI-OPC aerosol $V_a$ values is good between 0 and 7 UTC. In the following hours both instruments register an increase of the aerosol volume, although with some discrepancies in absolute values. Starting from 18 UTC, the ALC derives an aerosol volume concentration higher than the OPC one by a factor of 3-3.5. This disagreement could be related to both the presence/arrival of fine particles (<0.3 µm) not measured by the optical counter (see for example Diémoz et al., 2018a), or to aerosol hygroscopic effects (increase of volume associated to hygroscopic growth seen by ALC but not by the OPC which dries the air samples). This latter effect is confirmed by the large RH values (RH > 90%) measured after 18 UTC. The lower panel shows a good agreement between the ALC-derived and the Fidas OPC $V_a$ values, in particular until 4 UTC and after 16 UTC. Some differences emerge around 7 UTC and between 11 and 15 UTC, where the ALC volume is lower by a factor of 2 compared to the in situ Fidas $V_a$ values. The smaller minimum detectable size of the Fidas OPC instrument with respect to the OPS is likely the reason for the better accord between ALC and OPC $V_a$ values in this test date. In this case, the effect of RH seems to be less important, and indeed RH values keep lower than 90%.

In general, high RH values (RH >= 90%) are known to markedly affect the aerosol mass estimation from remote sensing techniques and its relationship with 'reference' PM2.5 or PM10 measurements methods, usually performed in dried conditions (e. g. Barnaba et al., 2010; Adam et al., 2012, Li et al., 2016, Li et al., 2017). This theme is also discussed in Diemoz et al. 2018a for the ALC measurement site of Figure 8. Nevertheless, even with the mentioned

limitations, results in Fig. 8 well show the potential of the developed method in providing sound values of aerosol
volume, and hence, mass, in average-RH regimes, giving support to more standard PM10 air quality monitoring.
To give a further example in this direction, the model–assisted retrievals of aerosol mass over a longer time period were
used to derive daily-mean aerosol mass concentrations (PM10), a measurement typical of air quality stations. To this
purpose, for the two-months period June-July 2012, we derived daily mean values of aerosol volume at the SPC site
using the functional relationships $V_a = V_a(\beta_a)$, and then converted these into mass (PM10) using typical values of
aerosol densities ($\rho_a$). Results are shown in Figure 9. It compares the daily average PM10 concentration measured in
situ at SPC by the Italian Regional Environmental Protection Agency (ARPA, red solid curve) and the model-assisted,
ALC-derived daily mass concentration obtained assuming both a fixed particle density $\rho_a = 2$ g/cm$^3$ (blue dotted curve),
and a range of it between 1.5-2.5 g/cm$^3$ (shaded area), this range covering approximately the typical $\rho_a$ values at the
SPC site. Yellow shaded areas indicate the presence of dust events (e.g. Bucci et al., 2018) that are excluded from the
results reported in the next paragraph.
More in detail, the daily-mean, ALC-derived mass concentrations were estimated in two steps: 1) estimation of hourly
mass values for the selected height; 2) computation of the daily values through the median of the hourly values. To
guarantee a good daily representativeness, the second step is applied only to those days in which at least 50% of the
hourly values is available in all the following temporal ranges: 00 - 05 UTC, 06 - 11 UTC, 12 - 17 UTC, 18 - 23 UTC.
Note that, due to the uncertainties associated to the $O(z)$ in the first hundreds of meters (as previously mentioned, the
ALC system at SPC has an old firmware, and its overlap function is not optimally characterized), we used the 225 m
level as more trustworthy to estimate ALC mass concentration. On the other hand, during the considered period of the
year (i.e. June and July), the comparison to ground-level PM10 at SPC is expected to be only slightly affected by this
height difference, particularly in daytime, due to the strong convection within the mixing layer. Possible exception
could be in nocturnal conditions when vertical gradients in the lowermost hundreds of meters can occur. However, our
statistical (3-year) ALC records show the mixing layer height at SPC to descend below 250 m only 4-5 hours per day in
July (usually between 22 and 3 UTC, i.e., when emissions are at a minimum). Overall, Figure 9 confirms a good
agreement between the ALC-derived and the ARPA reference PM10 values, with a correlation coefficient (R) of 0.64.
In fact, mean, absolute mean and relative differences, between the two series are: $<dPM10> = 2.3 \pm 6.0$ g/cm$^3$,
$<|dPM10|> = 4.8 \pm 4.3$ g/cm$^3$ and $<(dPM10/PM10)> = 0.14 \pm 0.27$. This agreement attests that SPC site can indeed be
considered an 'average' continental site and suggests the potential of this approach to derive information on aerosol
volume and mass. Still, due to the specificity of each site and to the limited period considered here, these results cannot
be taken as representative of all continental sites at all times. Further studies at different places and over longer time
periods would be necessary to better assess the uncertainty of the proposed retrieval, including uncertainties due to the
variability of 'continental' conditions (in terms of particle size distribution, compositions, hygroscopic effects, etc…),
but also of the instrument-dependent performances (e.g. overlap corrections, etc…).
**4    Summary and Discussion**
Thanks to their low construction/operation costs and to their capability at providing continuous, unattended
measurements, the use of automated-lidar-ceilometers (ALCs) for aerosol characterization has increased in the recent
years. Several numerical approaches were recently proposed to estimate the aerosol vertical profile either using
ceilometer measurement only, or coupling these with ancillary measurements (e.g., Stachlewska et al., 2010; Flentje et
al., 2010; Wiegner et al., 2012; Wiegner et al., 2014; Cazorla et al., 2017, Román et al., 2018).
This work proposes a methodology to retrieve key aerosol properties (as extinction coefficient, surface area and
volume, thus mass) from lidar/ALC measurements using in support the results from a specifically developed aerosol
numerical model to drive the retrievals. In particular, the numerical model uses a "Monte-Carlo" approach to simulate a
large set (20000) of aerosol microphysical properties intended to reproduce the variability of 'average' (clean-to-
moderately polluted) continental conditions, i.e., those expected to dominate over Europe. Based on the assumption of
particle sphericity, relevant computations of aerosol physical (surface area and volume, $S_a$ and $V_a$) and optical
(backscattering and extinction coefficients, $\beta_a$ and $\alpha_a$ through Mie scattering theory) properties were performed at three
commonly used lidar wavelengths (i.e., at the Nd:YAG laser harmonics 355, 532, 1064 nm). Fitting procedures of this
large set (20,000) of $\beta_a$ vs. $\alpha_a$, $S_a$ and $V_a$ data-pairs were then used to derive mean functional relationships linking $\beta_a$ to
$\alpha_a$, $S_a$ and $V_a$, respectively. The model's statistical uncertainties (i. e., those related to the variability of the
microphysical parameters used in input to the computations of the bulk physical/optical properties) associated to these
so-derived mean relationships were found to be within 30% and 40% for $\beta_a$ vs $\alpha_a$ and $\beta_a$ vs $V_a$, respectively, while $\beta_a$ vs
$S_a$ exhibits a larger dispersion (relative standard uncertainty of 40%-70%, depending on $\beta_a$). It is worth mentioning that
these are higher than those associated to the retrievals of aerosol bulk parameters using the complete set of Raman
lidars observations (three aerosol backscattering and two extinction coefficients, i.e., the so called 3+2 approach),
assuming, as in our case, no random uncertainty in the lidar input data. For example, Veseloski et al. (2012) found a
maximum uncertainty of 15% for particle volume and surface area estimation, in the case of 0% random uncertainty in
the lidar input data. Note however, that such multi-wavelength lidar systems are 10 to 20 times more expensive than
ALC systems, need to be operated by highly trained operators, and are rarely run all day round.
The model results also allowed exploring the expected dependence of the (continental aerosol) lidar ratio (LR) on $\beta_a$ at
355, 532 and 1064 nm, and in turn, the mean, 'weighted'-LR value at each wavelength (found to be 50.1 ± 17.9 sr, 49.6
± 16.0 sr and 37.7 ± 12.6 sr, at 355, 532 and 1064 nm respectively). Availability in literature of LR values at 1064 nm
are scarce and its monotonic increase with $\beta_a$ found in this work (Figure 3) suggests that the use of a fixed LR value for
the inversion of ALC signals should be done with caution and carefully evaluated case by case. A similar, non-
monotonic behavior characterizes the shapes of LR vs $\beta_a$ curve at 355 and 532 nm.
We tested the reliability of our model results in two ways: 1) the model numerical computations were compared to
'real' lidar measurements (specifically selected within the EARLINET database), and 2) the model-assisted retrievals of
aerosol optical (AOT) and physical ($V_a$, PM10) properties by real, operational ALC systems were compared to
corresponding 'reference' measurements performed by co-located, independent instrumentation.
In particular, in task 1) our simulations were compared to backscatter and extinction coefficients at 532 and 355 nm
independently retrieved by advanced Raman lidar systems operating at different EARLINET sites in Europe (namely
Hamburg and Leipzig in Germany, Madrid in Spain, Lecce and Potenza in Italy). The model simulations were found to
statistically well match the observations (Figures 4, 5 and C1). Mean discrepancies between model and measurement-
based LR were found to be lower than 20%, suggesting a good capability of the assumed aerosol model (and admitted
range of variability) to represent 'real', 'average continental' aerosol conditions in different sites across Europe. Some
differences emerged for so southern Italy EARLINET sites, possibly affected by the influence of marine aerosols,
leading to lower LR values for high values of $\beta_a$.
For task 2) we applied the proposed model-based inversion to different ALC systems (Lufft CHM-15k), part of the
Italian ALICENET network. We firstly tested the ability of the proposed approach to derive aerosol extinction by
comparing hourly-mean, vertically-integrated $\alpha_a$ (i.e., hourly mean AOT) derived by three ALC systems to
corresponding AOT measurements from co-located sun-photometers (ALICENET sites of Aosta San Cristophe (ASC),

San Pietro Capofiume (SPC) and Rome Tor Vergata (RTV), Figure 7). ALC-sun photometer agreement was found to be within 30%. Tests on the use of fixed LR were also performed to investigate the advantage of the proposed approach with respect to more standard ones. To this purpose, we used the (1064 nm) fixed-LR value suggested by the E-Profile EUMETSAT Program and the 'weighted mean' derived from our model (52 sr and 38 sr, respectively). While for the ASC site negligible differences were found among the three retrieval schemes, for both SPC and RTV sites the best ALC – sun photometer accordance in AOT is reached when employing the model-based or the fixed LR=38 sr inversion schemes, with a mean error around 16-19 % and 31-33 % for SPC and RTV, respectively. Applying the fixed LR value of 52 sr produces an overestimation of the AOTs, with a mean relative bias equal to 33 % and 44 % at SPC and RTV, respectively. This suggests that, at 1064 nm, the LR value for continental aerosol is lower than the one assumed by the E-Profile procedure and, more in general, this highlights the advantage of a procedure not requiring an a-priori, and to some extent arbitrary, choice of the LR value.

As a second test in task 2, values of aerosol volume (and mass) derived using the model-assisted ALC retrieval were compared to in situ aerosol measurements performed by OPCs and PM10 analyzers. A continuous, two-months comparison (June – July 2012) between daily average aerosol mass concentration as measured in situ and derived by ALC (in the lowest altitudes) at SPC, showed a mean relative difference of around 15% (Figure 9).

Overall, the good results obtained in our validation efforts are encouraging but necessarily related to the specific conditions at the measuring sites considered and to the characteristics of the instruments employed. They are therefore not necessarily representative of results obtainable in all European continental sites, and at all times. Further tests using wider datasets covering a variety of sites and ALC instrumentation would be desirable to better understand potential and limits of the applicability of the proposed method over the larger scale. An obvious intrinsic limitation is that the method is dependent on the considered aerosol type which in this study was tuned to reproduce average continental aerosol conditions. Errors associated to the application of the derived functional relationship might be larger if more 'specific' aerosol conditions (e.g. contamination by sea salt or desert dust particles) affect a given site. In the future, the information coming from ALC systems with an additional depolarization channel (as tested in the DIAPASON Project, Gobbi et al., 2018) could be used to force the retrieval to different model schemes (e.g. switching from 'no dust' to 'dust' schemes conditions) in the same vertical profile. This will enhance the capabilities of ALCs to operatively estimate and characterize the aerosol optical properties (e.g. Gasteiger and Freudenthaler, 2014).

Additionally, although our validation exercises returned results well within the uncertainties related to the model statistical variability alone (i.e., the relative errors associated to the mean functional relationships), the expected total uncertainty to be associated to the method should include terms that have not been specifically addressed in this work, as for example the instrumental error itself.

On the other hand, the proposed approach has the main advantage of allowing the operational (i.e. 24/7) retrieval of fairly reliable, remote sensing profiles of aerosol optical ($\beta_a$, $\alpha_a$) and physical ($S_a$, $V_a$) properties (with associated uncertainties and limitations) by means of relatively simple and robust instruments. This could temporally and spatially complement the information coming from more advanced lidar networks (for example, the Raman channel of multi-wavelength system cannot be used in daylight conditions) and, more in general, could represent a valid option to deliver, in quasi real time, the 3D aerosol fields useful for operational air quality (e.g. integration of the in situ surface measurements) and for meteorological and climate monitoring (e.g. aerosol-cloud interaction and aerosol transport and dispersion processes).

**Acknowledgements**

This work was partly supported by the European Commission Life+ Project "DIAPASON" (LIFE+2010 ENV/IT/391). The study also contributes to the activities of the EU COST Action TOPROF (ES1303). The authors acknowledge EARLINET for providing aerosol lidar profiles through its website (http://access.earlinet.org/) and the EARLINET publishing group 2000-2010 (https://doi.org/10.1594/WDCC/EN_all_measurements_2000-2010). We thank the EARLINET PIs Ulla Wandinger (Leibniz Institute for Tropospheric Research, Leipzig, Germany), Manuel Pujadas (Centro de Investigaciones Energéticas, Medioambientales y Tecnológicas, Department of Environment, Madrid, Spain), Maria Rita Perrone (Department of Mathematics and Physics, Universita' del Salento, Italy) and Aldo Amodeo (Istituto di Metodologie per l'Analisi Ambientale, CNR-IMAA, Italy) and their staff for establishing, maintaining and running the EARLINET instruments at Leipzig (LE), Madrid (MA), Lecce (LE) and Potenza (PO), respectively. ALC measurements at San Pietro Capofiume (SPC) were partly funded by the SuperSito project of the Italian Emilia-Romagna region (DRG no. 428/10). The authors also thank Angelo Lupi, Mauro Mazzola, and Vito Vitale (ISAC-CNR) for the management of the PREDE POM-02L Sun-sky radiometer measurements at SPC. AOT data analysis for San Pietro Capofiume and Aosta San Christophe was performed as part of a cooperative activity with the SKYNET network. We also acknowledge the AERONET team for the processing of the Rome-Tor Vergata data used in this research effort.

**Data availability**

AERONET Rome-Tor Vergata sun photometer AOT data were downloaded from the AERONET web page (AERONET, 2018). SKYNET sun photometer AOT data were downloaded from the SKYNET webpage (SKYNET, 2018). EARLINET backscattering and extinction coefficients were downloaded from the EARLINET webpage (EARLINET, 2018). ALICENET ALC raw data are available upon request at alicenet@isac.cnr.it.

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

Methods and recommendations addressing the relevant European Commission Guidelines in support to the Air Quality
Directive 2008/50/EC, Atmospheric Environment, 161, 288-305, doi: 10.1016/j.atmosenv.2017.04.038, 2017
Barreto, Á., Cuevas, E., Granados-Muñoz, M.-J., Alados-Arboledas, L., Romero, P. M., Gröbner, J., Kouremeti, N.,
Almansa, A. F., Stone, T., Toledano, C., Román, R., Sorokin, M., Holben, B., Canini, M., and Yela, M.: The new sun-
sky-lunar Cimel CE318-T multiband photometer – a comprehensive performance evaluation, Atmos. Meas. Tech., 9,
631-654, https://doi.org/10.5194/amt-9-631-2016, 2016.
Bohren, C. F., and Huffman, D. R.: Absorption and Scattering of Light by Small Particles, Wiley-Interscience, New
York, pp 103, 119-122, 477, 1983.
Bösenberg, J., Ansmann, A., Baldasano, J. M., Balis, D., Böckmann, C., Calpini, B., Chaikovsky, A., Flamant, P.,
Hagard, A., Mitev, V., Papayannis, A., Pelon, J., Resendes, D., Schneider, J., Spinelli, N., Trickl, T., Vaughan, G.,
Visconti, G., and Wiegner, M.: EARLINET: a European Aerosol Research Lidar Network, in: Advances in Laser
Remote Sensing, edited by: Dabas, A., Loth, C., and Pelon, J., Ecole polytechnique, Palaiseau Cedex, France, 155–158,
723 2001.

Bösenberg, J., Matthias, V., Amodeo, A., Amoiridis, V., Ansmann, A., Baldasano, J. M., Balin, I., Balis, D.,
Böckmann, C., Boselli, A., Carlsson, G., Chaikovsky, A., Chourdakis, G., Comerón, A., De Tomasi, F., Eixmann, R.,
Freudenthaler, V., Giehl, H., Grigorov, I., Hågård, A., Iarlori, M., Kirsche, A., Kolarov, G., Komguem, L., Kreipl, S.,
Kumpf, W., Larchevêque, G., Linné, H., Matthey, R., Mattis, I., Mekler, A., Mironova, I., Mitev, V., Mona, L., Müller,
D., Music, S., Nickovic, S., Pandolfi, M., Papayannis, A., Pappalardo, G., Pelon, J., Pérez, C., Perrone, R. M., Persson,
R., Resendes, D. P., Rizi, V., Rocadenbosch, F., Rodrigues, A., Sauvage, L., Schneidenbach, L., Schumacher, R.,
Shcherbakov, V., Simeonov, V., Sobolewski, P., Spinelli, N., Stachlewska, I., Stoyanov, D., Trickl, T., Tsaknakis, G.,
Vaughan, G., Wandinger, U., Wang, X., Wiegner, M., Zavrtanik, M., and Zerefos. C.: EARLINET: A European
Aerosol Research Lidar Network to Establish an Aerosol Climatology, Max-Planck-Institut Report No. 348, 2003.
Boucher, O., Randall, D., Artaxo, P., Bretherton, C., Feingold, G., Forster, P., Kerminen, V.-M., Kondo, Y., Liao, H.,
Lohmann, U., Rasch, P., Satheesh, S. K., Sherwood, S., Stevens, B., and Zhang, X.Y.: Clouds and Aerosols. In: Climate
Change 2013: The Physical Science Basis. Contribution of Working Group I to the Fifth Assessment Report of the
Intergovernmental Panel on Climate Change [Stocker, T.F., D. Qin, G.-K. Plattner, M. Tignor, S.K. Allen, J. Boschung,
A. Nauels, Y. Xia, V. Bex and P.M. Midgley (eds.)]. Cambridge University Press, Cambridge, United Kingdom and
New York, NY, USA, pp. 571–658, doi:10.1017/CBO9781107415324.016, 2013.
Bucci, S., Cristofanelli, P., Decesari, S., Marinoni, A., Sandrini, S., Größ, J., Wiedensohler, A., Di Marco, C. F.,
Nemitz, E., Cairo, F., Di Liberto, L., and Fierli, F.: Vertical distribution of aerosol optical properties in the Po Valley
during the 2012 summer campaigns, Atmos. Chem. Phys., 18, 5371-5389, https://doi.org/10.5194/acp-18-5371-2018,
742 2018.

Cazorla, A., Casquero-Vera, J. A., Román, R., Guerrero-Rascado, J. L., Toledano, C., Cachorro, V. E., Orza, J. A. G.,
Cancillo, M. L., Serrano, A., Titos, G., Pandolfi, M., Alastuey, A., Hanrieder, N., and Alados-Arboledas, L.: Near-real-
time processing of a ceilometer network assisted with sun-photometer data: monitoring a dust outbreak over the Iberian
Peninsula, Atmos. Chem. Phys., 17, 11861-11876, https://doi.org/10.5194/acp-17-11861-2017, 2017
d'Almeida, G. A., Koepke, P., and Shettle, E. P.: Atmospheric Aerosol-Global Climatology and Radiative
Characteristics, A. Deepack Hampton, VA. 561 pp., 1991.
D'Amato, G., Baena-Cagnani, C. E., Cecchi, L., Annesi-Maesano, I., Nunes, C., Ansotegui, I., D'Amato, M., Licciardi,
G., Sofia, M., and Canonica, W. G.: Climate change, air pollution and extreme events leading to increasing prevalence
of allergic respiratory diseases, Multidisciplinary Respiratory Medicine, 8(1), 12. http://doi.org/10.1186/2049-6958-8-
752 12, 2013.

De Tomasi, F., Tafuro, A. M., and Perrone, M. R.: Height and seasonal dependence of aerosol optical properties over
south-east Italy, J. Geophys. Res., 111, D10203, doi:10.1029/2005JD006779, 2006.
Diémoz, H., Barnaba, F., Magri, T., Pession, G., Dionisi, D., Pittavino, S., Tombolato, I. K. F., M., Campanelli, M.,
Della Ceca, L., Hervo, M., Di Liberto, L., Ferrero, L., and Gobbi, G. P.: Transport of Po Valley aerosol pollution to the
northwestern Alps. Part 1: phenomenology, Atmos. Chem. Phys., in preparation, 2018a.
Diémoz, H., Barnaba, F., Magri, T., Pession, G., Pittavino, S., Tombolato, I. K. F., M., Campanelli, M. and Gobbi, G.
P.: Transport of Po Valley aerosol pollution to the northwestern Alps. Part 2: long-term impact on air quality Atmos.
Chem. Phys., in preparation, 2018b.
Dinar, E., Abo Riziq, A., Spindler, C., Erlick, C., Kiss, G., and Rudich, Y.: The complex refractive index of
atmospheric and model humic   like substances (HULIS) retrieved by a cavity ring down aerosol spectrometer (CRD-
AS), Faraday Discuss., 137, 279–295, 2008.
Düsing, S., Wehner, B., Seifert, P., Ansmann, A., Baars, H., Ditas, F., Henning, S., Ma, N., Poulain, L., Siebert, H.,
Wiedensohler, A., and Macke, A.: Helicopter-borne observations of the continental background aerosol in combination
with remote sensing and ground-based measurements, Atmos. Chem. Phys., 18, 1263-1290, https://doi.org/10.5194/acp-
767 18-1263-2018, 2018.

EARLINET: EARLINET data base, available at: http://www.earlinet.org, last access: 11 January 2018.
EC, 2008. Directive 2008/50/EC of the European Parliament and of the Council of 21 May 2008 on Ambient Air
Quality and Cleaner Air for Europe. OJ L 152, 11.6.2008, pp. 1e44. http://eur-
lex.europa.eu/LexUriServ/LexUriServ.do?uri¼OJ: L:2008:152:0001:0044:EN:PDF, 2008.
Feingold, G., McComiskey, A., Yamaguchi, T., Johnson, J. S., Carslaw, K. S., and Schmidt, K. S.: New approaches to
quantifying aerosol influence on the cloud radiative effect. *Proceedings of the National Academy of Sciences of the*
*United States of America*, *113*(21), 5812–5819, 2016.
Fernald, F. G., Herman, B. M., and Reagan J. A.: Determination of Aerosol Height Distributions by Lidar, J. Appl.
Meteor., 11, 482–489, doi:10.1175/1520-0450(1972)011h0482:DOAHDBi2.0.CO;2, 1972.
Flentje, H., Claude, H., Elste, T., Gilge, S., Köhler, U., PlassDülmer, C., Steinbrecht, W., Thomas, W., Werner, A., and
Fricke, W.: The Eyjafjallajökull eruption in April 2010 – detection of volcanic plume using in-situ measurements,
ozone sondes and lidar-ceilometer profiles, Atmos. Chem. Phys., 10, 10085–10092, doi:10.5194/acp-10-10085-2010,
780 2010.

Gasteiger, J., Groß, S., Freudenthaler, V., and Wiegner, M.: Volcanic ash from Iceland over Munich: mass
concentration retrieved from ground-based remote sensing measurements, Atmos. Chem. Phys., 11, 2209–2223,
doi:10.5194/acp-11-2209- 2011, 2011.
Gasteiger, J. and Freudenthaler, V.: Benefit of depolarization ratio at λ = 1064 nm for the retrieval of the aerosol
microphysics from lidar measurements, Atmos. Meas. Tech., 7, 3773-3781, https://doi.org/10.5194/amt-7-3773-2014,
786 2014.

Gobbi, G. P.: Lidar estimation of stratospheric aerosol properties: Surface, volume, and extinction to backscatter ratio.
J. Geophys. Res., 100, 11 219–11 235, 1995.
Gobbi, G. P., Barnaba, F. Di Liberto, L., Bolignano, A., Lucarelli, F., Nava, S., Perrino, C., Pietrodangelo, A., Basar, S.,
Costabile, F., Dionisi, D., Rizza, U., Canepari, S., Sozzi, R., Morelli, M., Manigrasso, M., Drewnick, R. F.,
Struckmeier, C., Poenitz, K., and Wille, H.: An integrated view of Saharan Dust Advections to Italy and the Central
Mediterranean: Main Outcomes of the 'DIAPASON' Project, submitted to Atmospheric Environment, 2018.
Haarig, M., Engelmann, R., Ansmann, A., Veselovskii, I., Whiteman, D. N., and Althausen, D.: 1064 nm rotational
Raman lidar for particle extinction and lidar-ratio profiling: cirrus case study, Atmos. Meas. Tech., 9, 4269–4278,
https://doi.org/10.5194/amt9-4269-2016, 2016.
Hand, J. L., and Malm,W. C.: Review of aerosol mass scattering efficiencies from ground-based measurements since
1990, J. Geophys. Res.,112, D16203, doi:10.1029/2007JD008484, 2007.
Hanel, G.: The properties of atmospheric aerosol particles as function of the relative humidity at thermodynamic
equilibrium with the surrounding moist air, Adv. Geophys., 19, 73–188, 1976
Haywood, J. M., and Boucher, O.: Estimates of the direct and indirect radiative forcing due to tropospheric aerosols: A
review, Rev. Geophys., 38, 513–543, 2000.
He, T.-Y., Stanič, S., Gao, F., Bergant, K., Veberič, D., Song, X.-Q., and Dolžan, A.: Tracking of urban aerosols using
combined LIDAR-based remote sensing and ground-based measurements, Atmos. Meas. Tech., 5, 891-900,
https://doi.org/10.5194/amt-5-891-2012, 2012.
Heese, B., and Wiegner, M.: Vertical aerosol profiles from Raman polarization lidar observations during the dry season
AMMA field campaign, J. Geophys. Res.- Atmos., 113, doi:10.1029/2007JD009487, 2008.
Heese, B., Flentje, H., Althausen, D., Ansmann, A., and Frey, S.: Ceilometer lidar comparison: backscatter coefficient
retrieval and signal-to-noise ratio determination, Atmos. Meas. Tech., 3, 1763–1770, doi:10.5194/amt-3-1763-2010,
809 2010.

Hervo, M., Poltera, Y., and Haefele, A.: An empirical method to correct for temperature-dependent variations in the
overlap function of CHM15k ceilometers, Atmos. Meas. Tech., 9, 2947-2959, https://doi.org/10.5194/amt-9-2947-2016,
812 2016.

Hess, M., Koepke, P., and Schult, I.: Optical properties of aerosols and clouds: The software package OPAC. Bull.
Amer. Meteor. Soc., 79, 831–844, 1998.
Holben, B. N., Eck, T. F., Slutsker, I., Tanré, D., Buis, J. P., Setzer, A., Vermote, E.; Reagan, J. A.; Kaufman, Y. J.,
Nakajima, T., Lavenu, F., Jankowiak, I., and Smirnov, A.: AERONET - A federated instrument network and data
archive for aerosol characterization. *Remote Sensing of Environment*, *66*(1), 1-16. DOI: 10.1016/S0034-4257(98)00031-
818 5, 1998.

Klett, J. D.: Stable analytical inversion solution for processing lidar returns, *Appl. Optics*, **20**, 211–220, 1981.
Klett, J. D.: Lidar inversion with variable backscatter/extinction ratios, Appl. Opt., 24, 1638–1643,
doi:10.1364/AO.24.001638, 1985.
Koetz, B.; Sun, G.; Morsdorf, F.; Ranson, K.J.; Kneubuhler, M.; Itten, K.; Allgower, B., "Inversion of Combined
Radiative Transfer Models for Imaging Spectrometer and LIDAR Data," *2006 IEEE* International Symposium on
Geoscience and Remote Sensing, Denver, CO, pp. 395-398. doi: 10.1109/IGARSS.2006.106, 2006.
Kovalev, V. A., and Eichinger , W. E.: Elastic Lidar. Theory, Practice, and Analysis Methods, 615 pp., John Wiley &
Sons, Weinheim, Germany, 2004.
Lelieveld, J.; Evans, J.S.; Fnais, M.; Giannadaki, D.; Pozzer, A., The contribution of outdoor air pollution sources to
premature mortality on a global scale, Nature, 525, 367–371, 2015.
Levy, R. C., Remer, L. A., and Dubovik, O.: Global aerosol optical properties and application to Moderate Resolution
Imaging Spectroradiometer aerosol retrieval over land *J. Geophys. Res., Vol. 112, D13210, doi:*
*10.1029/2006JD007815,* 2007.
Lewandowski, P. A., Eichinger, W. E., Holder, H., Prueger, J., Wang, J., and Kleinman, L. I.: Vertical distribution of
aerosols in the vicinity of Mexico City during MILAGRO-2006 Campaign, Atmos. Chem. Phys., 10, 1017-1030,
https://doi.org/10.5194/acp-10-1017-2010, 2010.
Li, S., Joseph, E., and Min, Q.: Remote sensing of ground-level PM2.5 combining AOD and backscattering profile,
Remote Sens. Environ., 183, 120–128, https://doi.org/10.1016/j.rse.2016.05.025, 2016.
Li, S., Joseph, E., Min, Q., Yin, B., Sakai, R., and Payne, M. K.: Remote sensing of PM2.5 during cloudy and nighttime
periods using ceilometer backscatter, Atmos. Meas. Tech., 10, 2093-2104, https://doi.org/10.5194/amt-10-2093-2017,
839 2017.

Lohmann, U., and Feichter, J.: Global indirect aerosol effects: A review, Atmos. Chem. Phys., 5, 715–737, 2005.
Lopatin, A., Dubovik, O., Chaikovsky, A., Goloub, P., Lapyonok, T., Tanré, D., and Litvinov, P.: Enhancement of
aerosol characterization using synergy of lidar and sun-photometer coincident observations: the GARRLiC algorithm,
Atmos. Meas. Tech., 6, 2065-2088, doi:10.5194/amt-6-2065-2013, 2013.
Lolli, S., Madonna, F., Rosoldi, M., Campbell, J. R., Welton, E. J., Lewis, J. R., Gu, Y., and Pappalardo, G.: Impact of
varying lidar measurement and data processing techniques in evaluating cirrus cloud and aerosol direct radiative effects,
Atmos. Meas. Tech., 11, 1639-1651, https://doi.org/10.5194/amt-11-1639-2018, 2018.
Madonna, F., Amodeo, A., Boselli, A., Cornacchia, C., Cuomo, V., D'Amico, G., Giunta, A., Mona, L., and
Pappalardo, G.: CIAO: the CNR-IMAA advanced observatory for atmospheric research, Atmos. Meas. Tech., 4, 1191–
1208, doi:10.5194/amt4-1191-2011, 2011.
Madonna, F., Amato, F., Vande Hey, J., and Pappalardo, G.: Ceilometer aerosol profiling versus Raman lidar in the
frame of the INTERACT campaign of ACTRIS, Atmos. Meas. Tech., 8, 2207-2223, https://doi.org/10.5194/amt-8-
852 2207-2015, 2015.

Madonna, F., Rosoldi, M., Lolli, S., Amato, F., Vande Hey, J., Dhillon, R., Zheng, Y., Brettle, M., and Pappalardo, G.: Intercomparison of aerosol measurements performed with multi-wavelength Raman lidars, automatic lidars and ceilometers in the framework of INTERACT-II campaign, Atmos. Meas. Tech., 11, 2459-2475, https://doi.org/10.5194/amt-11-2459-2018, 2018.

Mamali, D., Marinou, E., Sciare, J., Pikridas, M., Kokkalis, P., Kottas, M., Binietoglou, I., Tsekeri, A., Keleshis, C., Engelmann, R., Baars, H., Ansmann, A., Amiridis, V., Russchenberg, H., and Biskos, G.: Vertical profiles of aerosol mass concentrations observed during dust events by unmanned airborne in-situ and remote sensing instruments, Atmos. Meas. Tech. Discuss., https://doi.org/10.5194/amt-2017-422, in review, 2018.

Mamouri, R. E. and Ansmann, A.: Fine and coarse dust separation with polarization lidar, Atmospheric Measurement Techniques, 7, 3717– 5 3735, 2014.

Mamouri, R.-E. and Ansmann, A.: Potential of polarization lidar to provide profiles of CCN-and INP-relevant aerosol parameters, Atmospheric Chemistry and Physics, 16, 5905–5931, 2016.

Matthias, V., and Bösenberg, J.: Aerosol climatology for the planetary boundary layer derived from regular lidar measurements, *Atmos. Res.*, **63**, 221–245, 2002.

Menut, L., Flamant, C., Pelon, J., Valentin, R., Flamant, P. H., Dupont, E., and Carissimo, B.: Study of the boundary layer structure over the Paris agglomeration as observed during the ECLAP Experiment, in: Advances in atmospheric remote sensing with lidar, edited by: Ansmann, A., Neuber, R., Rairoux, P., and Wandinger, U., Springer, Berlin, 15– 18, 1997.

McGill, M. J., Yorks, J. E., Scott, V. S., Kupchock, A. W., and Selmer, P. A.: The Cloud-Aerosol Transport System (CATS): A technology demonstration on the International Space Station, in: Lidar Remote Sensing for Environmental Monitoring XV (vol. 9612, p. 96120A). International Society for Optics and Photonics, Proc. SPIE, 9612, 96120A, https://doi.org/10.1117/12.2190841, 2015.

Mishchenko, M.I., Travis, L.D., Kahn, R.A., and West, R.A.: Modeling phase functions for dustlike tropospheric aerosols using a mixture of randomly oriented polydisperse spheroids. *J. Geophys. Res.*, **102**, 16831-16847, doi:10.1029/96JD02110, 1997.

Mona, L., Amodeo, A., Pandolfi, M., and Pappalardo, G.: Saharan dust intrusions in the Mediterranean area: Three years of Raman lidar measurements, J. Geophys. Res., 111, D16203, doi:10.1029/2005JD006569, 2006.

Omar, A. H., Winker, D. M., Kittaka, C., Vaughan, M. A., Liu, Z. Y., Hu, Y. X., Trepte, C. R., Rogers, R. R., Ferrare, R. A., Lee, K. P., Kuehn, R. E., and Hostetler, C. A.: The CALIPSO automated aerosol classification and lidar ratio selection algorithm, J. Atmos. Ocean. Tech., 26, 1994–2014, doi:10.1175/2009jtecha1231.1, 2009.

Papagiannopoulos, N., Mona, L., Alados-Arboledas, L., Amiridis, V., Baars, H., Binietoglou, I., Bortoli, D., D'Amico, G., Giunta, A., Guerrero-Rascado, J. L., Schwarz, A., Pereira, S., Spinelli, N., Wandinger, U., Wang, X., and Pappalardo, G.: CALIPSO climatological products: evaluation and suggestions from EARLINET, Atmos. Chem. Phys., 16, 2341-2357, https://doi.org/10.5194/acp-16-2341-2016, 2016.

Pappalardo, G., Wandinger, U., Mona, L., Hiebsch, A., Mattis, I., Amodeo, A., Ansmann, A., Seifert, P., Linne, H., Apituley, A., Alados Arboledas, L., Balis, D., Chaikovsky, A., D'Amico, G., De Tomasi, F., Freudenthaler, V., Giannakaki, E., Giunta, A., Grigorov, I., Iarlori, M., Madonna, F., Mamouri, R.-E., Nasti, L., Papayannis, A.,

Pietruczuk, A., Pujadas, M., Rizi, V., Rocadenbosch, F., Russo, F., Schnell, F., Spinelli, N., Wang, X., and Wiegner, M.: EARLINET correlative measurements for CALIPSO: first intercomparison results, J. Geophys. Res., 115, D00H19, doi:10.1029/2009JD012147, 2010.

Pappalardo, G., Amodeo, A., Apituley, A., Comeron, A., Freudenthaler, V., Linné, H., Ansmann, A., Bösenberg, J., D'Amico, G., Mattis, I., Mona, L., Wandinger, U., Amiridis, V., Alados-Arboledas, L., Nicolae, D., and Wiegner, M.: EARLINET: towards an advanced sustainable European aerosol lidar network, Atmos. Meas. Tech., 7, 2389-2409, https://doi.org/10.5194/amt-7-2389-2014, 2014.

Patterson, E. M., Kiang, C. S., Delany, A. C., Wartburg, A. F., Leslie, D., and Huebert, B. J: Global measurements of aerosols in remote continental and marine regions: Concentrations, size distributions, and optical properties. J. Geophys. Res., 85, 7361–7375, 1980.

Perrone, M. R., De Tomasi, F., and Gobbi, G. P.: Vertically resolved aerosol properties by multi-wavelength lidar measurements, Atmos. Chem. Phys., 14, 1185-1204, doi:10.5194/acp-14-1185-2014, 2014.

Pletscher, K., Weiss, M., and Moelter, L.: Simultaneous determination of PM fractions, particle number and particle size distribution in high time resolution applying one and the same optical measurement technique, Gefahrst. Reinhalt. L., 76, 425-436, http://www.gefahrstoffe.de/gest/article.php?data[article_id]=86622, 2016.

Putaud, J.P., Van Dingenen, R., Alastuey, A., Bauer, H., Birmili, W., Cyrys, J., Flentje, H., Fuzzi, S., et al.: A European aerosol phenomenology - 3: physical and chemical characteristics of particulate matter from 60 rural, urban, and kerbside sites across Europe, Atmos. Environ., 44, 1308 – 1320, 2010.

R. Román, Benavent-Oltra, J. A., Casquero-Vera, J. A., Lopatin, A., Cazorla, A., Lyamani, H., Denjean, C., Fuertes, D. Pérez-Ramírez, D., Torres, B., Toledano, C., Dubovik, O., Cachorro, V. E., de Frutos, A. M., Olmo, F. J., and Alados-Arboledas, L.: Retrieval of aerosol profiles combining sunphotometer and ceilometer measurements in GRASP code, Atm. Res., Vol. 204, 15 May 2018, 161-177, https://doi.org/10.1016/j.atmosres.2018.01.021, 2018.

Shipley, S. T. Tracy, D. H., Eloranta, E. W., Tauger, J. T., Sroga, J. T., Roesler, F. L., and Weinman J. A.: High spectral resolution lidar to measure optical scattering properties of atmospheric aerosols. 1: Theory and instrumentation, Appl. Opt. 22, 3716–3724, 1983.

Sicard, M., Guerrero-Rascado, J. L., Navas-Guzman, F., Preißler, J., ´Molero, F., Tomas, S., Bravo-Aranda, J. A., Comeron, A., Rocadenbosch, F., Wagner, F., Pujadas, M., and Alados-Arboledas, L.: Monitoring of the Eyjafjallajokull volcanic aerosol plume over ¨ the Iberian Peninsula by means of four EARLINET lidar stations, Atmos. Chem. Phys., 12, 3115–3130, doi:10.5194/acp-12-3115- 2012, 2012.

SKYNET: SKYNET data base, available at: http://www.skynet-isdc.org/index.php, last access: 15 September 2017.

Stevens, B., and Feingold, G.: Untangling aerosol effects on clouds and precipitation in a buffered system. Nature 461(7264):607–613, 2009.

TSI, Model 3330 optical particle sizer spectrometer operation and service manual, P/N 6004403, Revision E, April 2011.

Tosca, M. G., Campbell, J., Garay, M., Lolli, S., Seidel, F. C., Marquis, J., and Kalashnikova, O.: Attributing accelerated summertime warming in the southeast united states to recent reductions in aerosol burden: Indications from vertically-resolved observations, Remote Sens., 9, 674, https://doi.org/10.3390/rs9070674, 2017.

Van Dingenen, R., Raes, F., Putaud, J.-P., Baltensperger, U., Charron, A., Facchini, M. C., Decesari, S., et al.: A
European aerosol phenomenology – 1: physical characteristics of particulate matter at kerbside, urban, rural and
background sites in Europe, Atmos. Environ., 38, 2561 – 2577, 2004.
Veselovskii, I., Dubovik, O., Kolgotin, A., Korenskiy, M., Whiteman, D. N., Allakhverdiev, K., and Huseyinoglu, F.:
Linear estimation of particle bulk parameters from multi-wavelength lidar measurements, Atmos. Meas. Tech., 5, 1135-
1145, https://doi.org/10.5194/amt-5-1135-2012, 2012.
Wagner, R., Ajtai, T., Kandler, K., Lieke, K., Linke, C., Müller, T., Schnaiter, M., and Vragel, M.: Complex refractive
indices of Saharan dust samples at visible and near UV wavelengths: a laboratory study, Atmos. Chem. Phys., 12, 2491-
2512, doi:10.5194/acp-12-2491-2012, 2012.
Whitby, K. T.: Physical Characteristics of Sulfur Aerosols. Atmos. Environ. 12:135-159, 1978.
Wiegner, M. and Geiß, A.: Aerosol profiling with the Jenoptik ALC CHM15kx, Atmos. Meas. Tech., 5, 1953–1964,
doi:10.5194/amt-5-1953-2012, 2012.
Wiegner, M., Madonna, F., Binietoglou, I., Forkel, R., Gasteiger, J., Geiß, A., Pappalardo, G., Schäfer, K., and Thomas,
W.: What is the benefit of ceilometers for aerosol remote sensing? An answer from EARLINET, Atmos. Meas. Tech.,
7, 1979-1997, https://doi.org/10.5194/amt-7-1979-2014, 2014.
Winker, D. M., Pelon, J. R., and McCormick, M. P.: The CALIPSO mission: Spaceborne lidar for observation of
aerosol and clouds, SPIE Proc. Ser., vol. 4893, Soc. Photo-Opt. Instrum. Eng., Bellingham, Washington, 2003.
World Health Organization, Review of Evidence on Health Aspects of Air Pollution REVIHAAP Project, Technical
Report, World Health Organization: Copenhagen, Denmark, 2013.
Yorks, J. E., McGill, M. J., Palm, S. P., Hlavka, D. L., Selmer, P. A., Nowottnick, E. P., Vaughan, M. A., Rodier, S. D.,
and Hart, W. D.: An overview of the CATS level 1 processing algorithms and data products, Geophys. Res. Lett., 43,
4632–4639, https://doi.org/10.1002/2016GL068006, 2016.

**Table 1. Aerosol parameter values as reported in literature for continental-type aerosols.**

| Reference | $r_1$ ($\mu m$) $\sigma_1$ | $r_2$ ($\mu m$) $\sigma_2$ | $r_3$ ($\mu m$) $\sigma_3$ | $N_1/N_{tot}$ (%) | $N_2/N_{tot}$ (%) | $N_3/N_{tot}$ (%) | $m_{r\_1}$, $m_{im\_1}$ | $m_{r\_2}$ $m_{im\_2}$ | $m_{r\_3}$ $m_{im\_3}$ | $N_{tot}$ ($cm^{-3}$) | Aerosol type |
|---|---|---|---|---|---|---|---|---|---|---|---|
| *Whitby (1978)[1]* | 0.008 1.6 | 0.034 2.1 | 0.46 2.2 | 0.56 | 0.44 | $4 \times 10^{-4}$ | - | - | - | 1800 | Clean continental |
| *D'Almeida et al. (1991)[2]* | 0.012 2.0 | 0.029 2.24 | 0.471 2.51 | 0.06 | 0.94 | $2 \times 10^{-6}$ | 1.75 0.44 | 1.53 0.012 | 1.53 0.008 | 20000 | Average continental |
| *Hess et al. (1998)[2]* | 0.012 2.0 | 0.021 2.24 | 0.471 2.51 | 0.56 | 0.44 | $0.3 \times 10^{-4}$ | 1.75 0.44 | 1.53 0.012 | 1.53 0.008 | 15300 | Average continental |
| *Barnaba and Gobbi (2004a)[1]* | 0.007-0.012 1.7 – 2.0 | 0.021 – 0.077 2.03 – 2.24 | 0.403 – 0.5 2.11 – 2.24 | 6.1– 54.2 | 45.8 – 93.9 | $(2 –26.1) \times 10^{-4}$ | 1.25–2.00 0.07–1.00 | 1.53 $6 \times 10^{-3}$ | 1.53 $8 \times 10^{-3}$ | $10^3 - 10^4$ | |
| *Omar et al. (2009)[1]* | - | 0.093-0.10 1.53-1.61 | 0.68-0.76 1.9-2.1 | - | 0.999-1 | $(0.02-3) \times 10^{-4}$ | - | 1.38-1.40 $(0.1 –6.3) \times 10^{-3}$ | 1.40-1.46 $(3.4-6.3) \times 10^{-3}$ | - | Clean and polluted continental |
| *Levy et al. (2007)[2]* | 0.018 2.0 | 0.005 2.97 | 0.5 2.97 | 1 | $1 \times 10^{-7}$ | $1 \times 10^{-13}$ | 1.75 0.44 | 1.53 $6 \times 10^{-3}$ | 1.53 $8 \times 10^{-3}$ | - | |
| *Barnaba et al. (2007)[1]* | - | 0.05-0.1 1.35-1.70 | 0.4-0.5 1.5-2.0 | - | 0.98-0.99 | 0.01-0.02 | - | 1.35-1.55 $(2.5 –20) \times 10^{-3}$ | 1.53-1.6 $(1.0 –80) \times 10^{-4}$ | $1-3 \times 10^3$ | Continental - coastal |
| *Amiridis et al. (2015)[1]* | - | 0.03-0.9 1.6-2.2 | 0.47-0.69 1.9-2.5 | - | 1 | $(4 – 8) \times 10^{-7}$ | - | 1.42-1.45 $(2.3 –6) \times 10^{-3}$ | 1.45-1.53 $(2.3 –6) \times 10^{-3}$ | - | Clean and polluted continental |

[1]The refractive index is at λ=532 nm.
[2]The refractive index is at λ=550 nm.

 **Table 2. Variability ranges used in this study. Values refer to ground and dry conditions (see text for details).**

| Parameter | Mode I | Mode II | Mode III |
|---|---|---|---|
| $r_i$ (μm) | 0.005 - 0.03 | 0.03 - 0.1 | 0.3 - 0.5 |
| $\sigma$ | 1.35 – 1.7 | 1.35 – 1.7 | 1.5 – 2.4 |
| $N_i/N_{tot}$ (%) | 10 - 60 | 40 - 90 | 0.01 – 0.5 |
| $m_{r\_i}$ (355 nm) | 1.40 – 1.80 | 1.40 – 1.70 | 1.50 – 1.60 |
| (532 nm) | 1.40 – 1.80 | 1.40 – 1.70 | 1.50 – 1.60 |
| (1064 nm) | 1.42 – 1.82 | 1.37 – 1.66 | 1.50 – 1.60 |
| $m_{im\_i}$ (355 nm) | $1\times10^{-2} - 0.47$ | $1\times10^{-4} - 0.010$ | $1\times10^{-4} - 0.02$ |
| (532 nm) | $9\times10^{-3} - 0.44$ | $1.2\times10^{-4} - 0.012$ | $1\times10^{-4} - 0.01$ |
| (1064 nm) | $9\times10^{-3} - 0.44$ | $1.5\times10^{-4} - 0.015$ | $1\times10^{-4} - 0.005$ |
| $N_{tot}$ (cm$^{-3}$) | 500 – 10000 | | |

**Table 3. Parameters of the Seventh-Order Polynomial Fits ($y=a_0+a_1x+a_2x^2+a_3x^3+a_4x^4+a_5x^5+a_6x^6+a_7x^7$) for $\lambda$ = 1064 nm, with**
$x=\log(\beta_a)$ **(in km$^{-1}$ sr$^{-1}$) and $y=\log(\alpha_a, S_a,$ or $V_a)$ in (km$^{-1}$, cm$^2$/cm$^3$ and cm$^3$/cm$^3$, respectively).**

| Functional relantionship at 1064 nm | Extinction coefficient | Surface area | Volume |
|---|---|---|---|
| $a_0$ | 3.797837507651898 | 12.019452592845141 | -5.314834128998254 |
| $a_1$ | 3.294032541389781 | 30.825966279368547 | 2.500484347793244 |
| $a_2$ | 0.962603336867675 | 24.518531616019207 | -1.196109537503000 |
| $a_3$ | 0.241796629870675 | 10.625241994796593 | -1.583236058579546 |
| $a_4$ | 0.064609145804688 | 2.634051072085453 | -0.681801883947768 |
| $a_5$ | 0.017721752150233 | 0.373150843707711 | -0.145232662646142 |
| $a_6$ | 0.002722551625862 | 0.027971628176431 | -0.015471229968392 |
| $a_7$ | 0.000157245409783 | 0.000854381337164 | -0.000658925756875 |


**Table 4. Mean weighted LR at 355, 532 and 532 nm derived in this work and comparison to the corresponding aerosol**
**subtypes (clean continental, CC, and polluted continental, PC) from relevant literature.**

| LR (sr) | λ =355 nm | λ =532 nm | λ =1064 nm |
|---|---|---|---|
| *Omar et al., (2009) (Calipso aerosol model)* | - | 70 ± 25 (PC) | 30 (PC) |
| | - | 35 ± 16 (CC) | 30 (CC) |
| *Amiridis et al. (2015) (LIVAS database)* | 59.5* (PC) | 64 (PC) | - |
| | 56.5* (CC) | 54 (CC) | - |
| *Papagiannopoulos et al.. (2016) (EARLINET measurements)* | - | 62 ±10 (PC) | - |
| | - | 47 ± 4 (CC) | - |
| *Düsing et al.. (2018) (in-situ and lidar measurements)* | 55 | 55 | 30; 15** |
| *This work* | 50.1 ± 17.9 | 49.6 ± 16.0 | 37.7 ± 12.6 |

* derived using the extinction-related and backscatter-related Ångström exponents given by Amiridis et al. (2013)
** see the explanation in the text for the two different values

**Table 5. Extinction-to-volume conversion factors, $c_v = V_a/\alpha_a$ (and corresponding 'mass-to-extinctions efficiency' values, MEE**
**$= \alpha_a/(V_a \ast \rho_a)$, given assuming $\rho_a = 2$ g/cm$^3$) of continental particles as derived from our model at different wavelengths**

| Reference | $c_v$ [$10^{-6}$m] (corresponding MEE, [$m^2 g^{-1}$]) | | | Notes |
|---|---|---|---|---|
| Wavelength [nm] | 355 | 532 | 1064 | |
| Hess et al. (1998) | - | 0.35 (1.43) | - | OPAC, clean continental model |
| Hess et al. (1998) | - | 0.28 (1.79) | - | Opac, polluted continental model |
| Barnaba and Gobbi (2004b) | - | 0.18 (2.78) | - | Continental model |
| Ansmann et al. (2011b) | - | 0.18 (2.78) | - | Germany, fine aerosol fraction |
| Lewandosky et al. (2010) | - | - | 0.77 – 2 (0.25-0.65) | Mexico city basin |
| Sicard et al. (2012) | - | 0.26 (1.92) | - | AERONET, Spain |
| Mamouri and Ansmann (2017) | 0.17 (2.94) | 0.30 (1.67) | 0.96 (0.52) | Germany, continental anthropogenic pollution |
| Mamouri and Ansmann (2017) | 0.23 (2.17) | 0.41 (1.22) | 1.41 (0.35) | Cyprus, continental anthropogenic pollution |
| Mamali et al. (2018) | | 0.14, 0.24 (3.57, 2.03) | | Cyprus, fine non-dust aerosol fraction |
| This work | 0.12 (4.17) | 0.19 (2.63) | 0.60 (0.83) | Continental (clean to moderately polluted) |


**Table 6. Main characteristics of the dataset of the EARLINET continental sites considered in this study. The listed dataset**
**refers to the data downloaded from the EARLINET site (last access on the 11[th] of January 2018).**

| Station | Number of points at 355 and at 532 nm) | Altitude range (Δz, in km) | Period |
|---|---|---|---|
| Lecce (LC) 40.33 N, 18.10 E, 30 m a.s.l. | 1012 – 109 | 1 – 4 | Aug2007 – Oct2013 |
| Leipzig (LE) 51.35 N, 12.43 E, 90 m a.s.l., | 5186 – 4549 | 1.5 – 4 | Aug2008 – Sept2016 |
| Potenza (PO) 40.6 N, 15.72 E, 760 m a.s.l. | 1244 – 219 | 1.5 – 4 | May2000 – Aug2009 |
| Hamburg (HH) 53.57 N, 9.97 E, 25 m a.s.l. | 243 – n.a. | 0.5 – 4 | Apr2001 – Oct2002 |
| Madrid (MA) 40.45 N, 3.73 E, 669 m a.s.l. | n.a. – 492 | 0.5 – 4 | Jun2006 – Jun2008 |


**Table 7 Mean LR discrepancies between our model results and EARLINET measurements and weighted LR at 355 and 532**
**nm for the considered EARLINET stations.**

| Station | $[(LR_{mod} - LR_{meas})/LR_{meas}]*100$ | | EARLINET weighted LR (sr) | |
|---|---|---|---|---|
| | $\lambda = 355\ nm$ | $\lambda = 532\ nm$ | $\lambda = 355\ nm$ | $\lambda = 532\ nm$ |
| LC | 10 | 18 | 51.8 | 44.5 |
| LE | 6 | 9 | 52.6 | 51.0 |
| PO | 17 | 7 | 44.9 | 57.2 |
| HH | 5 | - | 53.3 | - |
| MA | - | 6 | - | 54.2 |


**Table 8. Main characteristics of the ALC and co-located sun-sky radiometer equipment located at the considered**
**ALICENET sites.**

| | Site type | ALC model | ALC firmware | Sun photometer model |
|---|---|---|---|---|
| *ASC* | alpine | Nimbus CHM150104 | 0.743 | POM-02 |
| *SPC* | rural | Nimbus CHM110115 | 0.556 | POM-02L |
| *RTV* | semi-rural | Nimbus CHM070052 | 0.720 | CIMEL CE-318 |


**Table 9. Results of the comparison between the AOT measured by sun-photometers and the one derived by ALCs (model-**
**based and fixed LR inversion schemes) at three ALICENET stations. Mean differences (expressed in terms of <dAOT> =**
**<(AOT$_{ceil}$-AOT$_{phot}$)>, <|dAOT|> (module), <dAOT/AOT> and <|dAOTI/AOT|>) are reported with their standard deviations.**

| *ALICENET sites* | <dAOT> | <|dAOT|> | <dAOT/AOT> | <|dAOTI/AOT|> |
|---|---|---|---|---|
| **ASC** | | | | |
| *Variable LR from our model* | -0.004 ± 0.015 | 0.010 ± 0.013 | -0.25 ± 0.57 | 0.31 ± 0.35 |
| | 0.002 ± 0.021 | 0.009 ± 0.015 | 0.31 ± 0.58 | 0.33 ± 0.35 |
| *LR = 52 sr* | | | | |
| *LR = 38 sr* | -0.004 ± 0.014 | 0.009 ± 0.012 | -0.23 ± 0.43 | 0.30 ± 0.32 |
| **SPC** | | | | |
| *Variable LR from our model* | -0.001 ± 0.020 | 0.013 ± 0.016 | -0.005 ± 0.28 | 0.19 ± 0.20 |
| | 0.021 ± 0.026 | 0.026 ± 0.02 | 0.33 ± 0.35 | 0.38 ± 0.26 |
| *LR= 52 sr* | | | | |
| *LR= 38 sr* | -0.003 ± 0.019 | 0.011 ± 0.014 | -0.043 ± 0.24 | 0.16 ± 0.18 |
| **RTV** | | | | |
| *Variable LR from our model* | 0.004 ± 0.020 | 0.014 ± 0.014 | 0.11 ± 0.49 | 0.33 ± 0.30 |
| | 0.016 ± 0.023 | 0.021 ± 0.018 | 0.44 ± 0.59 | 0.49 ± 0.45 |
| *LR= 52 sr* | | | | |
| *LR= 38 sr* | 0.003 ± 0.019 | 0.013 ± 0.013 | 0.088 ± 0.460 | 0.31 ± 0.27 |



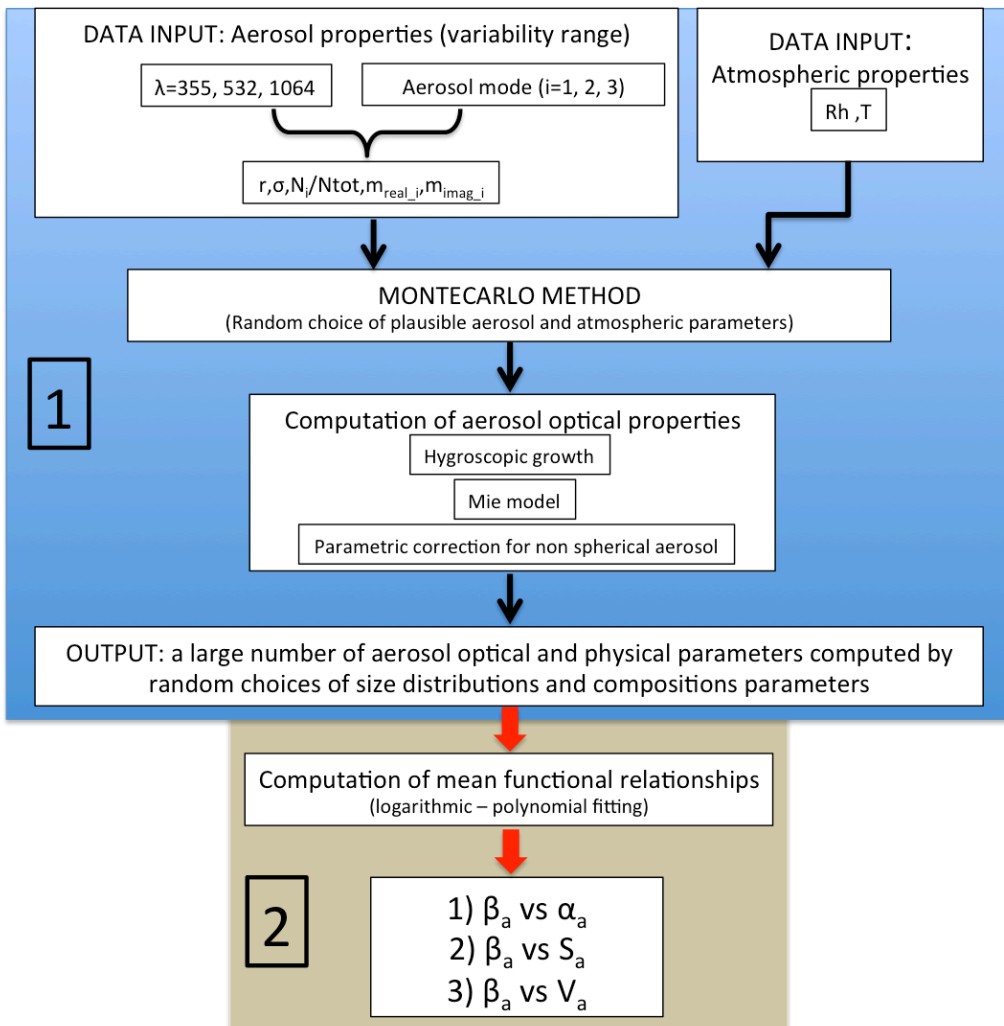


**Figure 1.** Schematic of the two-step model structure developed to obtain, as a result, functional relationships between the
aerosol backscatter ($\beta_a$) and the aerosol extinction, surface area and volume ($\alpha_a$, $S_a$ and $V_a$, respectively).


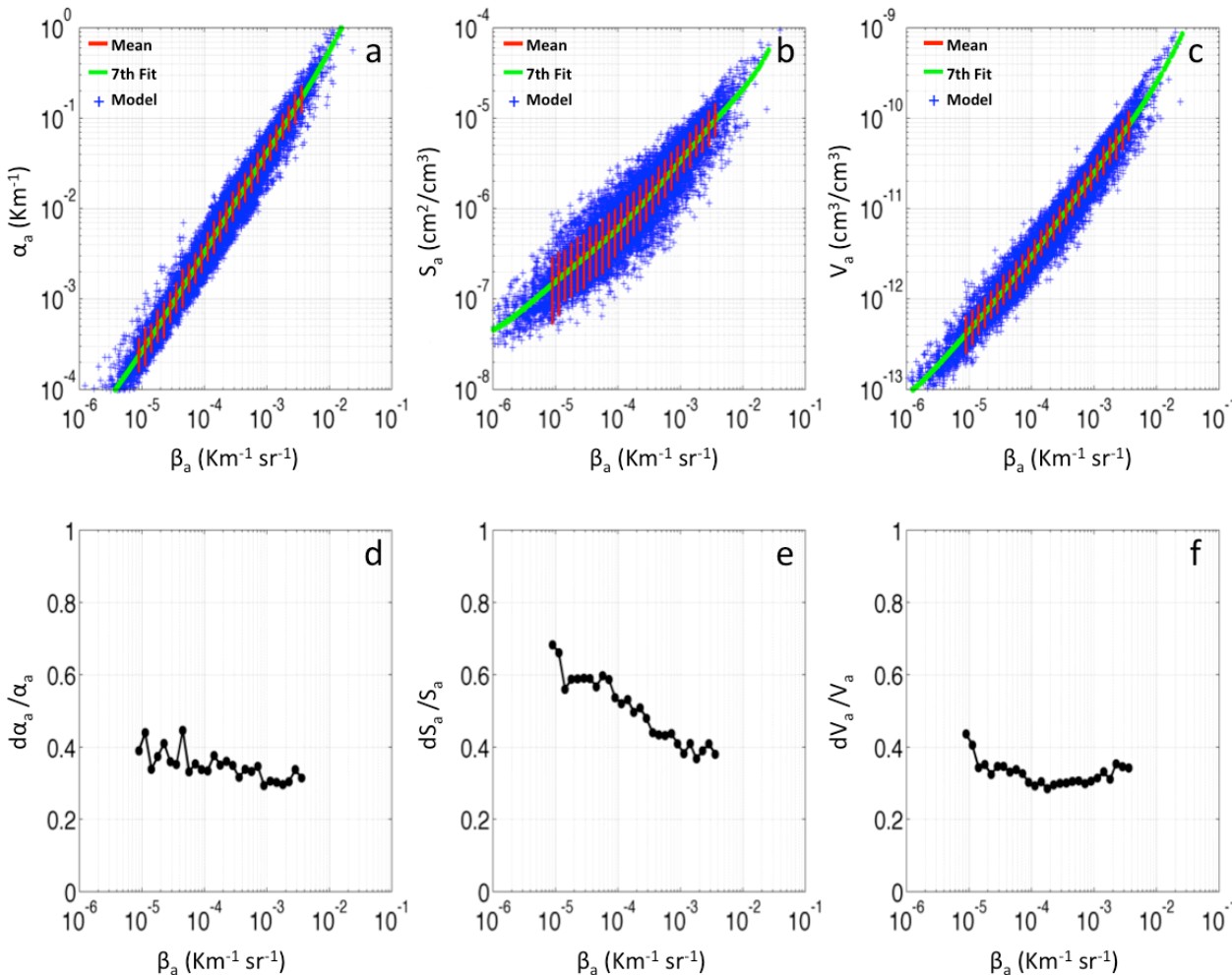


Figure 2. Scatterplots of a) $\alpha_a$ (km$^{-1}$), b) $S_a$ (cm$^2$/cm$^3$) and c) $V_a$ (cm$^3$/cm$^3$) vs backscatter $\beta_a$ (km$^{-1}$ sr$^{-1}$) and relevant relative errors (panels d, e, f, respectively) as derived from 20000 model computations (blue points) at $\lambda$ = 1064 nm. Red dots and error bars are the average values per decade of $\beta$ and their standard deviations, green lines are the 7th-order polynomial fit curve of the 20000 points.

999

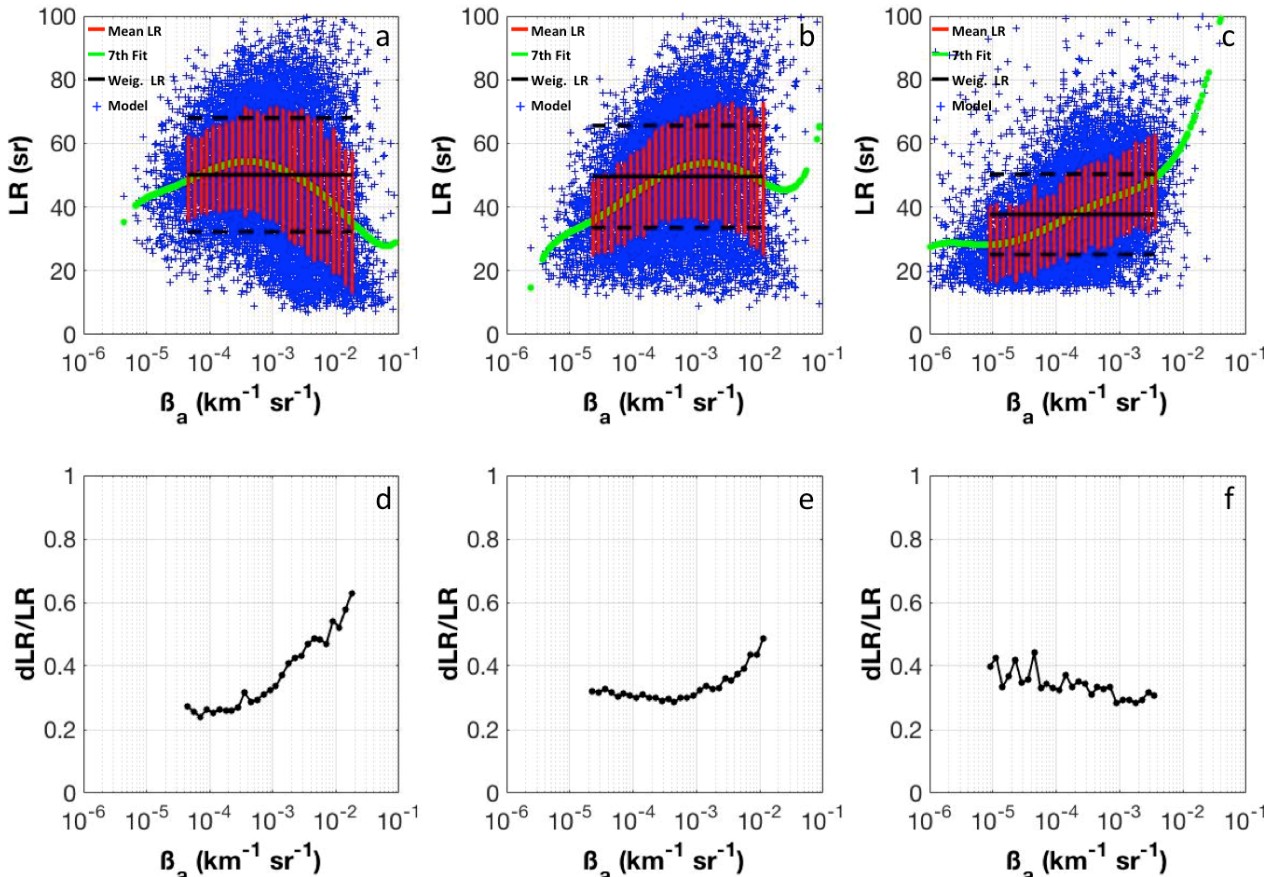

1000

Figure 3. Upper plots: scatterplots of LR (sr) versus $\beta_a$ (km$^{-1}$ sr$^{-1}$) at: a) 355 nm; b) 532 nm; c) 1064 nm (blue points). The 7th-order polynomial fit curve (green lines) and the average values per decade of ß together with their standard deviations (red points and red vertical bars, respectively) are also reported. Horizontal black lines are mean values of the 'weighted-LR' and ± 1 s. d. (solid and dotted lines, respectively). Lower plots: relative errors associated with the model-derived LR at d) 355 nm; e) 532 nm; f) 1064 nm.

1006

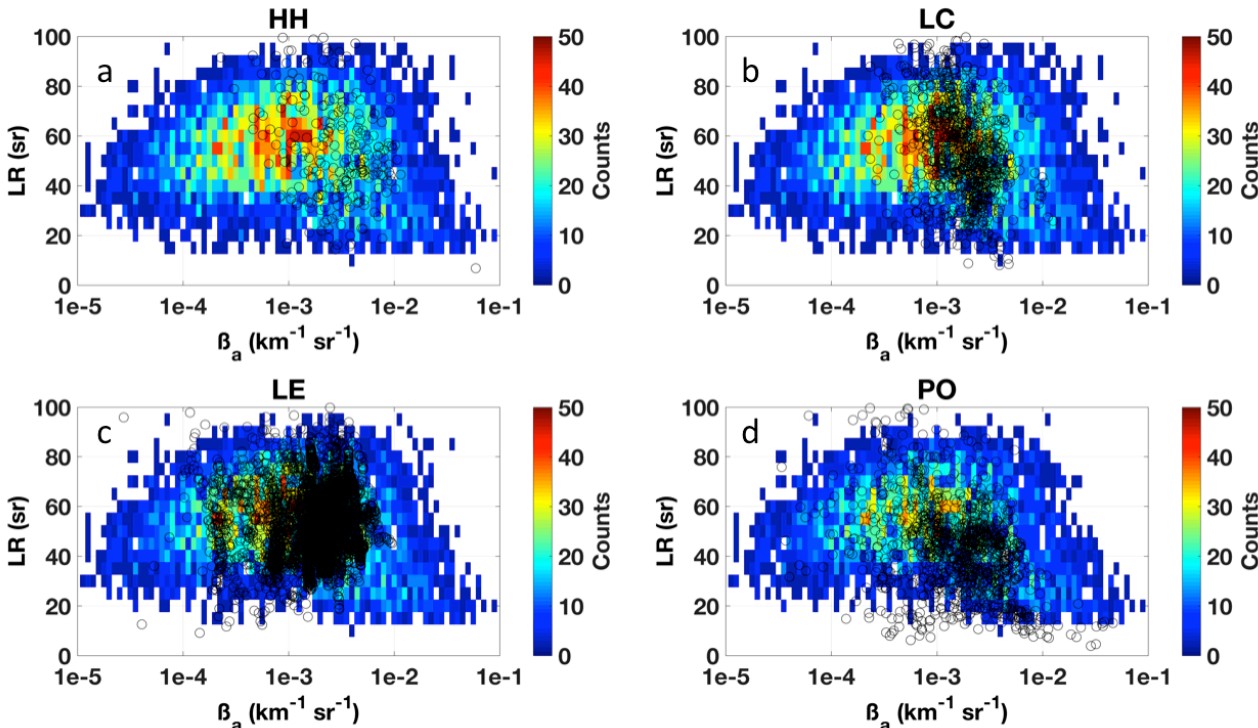

1007

Figure 4. Scatterplots of LR (sr) versus $\beta_a$ (km$^{-1}$ sr$^{-1}$) at 355 nm as simulated by our model (colored region) and measured by EARLINET lidars (black open circles) in Hamburg (Germany) (a), Lecce (Italy) (b), Leipzig (Germany) c) and Potenza (Italy) (d). The color area is the region of simulated values, the color code indicating the number of simulated values in each $\beta_a$ -LR pair (see legend). In particular, the color-2D histogram is computed using a semi-logarithmic box consisting of 10 equally spaced bins per decade of $\beta_a$ in the x-axis and 5 spaced LR values in the y-axis.


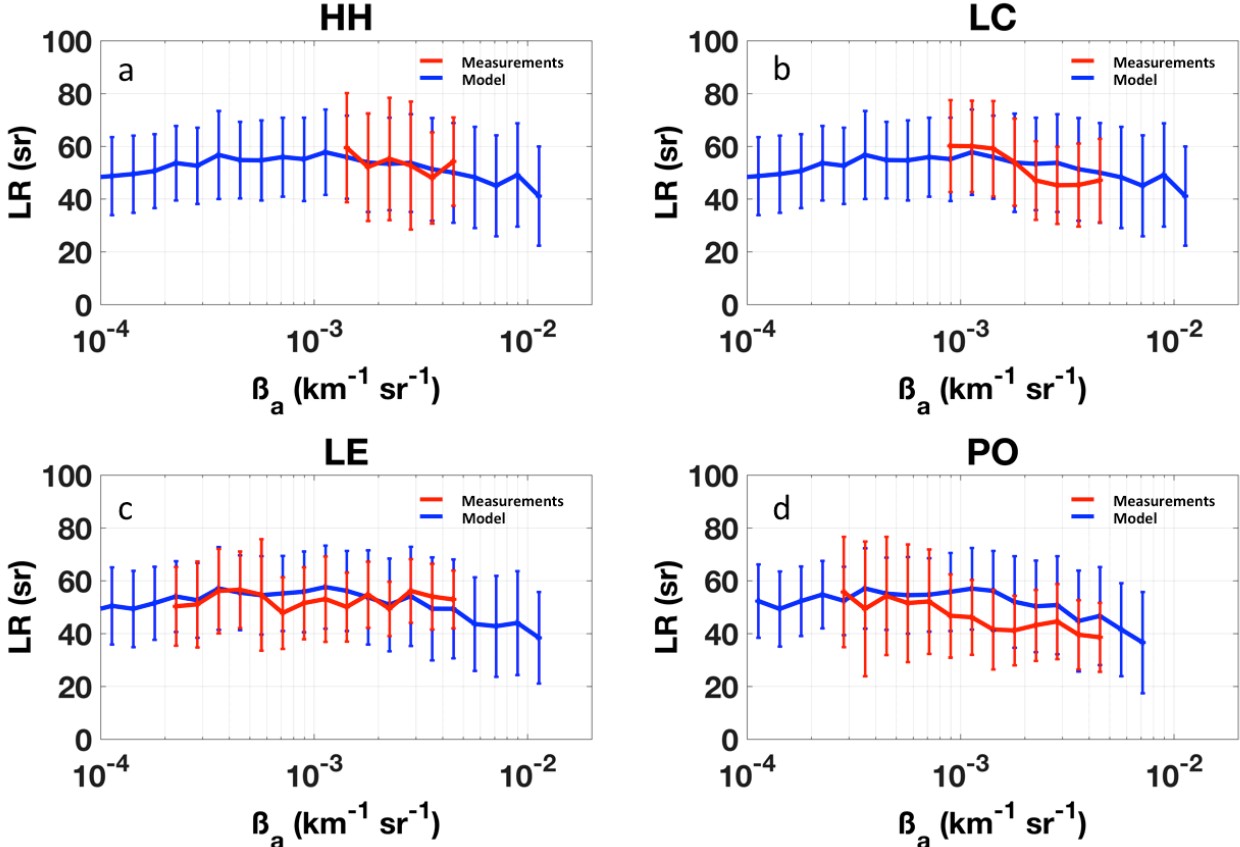


**Figure 5. Model-simulated (blue) and lidar measured (red) LR vs ß$_a$ mean curves at 355 nm calculated per 10 equally spaced**
**bins per decade of ß$_a$ in a) Hamburg, b) Lecce, c) Leipzig, and d) Potenza EARLINET lidar station. Vertical bars are the**
**associated standard deviations.**

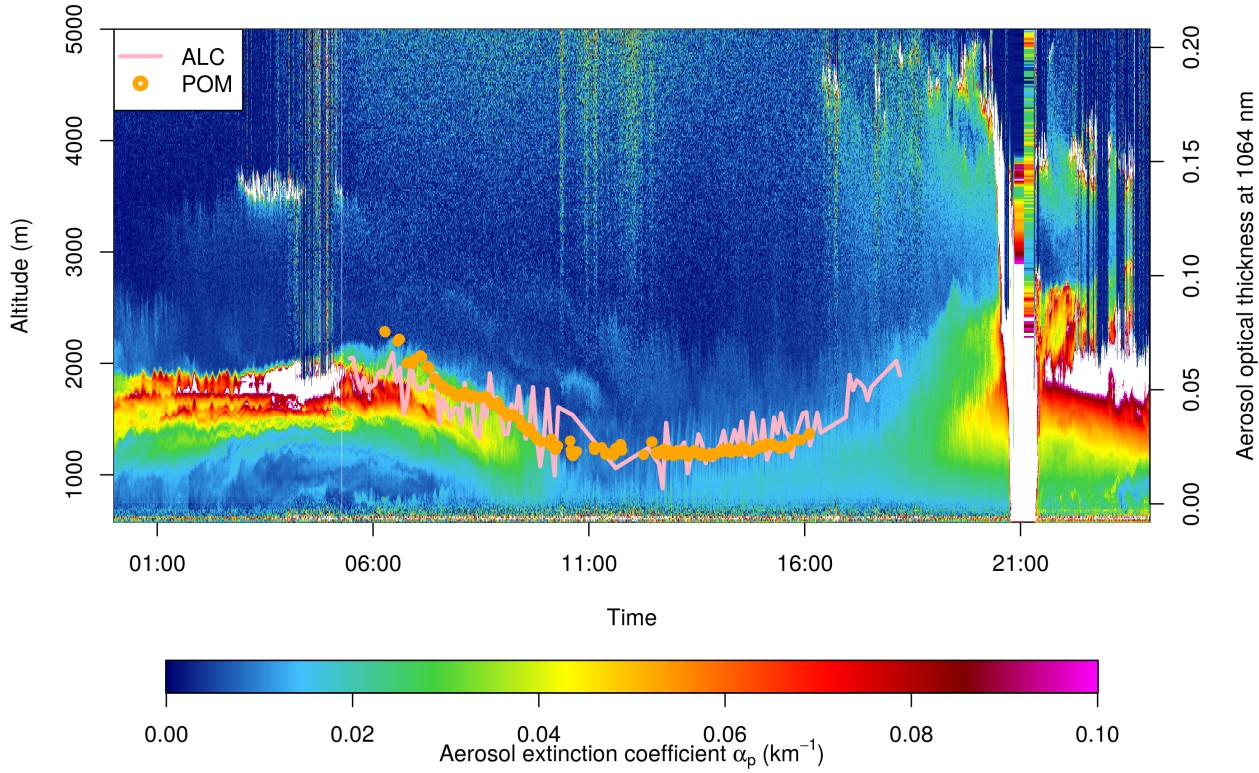


**Figure 6. Time-height cross-section of the aerosol extinction coefficients $\alpha_a$ [km$^{-1}$], as derived at 1064 nm on 26 June 2016 by the ALICENET ALC of Aosta San Christophe (Northern Italy). The orange circle points and the pink line are the AOT values (right y-axis, panel b) measured by a co-located POM-02L radiometer and estimated from the ALC following our approach.**


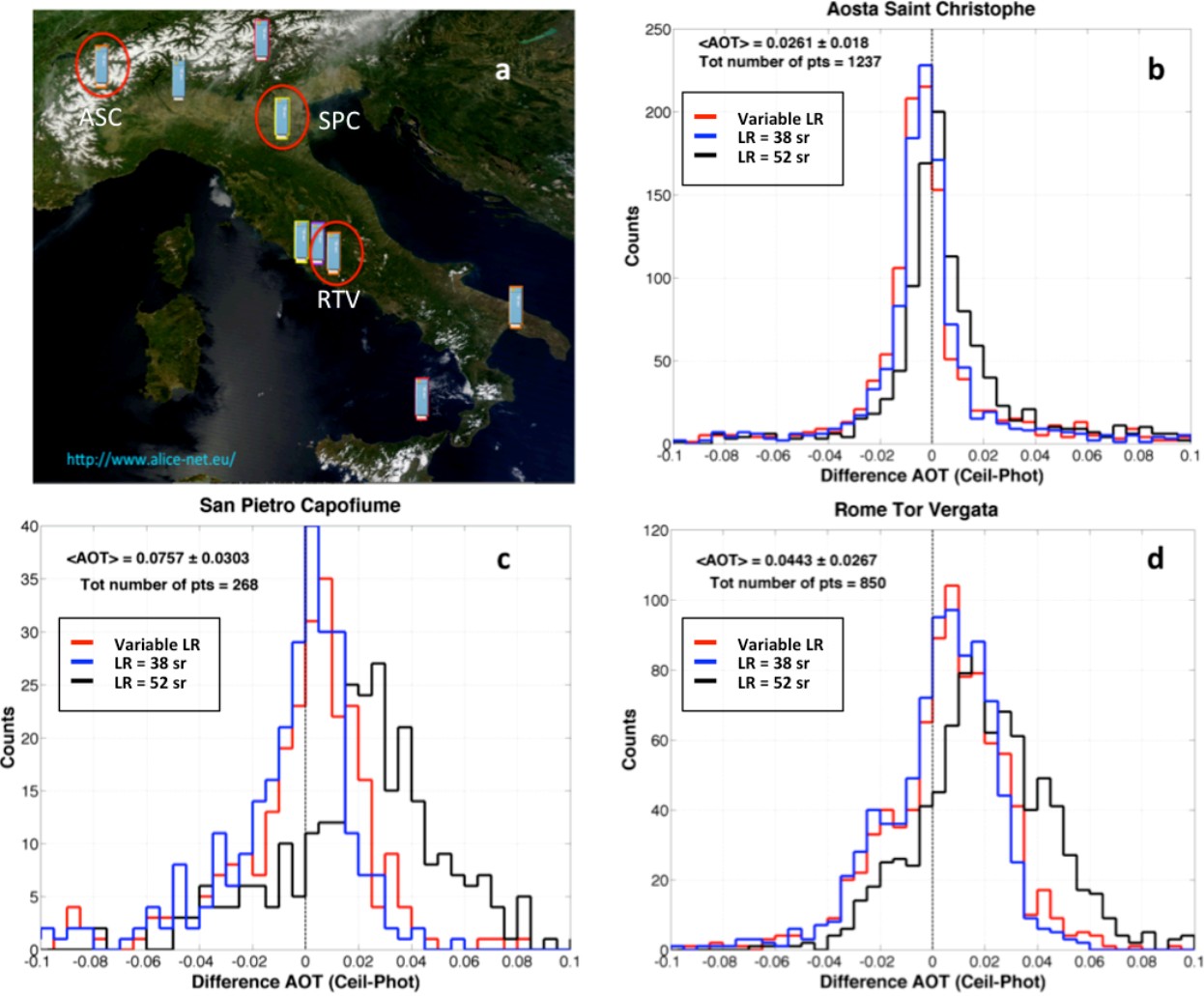


Figure 7. a) geographical map of the ALC network ALICENET. The red circles highlight the selected sites for this study: Aosta San Christophe (ASC), San Pietro Capofiume (SPC) and Rome Tor Vergata (ASC). b-d) histograms of the differences between the hourly-mean coincident AOTs at 1064 nm as derived by ALCs and measured by photometers, at ASC, SPC and RTV, respectively. The different colors (red, blue and black) depict the different inversion schemes: model-based inversion scheme, LR = 38 sr and LR = 52 sr, respectively. In each panel the values of the average measured AOT (and its associated standard deviation) and of the number of considered pairs are also reported.



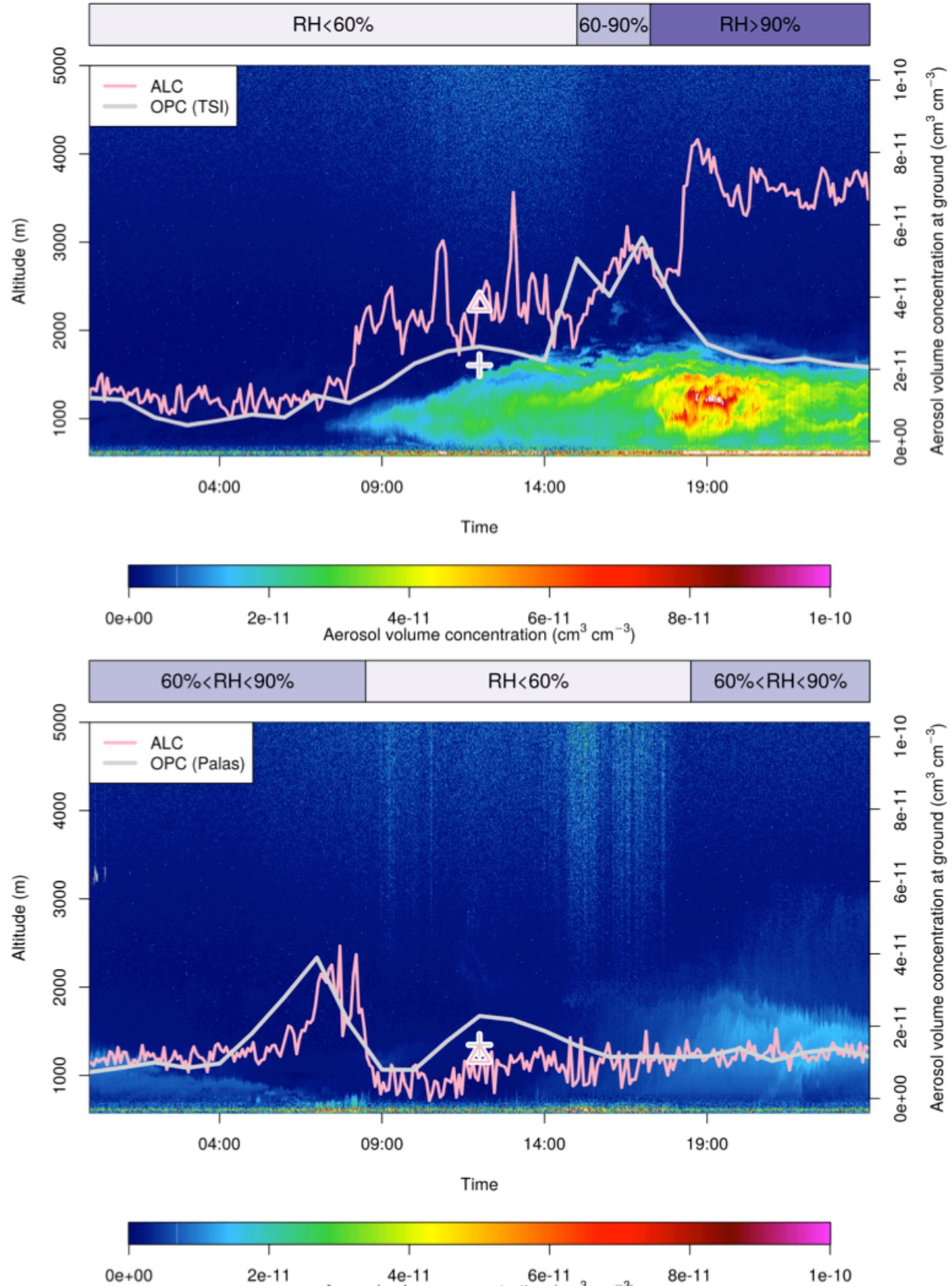


**Figure 8. Time-height cross-section of the aerosol volume concentration at Aosta San Christophe for 29 December 2016**
**(upper panel) and 05 September 2017 (lower panel). The right y-axis reports the volume concentration measured at surface**
**through TSI and Fidas®200s OPCs (upper and lower panels, grey curves) and the ALC-derived volume concentration at 75**
m (pink curves). The grey crosses and the pink triangles refer to the daily mean aerosol volume value derived by OPCs and
ALC measurements, respectively. The horizontal bars in the upper part of the panelse indicate the ranges (RH<60%,
60%<RH<90% and RH>90%, respectively) of the measured in-situ RH during the analyzed days.

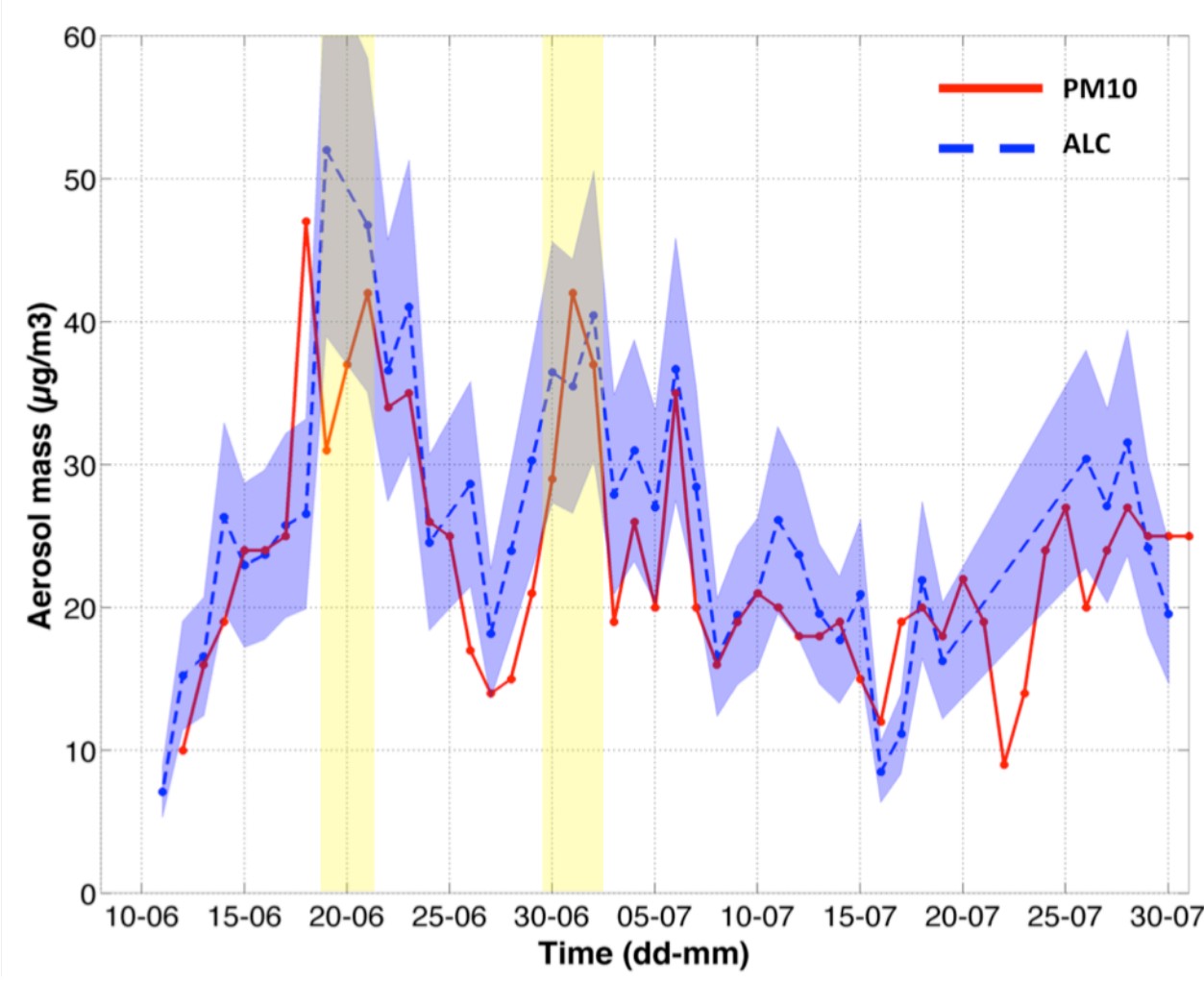


Figure 9. Daily-resolved aerosol mass concentration at SPC, for the period June – July 2012, estimated from ALC-derived
aerosol volume data at 225 m a.s.l. converted into mass using a fixed particle density $\rho_a = 2$ µg/m$^3$ (blue dotted line) and a
variable $\rho_a$ between 1.5 -2.5 µg/m$^3$ (shaded blue area). The red solid line is the daily PM10 concentration as measured by the
local Air Quality agency (ARPA). Vertical yellow shaded stripes indicate the presence of dust events.

**Appendix A: Model sensitivity tests**

To evaluate the proposed continental model configuration (hereafter CM0) and discuss its sensitivity to the variability
of the employed parameters, an overview of the impact on the model results produced by changing the limit of the
variability ranges of these parameters (i.e. using different model configuration, CMX) is given in this section.
The varied model (CMX-CM0) mean difference on the considered optical property (OP) has been quantified through
the following equation:
$$< \frac{dOP}{OP} > = \left(\frac{1}{Nbin}\right) \cdot \sum_{i=1}^{Nbin}[(< OP_{CMX} >_i - < OP_{CM0} >_i)/< OP_{CM0} >_i], \qquad (A.1)$$
where $N_{bin}$ is the total number of defined bins of $\beta_a$.
The results of the mean differences of $\alpha_a$ and LR for different ranges of $\beta_a$ and for the whole $\beta_a$ interval are reported on
table A1, where relevant sensitivity cases (i.e. relative mean difference greater than 1%) at $\lambda$=355 nm have been taken
into account.
CM1 refers to a model configuration without the first aerosol mode ($N_1$%=0). The overall decrease on the values of $\alpha_a$
and LR (around 3-4%) is due to the sum of significant and opposite effects for low and high values of $\beta_a$ where
$<d\alpha_a/\alpha_a>$ and $<dLR/LR>$ are of the order of -6% and 8%, respectively. Removing the coarser aerosol mode ($N_3$%=0),
causes positive mean values for $<d\alpha_a/\alpha_a>$ and $<dLR/LR>$ of the order of 5% (sensitivity case CM2). In this case, the
largest impact is observed for the $\beta_a$ range between $2\times10^{-4}$ and $2\times10^{-3}$ km$^{-1}$sr$^{-1}$.
An opposite result is obtained by decreasing the upper bound of the $r_2$ variability range ($r_2$=0.03 – 0.05 µm, CM3). In
fact also this model configuration leads to lower $\alpha_a$ and LR ($<d\alpha_a/\alpha_a>$ and $<dLR/LR>$ are equal to -6%, approximately).
In this case, the variation on the $r_2$ parameter affects the higher ranges of $\beta_a$ ($\beta_a$=$2\times10^{-4}$-$2\times10^{-2}$ km$^{-1}$sr$^{-1}$). Higher modal
radii for the coarse-mode particle ($r_3$=1 – 1.2 µm) in CM4 configuration leads to the increase of the contribution of
model-generated points with higher $\beta_a$ and causes lower values of $\alpha_a$ and LR ($<d\alpha_a/\alpha_a>$ and $<dLR/LR>$ are equal to -5%,
approximately) only for high values of $\beta_a$ ($\beta_a$=$2\times10^{-3}$-$2\times10^{-2}$ km$^{-1}$sr$^{-1}$), whereas the effect over the whole $\beta_a$ range is
around -1%.
The CM5 configuration accounts for the presence of more absorbing particles in the first aerosol mode, where the lower
bound of $m_{1im}$ has been increased by a factor of 10 ($m_{1im}$ =0.1-0.47). This produces a significant effect only for the
lower values of $\beta_a$ ($\beta_a$=$2\times10^{-5}$-$2\times10^{-4}$ km$^{-1}$sr$^{-1}$), with an increase of $\alpha_a$ and LR of approximately 4%. On the contrary,
increasing the lower bound of the real part of the second aerosol mode refractive index ($m_{2r}$ =1.55-1.70) has a large
impact on the considered parameters. In fact, the CM6 configuration largely underestimates both $\alpha_a$ and LR (around -
15% for both parameters) for all $\beta_a$ ranges.
The CM7 configuration refers to the impact of the total number of particles at the ground ($N_{tot}$). In this case, decreasing
the upper bound of the variability range of $N_{tot}$ by a factor of 2 ($N_{tot}$ =500-5000 cm$^{-3}$) lowers the mean values of $\alpha_a$ and
LR of around 5%. Nevertheless, this effect is totally due to the contribution of the $\beta_a$ values between $2\times10^{-3}$ and $2\times10^{-2}$
km$^{-1}$sr$^{-1}$, where $<d\alpha_a/\alpha_a>$ and $<dLR/LR>$ are around -10%. Assuming no increase with altitude for $\sigma_{1,2}$ (sensitivity case
CM8) produces relevant differences on the mean values of $\alpha_a$ and LR. In CM8, the overall overestimation of these two
parameters is quite limited ($<d\alpha_a/\alpha_a>$ = 6.3 and $<dLR/LR>$ =6.4), whereas a large and opposite impact is observed for

low and high values of $\beta_a$. In fact, $\langle d\alpha_a/\alpha_a\rangle$ ($\langle dLR/LR\rangle$) is equal to -14.1 (-13.9) and 18.5 (19.0) for $\beta_a = 2\times10^{-5} - 2\times10^{-4}$

and $\beta_a = 2\times10^{-5} - 2\times10^{-4}$ km$^{-1}$sr$^{-1}$, respectively. As explained by Barnaba et al. (2007), the dependence of $\sigma_{1,2}$ to the

altitude can be associated to the fact that, when increasing the distance from the main aerosol sources, the particle

processing is more efficient.

**Table B1. Mean differences of $\alpha_a$ and LR between different model sensitivity cases and the proposed continental model configuration.**

| Model configuration | $\beta_a$ (km$^{-1}$sr$^{-1}$) $2\times10^{-5}$-$2\times10^{-4}$ | | $\beta_a$ (km$^{-1}$sr$^{-1}$) $2\times10^{-4}$-$2\times10^{-3}$ | | $\beta_a$ (km$^{-1}$sr$^{-1}$) $2\times10^{-3}$-$2\times10^{-2}$ | | $\beta_a$ (km$^{-1}$sr$^{-1}$) $2\times10^{-5}$-$2\times10^{-2}$ | |
|---|---|---|---|---|---|---|---|---|
| | $\langle d\alpha_a/\alpha_a\rangle$ (%) | $\langle dLR/LR\rangle$ (%) | $\langle d\alpha_a/\alpha_a\rangle$ (%) | $\langle dLR/LR\rangle$ (%) | $\langle d\alpha_a/\alpha_a\rangle$ (%) | $\langle dLR/LR\rangle$ (%) | $\langle d\alpha_a/\alpha_a\rangle$ (%) | $\langle dLR/LR\rangle$ (%) |
| CM1 ($N_1\%=0$) | -6.2 | -6.4 | 3.1 | 3.2 | 7.8 | 7.9 | -3.7 | -3.5 |
| CM2 ($N_3\%=0$) | 4.7 | 4.9 | 8.6 | 8.9 | 2.8 | 2.7 | 5.3 | 5.4 |
| CM3 ($r_2=0.03-0.05$ μm) | -2.0 | -1.7 | -10.3 | -10.2 | -8.9 | -8.2 | -6.7 | -6.4 |
| CM4 ($r_3=1.0-1.2$ μm) | <1 | <1 | -2.1 | -2.0 | -5.24 | -5.3 | -1.2 | -1.0 |
| CM5 ($m_{1im}=0.1$-$0.47$) | 4.3 | 4.2 | <1 | <1 | <1 | <1 | 1.8 | 1.8 |
| CM6 ($m_{2r}=1.55-1.70$) | -10.9 | -10.9 | -16.2 | -16.3 | -18.9 | -19.1 | -15.3 | -15.3 |
| CM7 ($N_{TOT}=500$-$5000$) | <1 | <1 | <1 | <1 | -11.2 | -10.7 | -3.7 | -3.5 |
| CM8 ($\sigma_1$, $\sigma_2$ constant) | -14.1 | -13.9 | 6.4 | 6.1 | 18.5 | 19.0 | 6.3 | 6.4 |

**Appendix B: Model-based functional relationships at 355 and 532 nm**

The parameters of the seventh-order polynomial fit used to derive the functional relationships between log(x) and log(y)
(where x = ß$_a$ and y =α$_a$, S$_a$ or V$_a$) at λ = 355 and 532 nm are reported in Tab. A1 and Tab. A2, respectively.

**Table A1. Parameters of the Seventh-Order Polynomial Fits (y = a$_0$+ a$_1$x+a$_2$x$^2$+a$_3$x$^3$+a$_4$x$^4$+a$_5$x$^5$+a$_6$x$^6$+a$_7$x$^7$) for λ = 355 nm, with**
**x=log(ß$_a$) (in km$^{-1}$ sr$^{-1}$ unit) and y=log(α$_a$, S$_a$, or V$_a$) in (km$^{-1}$, cm$^2$/cm$^3$ and cm$^3$/cm$^3$, respectively).**

| Functional relantionship at 355 nm | Extinction coefficient | Surface area | Volume |
|---|---|---|---|
| a$_0$ | 3.797837507651898 | 12.019452592845141 | -5.314834128998254 |
| a$_1$ | 3.294032541389781 | 30.825966279368547 | 2.500484347793244 |
| a$_2$ | 0.962603336867675 | 24.518531616019207 | -1.196109537503000 |
| a$_3$ | 0.241796629870675 | 10.625241994796593 | -1.583236058579546 |
| a$_4$ | 0.064609145804688 | 2.634051072085453 | -0.681801883947768 |
| a$_5$ | 0.017721752150233 | 0.373150843707711 | -0.145232662646142 |
| a$_6$ | 0.002722551625862 | 0.027971628176431 | -0.015471229968392 |
| a$_7$ | 0.000157245409783 | 0.000854381337164 | -0.000658925756875 |



**Table A2. Parameters of the Seventh-Order Polynomial Fits** ($y = a_0 + a_1x + a_2x^2 + a_3x^3 + a_4x^4 + a_5x^5 + a_6x^6 + a_7x^7$) **for** $\lambda = 532$ **nm, with**
$x = \log(\beta_a)$ **(in km$^{-1}$ sr$^{-1}$ unit) and** $y = \log(\alpha_a, S_a, \text{ or } V_a)$ **in (km$^{-1}$, cm$^2$/cm$^3$ and cm$^3$/cm$^3$, respectively).**

| Functional relantionship at 532 nm | Extinction coefficient | Surface area | Volume |
|---|---|---|---|
| $a_0$ | 3.797837507651898 | 12.019452592845141 | -5.314834128998254 |
| $a_1$ | 3.294032541389781 | 30.825966279368547 | 2.500484347793244 |
| $a_2$ | 0.962603336867675 | 24.518531616019207 | -1.196109537503000 |
| $a_3$ | 0.241796629870675 | 10.625241994796593 | -1.583236058579546 |
| $a_4$ | 0.064609145804688 | 2.634051072085453 | -0.681801883947768 |
| $a_5$ | 0.017721752150233 | 0.373150843707711 | -0.145232662646142 |
| $a_6$ | 0.002722551625862 | 0.027971628176431 | -0.015471229968392 |
| $a_7$ | 0.000157245409783 | 0.000854381337164 | -0.000658925756875 |


 **Appendix C: Model – EARLINET comparison at 532 nm**

Figure C1 depicts the result of the comparison between EARLINET stations and our developed model (red and blue
curves, respectively) in terms of 'mean' LR per bin of $\beta_a$ at $\lambda$=532 nm. Note that only $\beta_a$ bins containing at least 1% of
the total modeled data were considered. Similarly to the results at 355 nm shown in section 4.1, a general good
agreement between the modeled and the measured LR values is found. As attested by the low value of the mean
discrepancy of Table 6, the modeled curve well fits with Madrid observations. Some major deviations are found for
Lecce, which, however, at 532 nm, has a very low number of considered points (i.e. 109).

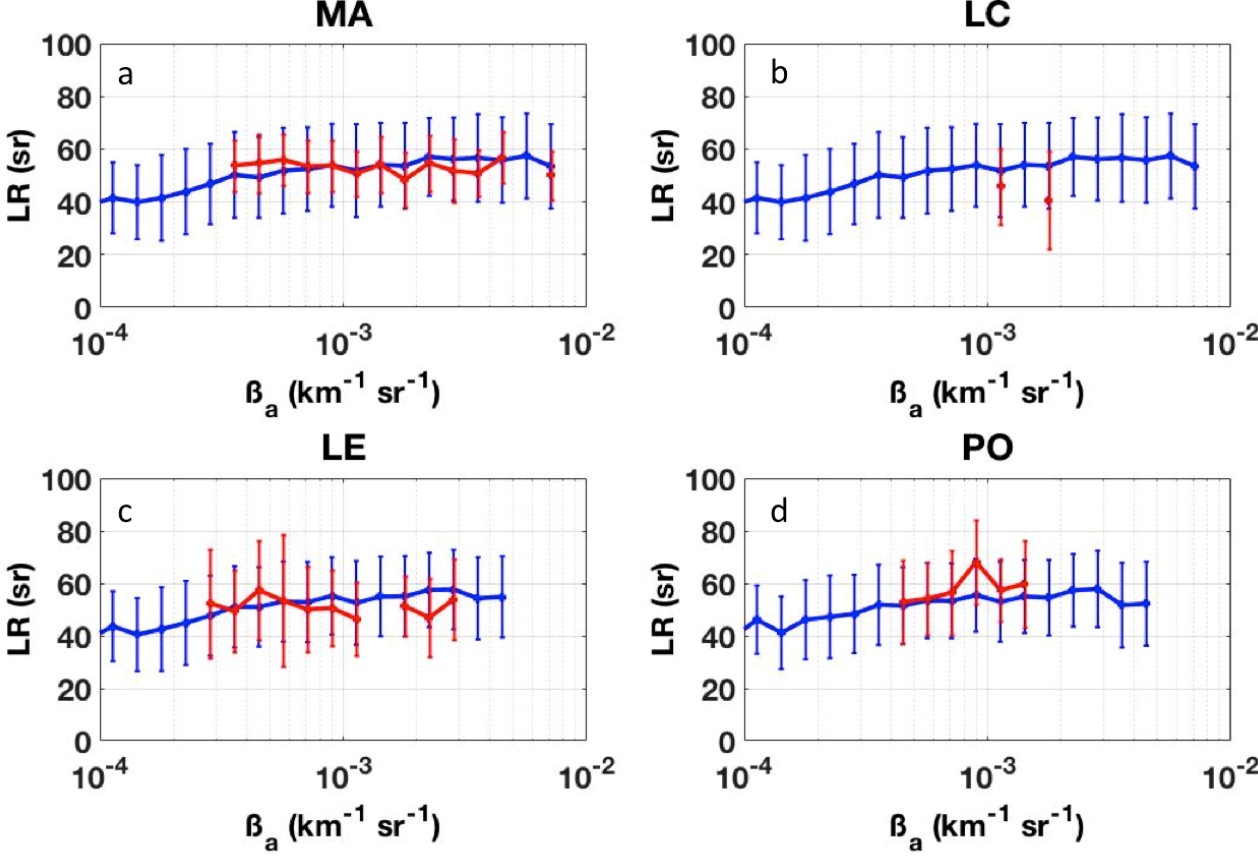


**Figure C1. Model-simulated (blue) and lidar measured (red) LR vs ßa mean curves at 532 nm calculated per 10 equally**
**spaced bins per decade of ßa in a) Madrid, b) Lecce, c) Leipzig, and d) Potenza EARLINET lidar station. Vertical bars are**
**the associated standard deviations.**
**Appendix D: Model sensitivity tests for optimal configurations at LC and PO sites**
According to the results reported in Tab. B1, two model configurations (CM0a and CM0b) have been set up to better
reproduce the EARLINET observations of LR vs $\beta_a$ at LC and PO sites, respectively. The comparison between these
two configurations, the EARLINET measurements and the CM0 set-up are illustrated in Fig. B1 (panel a and b for LC
and PO, respectively) in terms of LR mean value curves per 10 equally spaced bins per decade of $\beta_a$. Blue and red
colors have the same meaning of Fig. 5 (i.e. CM0 model and observation curves, respectively), black curves refer to the
LR vs $\beta_a$ estimated through the CM0a and CM0b model versions for LC and PO stations, respectively. Vertical bars are
the associated standard deviations.
The only difference between CM0a and CM0 configuration consists in the upper bound of the variability range of $N_{tot}$
(5000 vs 10000 cm$^{-3}$ at ground, respectively). This modification seems to fit the observed LR vs $\beta_a$ behavior at 355 nm.
The upper bound $N_{tot}$ value is similar to the one (i.e. $N_{tot}$ upper bound =3000 cm$^{-3}$ at ground) used in the work of
Barnaba et al. (2007) to characterize the optical properties of the continental aerosol present over southeastern Italy.
The computed mean model-measurement LR relative difference between CM0a configuration and LC Earlinet
measurements is around 5%.
Similarly, the CM0b configuration uses the same value for the upper bound of $N_{tot}$ variability range and, in addition,
higher values of the $r_3$ variability range of (1.0 – 1.2 μm vs 0.3 - 0.5 μm, respectively). As highlighted by the panel b of
Fig. B1, this model configuration allows well reproducing the LR vs $\beta_a$ behavior derived by EARLINET lidar Raman
measurements at 355 nm. This result seems to indicate the presence of coarser aerosols in a clean continental
environment. In comparison to the CM0 model, the mean model-measurement LR relative difference decreases from
17% to 6%.

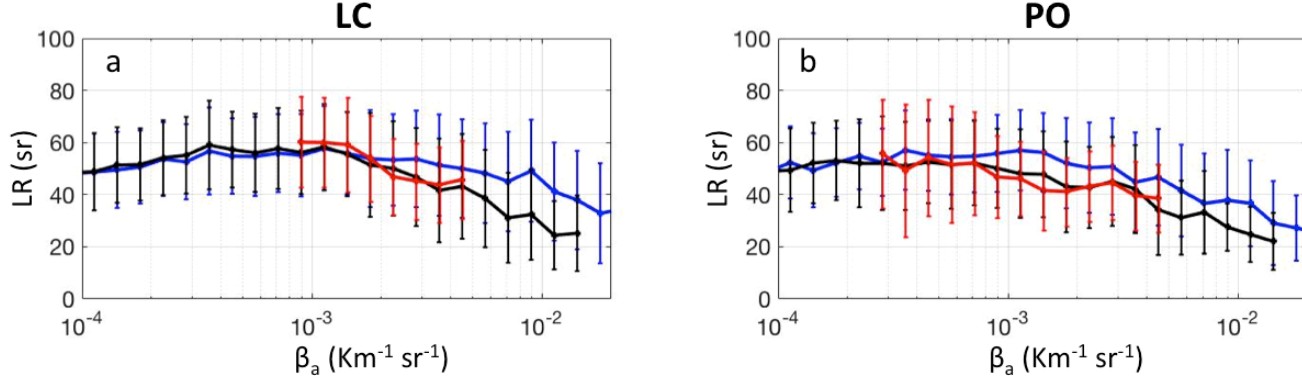


**Figure D1. Model-simulated (blue and black lines) and lidar measured (red lines) LR vs $\beta_a$ mean curves at 355 nm calculated**
**per 10 equally spaced bins per decade of $\beta_a$ for the LC and PO EARLINET lidar stations (panel a and b, respectively). Blue**
**color refers to CM0 model configuration, black color to CM0a and CM0b model configurations adapted to LC and PO sites,**
**respectively.**

 **Appendix E: ALC vs sun-photometer AOTs**

To have sense of both absolute and relative errors of AOT, we reported in this section the scatter plots between the
hourly-mean coincident AOTs at 1064 nm as derived by ALC model-based approach and those measured at 1020 nm
by the sun-photometers installed at RTV, SPC and ASC, respectively (Figure E1, E2 and E3). The corresponding linear
fit y = bx (red line), where x = sun-photometer AOT, y = Nimbus CHM15k AOT are also shown in the plots. The
values of the correlation coefficients for the three sites (R = 0.77, R=0.72 and R=0.73 for RTV, SPC and ASC,
respectively) attest a relatively good agreement between the two AOT measurements.

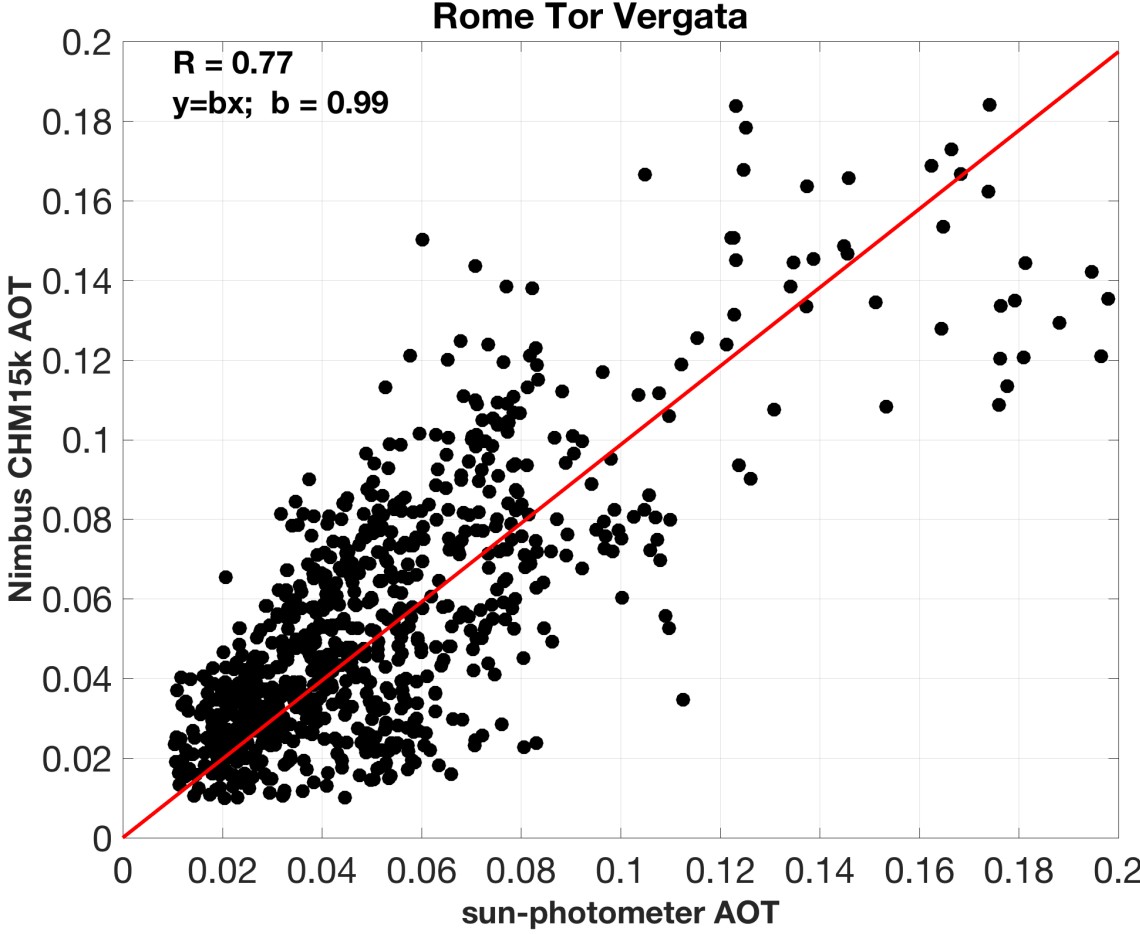


**Figure E1. Scatter plot between the hourly-mean coincident AOTs at 1064 nm as derived by the ALC model-based approach**

**and measured at 1020 nm by the AERONET photometer at RTV. The red line represents the linear fit y = bx between the**

**two datasets, where x = sun-photometer AOT; y = Nimbus CHM15k AOT.**


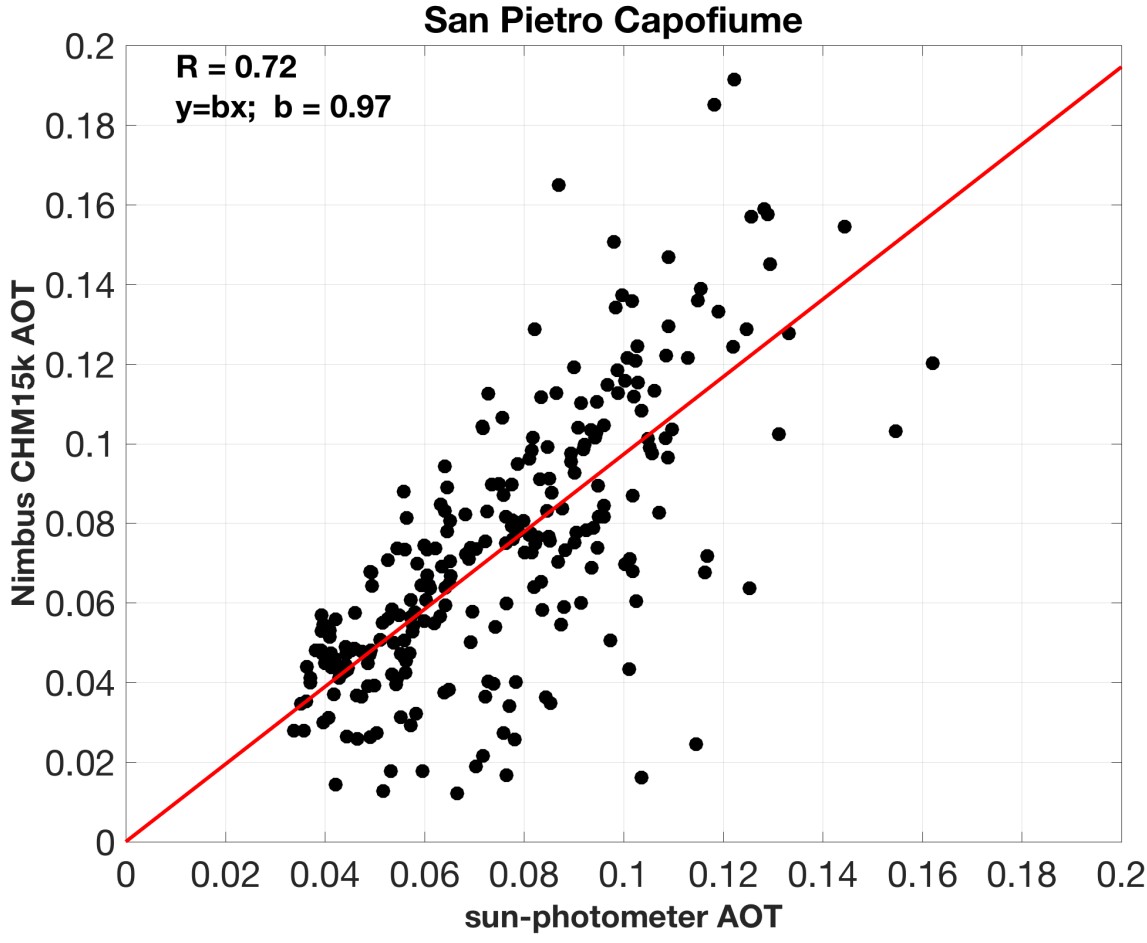


**Figure E2. Scatter plot between the hourly-mean coincident AOTs at 1064 nm as derived by the ALC model-based approach**
**and measured at 1020 nm by the SKYRAD photometer at SPC. The red line represents the linear fit y = bx between the two**
**datasets, where x = sun-photometer AOT; y = Nimbus CHM15k AOT.**





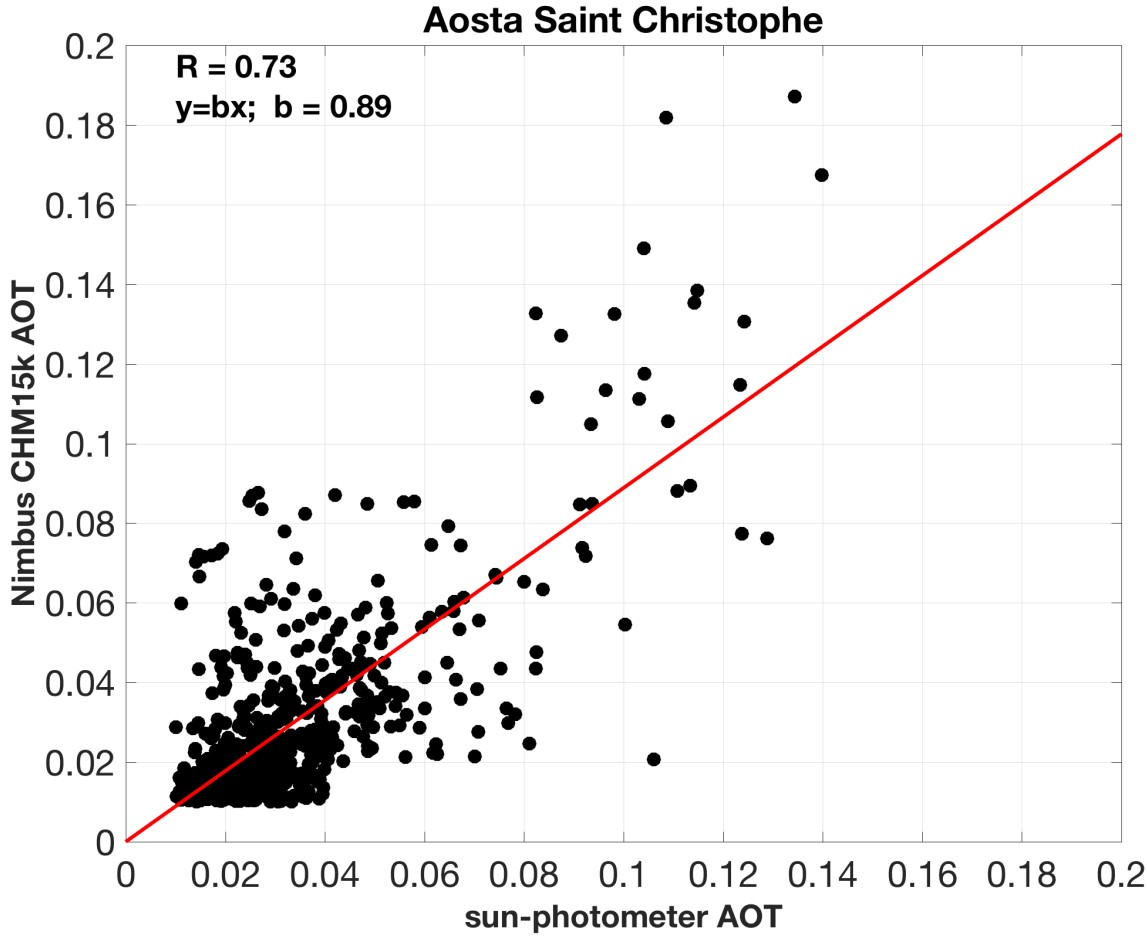


Figure E3. Scatter plot between the hourly-mean coincident AOTs at 1064 nm as derived by the ALC model-based approach
and measured at 1020 nm by the SKYRAD photometer at ASC. The red line represents the linear fit y = bx between the two
datasets, where x = sun-photometer AOT; y = Nimbus CHM15k AOT.



