# Peer review of "A multi-wavelength numerical model in support to quantitative"

_Atmospheric Measurement Techniques, 2018_

## Referee Comment (RC1) · Anonymous Referee #3 · 7 May 2018

General comments: The authors present an interesting study about retrieving aerosol properties (extinction coefficient (E), surface area (S) and volume (V)) from lidar and/or automated lidar-ceilometer (ALC) backscatter measurements. The key of the method is using a "Monte-Carlo" model to simulate the relationship between E, S, V and backscatter for different continental aerosol microphysical properties which could occur in real life and then implementing the relationships in the retrievals. Based on the 20000 model simulations, the relationship between lidar backscatter and aerosol E, S, V were

investigated and dependence of lidar ratio (LR) to the backscatter at three lidar wavelengths (355 nm, 532 nm, 1064 nm) were fitted. The model-based LR were tested by comparing model simulations with raman lidar observations at 355 nm and found agree well with observations. Then the method was implemented to retrieve AOT and aerosol volume, PM10 and the results were compared with in-situ measurements. Although this method has some limitations in retrieving aerosol volume, mass, it shows the potential of using ALC for aerosol properties retrieval. The paper is well written and structured. The method was explained clearly, and main assumptions and limitations of the method were discussed. The topic is well suited for the AMT. I have a few comments and recommendations before the paper can be published.

Line 128: Both the step 1 and step 2 are about the aerosol model, why didn't authors put them both in the same section (section 2)?

Line 140: Is the r_mi at here same as the r_i in the equation 1 or it is another parameter? What are the m_(r i) and m_(im i)?

Line 143: what are the specific rules?

Line 147: The description of m_(r i) and m_(im i) should be given at the first time when they appeared in the paper, see the related comment above. Secondly, more explanations about the real and imaginary refractive indices and how they are used in the aerosol optical properties calculation should be provided.

Line 151: What is the exact size range? The authors should indicate the range or refer the tables which shows the range of the parameters at here. Same as the mode 1, 2 and 3.

Line 154: Why did the authors only use those values at 355 nm?

Line 175-182: How did the authors decide to use those equations to stand the altitude-dependence? Some references should be added here. What is the BG1? Is it the BG01 mentioned before?

Line 197: The reference of the Mie theory or code should be added here.

Line 203: For mode 1 and mode 2, only the values of m_(r i) and m_(im i) at wavelength at 355 nm were introduced. How did the authors get the value at 1064 nm?

Line 215: The "(A)" should be after the "average".

Line 242: The maxima is the maxima of the fitting curve but not the maxima of the all samples. Right?

Line 245: For the wavelength 1064 nm, there are some samples with LR larger than 80 based on the figure 3c.

Line 292: Are the relative errors the errors of lidar measurements? What are the standard measurements (truth)?

Line 298: 5 sites were chosen, but why there are only 4 sites depicted in the figure 4.

Line 313: ãĂŰLRãĂŮ_mod,ãĂŰLRãĂŮ_meas should be explained at here.

Line 329: The table 6 should be referred at here.

Line 366-371: Although the retrieval method was introduced by other scientists before, it is better to discuss more about how to derive aerosol extinction from the ALC e.g. show the key equations. Audiences may have questions like what are the raw data of the ALC? Are the raw data the range corrected backscatter? Does the raw data already consider the attenuation of signal from height z to surface due to aerosol and molecular extinction?

Line 389: What made the authors to choose this threshold of AOT for cases screen?

Line 408-421: It is suggested to give a AOT VS AOT scatter plot at here. Then it will help the audiences to have sense of both absolute and relative errors of AOT.

Line 431: With the fixed LR=52, the bias (<|dAOT|>=0.021 and 0.006) are smaller than the model-based bias (<|dAOT|> =0.11, 0.13) shown in line 428, right? Why the

authors said it is larger?

Line 440-444: How did the authors calculate the aerosol volume? Was the retrieval based on equation 7 and 10? Authors should explain more about the retrieval at here.

Line 460-470: The hygroscopic growth could induce large differences between the in-situ measured and the ALC retrieved aerosol volume. In the work of Siwei Li et al. (2016, 2017), they discussed the impacts of aerosol size distribution in the retrieval of PM2.5 using ceilometers (Li et al., 2016) and relationship between relative humidity and PM2.5/ceilometer-backscatter ratio (Li et al., 2017). More discussion about volume, PM retrieval and comparisons of model-based retrieval with in-situ measurements e.g. model vs in-situ scatter plot should be added here. Adding aerosol size (can compare the in-situ measurements and angstrom exponent) and relative humidity information and analysis at here may help the authors to support their conclusion.

Line 477: What specific aerosol densities did the authors use in the retrieval and why?

Line 485: Why did the authors use different heights in estimation of surface aerosol volume (0-75 m) and mass (at 225 m)?

Line 488-490: Were the mean and relative difference between the two-series based on hourly average PM10 or daily average PM10? What is the absolute difference? What is the R between them?

Line 496: Where is the close bracket?

Line 549: Surface area and volume are not the optical properties. The results from this work showed that ancillary data information are needed to get accurate aerosol properties e.g. volume, mass. . ..

Line 551: What kind of meteorological monitoring can be provided by the method?

---

## Referee Comment (RC2) · Anonymous Referee #2 · 22 May 2018

General comment: The approach described in this manuscript can provide a helpful expansion of the data analysis of lidar-ceilometers. How much information can be added to the data however depends on the input to the model. Here, the model is trained with observations representing a continental European aerosol. Thus the results are representative for this type of aerosol and regions/times where/when it occurs. The approach in general is able to add significant benefit to climatologies derived from lidarceilometer networks and should therefore be published. The presentation of methods

and results is sound. Limitation of the applicability to specific situations have mostly clearly been addressed by the authors. The manuscript deals with as-far-as-possible exploitation of ceilometer data – that is good. But at the point of estimating mass concentrations (see below) I have concerns, because the results you show suggest that PM10 can be estimated within 10-20% accuracy by ceilometers, which I don't think is generally true. Given the uncertainties and assumptions involved, the presented time series comparison may not even be representative for your sites at all times. Though inversion of optical data is often remarkably good-natured and your 'calibration' works for the related conditions and regional climate, this does not take into account the complexity of PM10 measurements which reflects in the +/- 25% measurement accuracy in the EC 2008/50 directive.

The manuscript presents a model-based approach to infer extinction coefficients, particle surface- and volume/mass-concentrations from backscatter coefficients measured by lidar-ceilometers, based on statistical relations. Mie-calculations are performed for empirical ranges of particle sizes and refractive indices, yielding conversion factors which are stored into a look-up table. As the influence of unsperical particles is shortly discussed, the Mie approach seems sufficient. The aerosol modal representation and refractive indices for model input are based on a comprehensive literature survey. Owing to the size distributions and the range of refractive indices used for the ensemble calculations, it is valid for 'continental aerosol'. The inversion of ALC profiles uses state-of-the-art absolute calibration and the Rayleigh method according to Wiegner et al. 2012.

Evaluations are the most important part of the manuscript: The simulations (not including measurements) are evaluated against measurements: First the backscatter coefficient (BSC) vs Lidar ratio (LR) relation is compared against climatologies from EARLINET, CALIPSO and other networks. It is shown that average climatological LR are reproduced and that the frequency distribution of simulated BSC-LR pairs is roughly consistent with the corresponding distribution of EARLINET observations. There are, however, deviations, which are attributed to particle sizes and compositions which are, by design, not captured by the model.

Then, AOD inferred from CHM15k lidar-ceilometers are compared against each 1-2 years of data from 3 Italian stations with radiometers. Frequency distributions of the bias between inferred and measured AOD are shown and one example for illustration. The usual extrapolations of radiometers to 1064 nm and of the profile below 400m towards the ground (overlap) are done.

Thirdly, volume and mass concentrations are estimated from lidar-ceilometers, based on the proposed model and compared to in-situ measurements with optical particle spectrometers. Given your limitation to 'continental type aerosol', the large variability in essential conversion factors showing up in the statistical evaluations, uncertainties due to the overlap extrapolation, I wonder how representative these results are. As OPS measure dry aerosol while lidar/ALC measure optical parameters under ambient conditions: I can hardly believe that the parametrisation in your model and the information about atmospheric humidity as such is accurate enough to allow proper humidity correction in the range of the uncertainties given here. I think that these results can only be achieved under very specific conditions - aerosol type, stratification, homogeneity etc. This should be discussed in more detail.

Specific comments: Line 80: ... and affordable for aerosol applications, ...

Line 108: this is at best true in a climatological sense, but not on shorter time scales. But even on the long-term, Putaud et al. 2010 report large differences in the aerosol distribution over Europe

Line 181: In this formula mRH converges to 2m0 for rmi\_RH  $\mbox{ mmi_0, i.e. for a large aequous droplet. Replace by mRh = mW + <math display="inline">\ldots$

Line 187: In eq. 4 and 5, rim\_RH and miRH are the... should be ...rmi\_RH...

Line 225: It is unclear to me what that means - what is 1%?

СЗ

L 235: weighted-LR'  $\pm$  1 s. d. -> write: standard deviation

Line 305 and Fig 4: but only as a statistical ensemble average over all data without evaluating the temporal correlation

Line 319ff: Hamburg (and the others as well) is not really a continental site but considerable sea-salt contribution can be expected in the coarse mode, (at least for Hamburg) likely not less on average than from dust. So does the statistical agreement with your model results confirm the significance of your model? Are you sure, you'd get a worse agreement e.g. for Mace Head and would you expect to be able to draw significant conclusions about the aerosol type?

Line 420: why do you exclude desert dust days?

Line 434: you should note that these conclusions are valid only in a statistical sense

Line 452: the data from 0-75m are those from 300 m a.g. (where overlap correction is feasible) extrapolated to the ground?

Line 444ff: You should specify that these OPC channel data are given as diameters, while above you mostly discuss sizes in terms of radii

Line 480: what is the meaning of a particle density of 2  $\mu$ g/m3? Typical densities are of the order 1000 kg/m3. There seems to be a conversion factor included.

Line 486ff: Isn't the nocturnal boundary layer depth often in the range of few tens to hundreds of meters, even in summer? Strong vertical gradients in the lowest 200-300 m seem quite likely to me.

---

## Referee Comment (RC3) · Anonymous Referee #1 · 22 May 2018

The manuscript fits both the journal and special issue scopes, describing an interesting methodology to be applied to existing and/or future ceilometer networks to retrieve vertically-resolved aerosol microphysical parameters. Nevertheless, the paper shows some issues that I am very confident the authors will successfully address, making the manuscript ready for publication.

General Comments

[Figure]

It is very interesting to obtain microphysical aerosol properties using low-cost and low-maintenance instruments, however in the paper is missing a discussion on who is going to use the data (end users) and which are end users specific needs, in terms of accuracy. Likewise, it should be also introduced a subsection in the conclusions describing if the proposed approach meets the end user requirements.

The method associated uncertainty seems to be very optimistic. In fact, the uncertainty is mainly similar or a just a little greater with respect to the highly costly multi-wavelength lidar retrieval, developed by Veselovskii et al., 2002.

The model is based on a priori three-modal log-normal aerosol distribution. Changing the mixing ratio among modes and total particle number impact of course the result. The main problem linked with this approach, is that the model should be "tuned" on particular atmospheric condition (as for Lecce, where the total number of aerosols have been diminished, hypnotizing cleaner continental aerosols ).

Simulations from MonteCarlo are fitted with a 7th grade polynomial. Even if in the manuscript follows the approach described in Barnaba et al., papers, no explanation is given about this choice. More important, it is missing in the manuscript a model sensitivity study: how the results are affected, for example if a 3rd degree polynomial is used instead of 7? Is there for example a convergence in the results if the polynomial order is increased? Or is overfitting creating problems ? All those question should be properly addressed.

Specific Comments:

Pag 1 Line 19: Please be more specific about "continental aerosol type". Provide a very short description of the aerosol species belonging to this category.

Pag 2 Line 43: I suggest to classify aerosol effects on radiation budget into direct, semi-direct and indirect. Moreover, literature should be broaden (e.g. Feingold papers are missing)

Pag. 2 Line 54, again literature is poor, I would suggest to add at least Tosca et al., 2017 remote sensing

Pag.2 Line 57 I suggest to add reference to Lolli et al., 2018 AMT

Pag.2 Line 59-65 CALIPSO is not the only mission with lidar monitoring the atmosphere from space. Since few years there is also CATS. Please refer to York et al., 2016, McGill et al., 2016

Pag. 3 Line 85. I suggest to add to the reference Madonna et al., 2018 showing results of the new intercoparison campaign INTERACT-II

Pag. 3 Line 99 and 100: Both measurement units are wrong both for surface area and volume

Pag. 3 Line 107: please refer to the first comment

Pag. 4 Line 133: Please be more specific. Now the aerosol type is continental. Which is the difference with " average continental"?

Pag. 4 Line 139 Eq 1 suffers from hasty writing. Not all the variables are described in the manuscript

Pag. 4 Line 140 rmi is not present in Eq. 1

Pag. 4 Line 141 Even if it is clear , variables mri and mim are not defined here (few rows below yes) neither present in Eq.1

Pag. 4 Line 145 N shows wrong measurement units.

Pag. 5 Line 176-Hi is not defined. Why on equation 3 is arbitrarily used 5.5 Km ? A ref should be at least added.

Pag 7 Line 226 Measurement units are wrong.

Page 11 Line 399: Please specify which AERONET datasets were used in the manuscript and which version.

Pag 13 Line 465: usually humidity is higher at night
* * *

---

## Author Comment (AC1) · 9 Aug 2018

General Comments

**1 It is very interesting to obtain microphysical aerosol properties using low-cost and low maintenance instruments, however in the paper is missing a discussion on who is going to use the data (end users) and which are end users specific needs, in terms of accuracy. Likewise, it should be also introduced a subsection in the conclusions**

describing if the proposed approach meets the end user requirements.

- We thank the referee for his/her encouraging comments. As in other AMT-published methodologies (e.g., Veselovskii et al., 2002), there is not a specific "end-user" this work is addressed to. As in that case, we are simply proposing a methodology to retrieve some bulk aerosol properties from lidar measurements, and characterize relevant performances. This methodology can be used in a variety of fields concerning aerosol observations, so we believe it cannot be framed as a deliverable to answer specific "user requirements". To make our aims clearer, we better specified this point in the manuscript (line 95) in the following way: "Given the necessity to couple advancement in instrumental technology with tools capable of translating raw data into a robust, quantitative and usable information, we propose and characterize a methodology to be employed in elastic backscatter lidars and automated lidarceilometer applications to retrieve in a quasi-automatic way vertically-resolved profiles of some aerosol optical and microphysical properties. "

**2 The method-associated uncertainty seems to be very optimistic. In fact, the uncertainty is mainly similar or a just a little greater with respect to the highly costly multi-wavelength lidar retrieval, developed by Veselovskii et al., 2002.**

- In the work of Veselovskii et al. (2002) the uncertainty associated to volume and surface area retrievals is of 5% for both variables (using a 0.2 $\mu$m modal radius PSDs), and 15% and 2% (using a 2 $\mu$m modal radius PSDs). This is reported in Veselovskii's Table 1, input random uncertainty 0%, i.e. the instrumental error condition implicit in our work (our model uncertainty does not include lidar measurement uncertainty). With declared uncertainties of 30-40%, our methodology is 3-8 times less accurate than the two PSD cases addressed by Veselovskii et al. (2002), a result expected from instruments a factor 10-20 cheaper than the ones needed in that case. Furthermore, unlikely Veselovskii et al., (2002), our methodology outcomes are validated against real measurements. We added a comment on this in the conclusions (line 515): 'These are higher than those found by Veseloski et al. (2002), applying a method in the retrieval

of multi-wavelength lidar systems (10 to 20 times more expensive than ALC systems)'.

**3 The model is based on a priori three-modal log-normal aerosol distribution. Changing the mixing ratio among modes and total particle number impact of course the result. The main problem linked with this approach, is that the model should be "tuned" on particular atmospheric condition (as for Lecce, where the total number of aerosols have been diminished, hypnotizing cleaner continental aerosols).**

- We agree with the reviewer. This aspect is indeed discussed both in the manuscript (section 4.1) and in Appendix D, where we provided a sensitivity test tuning the model to better reproduce aerosol conditions at the Lecce and Potenza sites. We modified a sentence in the conclusion to further stress this point: 'An obvious intrinsic limitation is that the method is dependent on the considered aerosol type and in this study was tuned to reproduce average continental aerosol conditions. Errors associated to the application of the derived functional relationship might be larger if more 'specific' aerosol conditions (e.g. contamination by sea salt or desert dust particles) affect a given site.

**4 Simulations from MonteCarlo are fitted with a 7th grade polynomial. Even if in the manuscript follows the approach described in Barnaba et al., papers, no explanation is given about this choice. More important, it is missing in the manuscript a model sensitivity study: how the results are affected, for example if a 3rd degree polynomial is used instead of 7? Is there for example a convergence in the results if the polynomial order is increased? Or is overfitting creating problems? All those question should be properly addressed.**

- In the work of Barnaba et al. (2001) a 7th grade polynomial was used because "the polynomial fits need to be extended to the seventh degree to reach a good correlation coefficient (c2 >_0.98 ). See the Appendix A2 of the above-mentioned paper for c2 definition." Similar results have been obtained for this work. Thus, for this reason and for homogeneity with Barnaba et al. (2001) paper, we used 7th grade polynomial. A

convergence in the results is observed the polynomial order is increased. A following sentence was added in section 3: 'The choice of a seventh-order polynomial fit was made for homogeneity with BG01 and BG04a'.

Specific Comments:

**Pag 1 Line 19: Please be more specific about "continental aerosol type". Provide a very short description of the aerosol species belonging to this category.**

- Thank you. We added the following sentence: 'An average 'continental aerosol type' (i.e. clean to moderately polluted continental aerosol conditions, e.g., section 2.1) is addressed in this study'.

**Pag 2 Line 43: I suggest to classify aerosol effects on radiation budget into direct, semi-direct and indirect. Moreover, literature should be broaden (e.g. Feingold papers are missing)**

-Thank you. We modified the text as follows: 'Aerosol particles affect the Earth's radiation budget mainly by two different processes: 1) by scattering and absorbing both solar and terrestrial radiation (aerosol direct effect, Haywood and Boucher, 2000 and aerosol semi-direct effect, Johnson et al., 2004) and 2) by serving as cloud and ice condensation nuclei (aerosol indirect effect, Lohmann and Feichter, 2005, Stevens and Feingold, 2009 and Feingold et al., 2016).

**Pag. 2 Line 54, again literature is poor, I would suggest to add at least Tosca et al., 2017 remote sensing**

- Added, thank you for the suggestion.

**Pag.2 Line 57 I suggest to add reference to Lolli et al., 2018 AMT**

- Added, thank you for the suggestion.

**Pag.2 Line 59-65 CALIPSO is not the only mission with lidar monitoring the atmosphere from space. Since few years there is also CATS. Please refer to York et al.,**

2016, McGill et al., 2016

- Added, thank you for the suggestion.

**Pag. 3 Line 85. I suggest to add to the reference Madonna et al., 2018 showing results of the new intercoparison campaign INTERACT-II**

- Added, thank you for the suggestion

**Pag. 3 Line 99 and 100: Both measurement units are wrong both for surface area and volume**

- Corrected, thank you for this remark.

**Pag. 3 Line 107: please refer to the first comment**

- According to this and to a similar comment made by reviewer #2, we reformulated the sentence as follows: 'we address here an 'average-continental' aerosol type (i.e. clean to moderately polluted continental aerosol conditions, e.g., section 2.1), expected to climatologically dominate over most of Europe, despite the not negligible differences that can be encountered across Europe over both the short and the long-term (Putaud et al. 2010)'

**Pag. 4 Line 133: Please be more specific. Now the aerosol type is continental. Which is the difference with " average continental"?**

- We now refer to 'an average continental' aerosol type (i.e. clean to moderately polluted continental conditions, Hess et al., 1998)' in the text.

**Pag. 4 Line 139 Eq 1 suffers from hasty writing. Not all the variables are described in the manuscript**

- Corrected, thank you for this remark. Sorry for the confusion (see also reply to reviewer #3 on this same aspect).

**Pag. 4 Line 140 rmi is not present in Eq. 1**

-Corrected, thank you for this remark.

**Pag. 4 Line 141 Even if it is clear , variables mri and mim are not defined here (fewrows below yes) neither present in Eq.1**

- Corrected, thank you for this remark. (see also our reply above).

**Pag. 4 Line 145 N shows wrong measurement units.**

- Corrected, thank you for this remark.

**Pag. 5 Line 176-Hi is not defined. Why on equation 3 is arbitrarily used 5.5 Km ? A ref should be at least added.**

- Done, thank you for this remark.

**Pag 7 Line 226 Measurement units are wrong.**

- Corrected, thank you for this remark.

**Page 11 Line 399: Please specify which AERONET datasets were used in the manuscript and which version.**

- Done, thank you for this remark.

**Pag 13 Line 465: usually humidity is higher at night**

- The referee is right. We report here the reply to #3 reviewer that also had a comment about the impact of RH on the ALC volume retrieval. We know this effect is important in our volume estimates and relevant errors (e.g. also Barnaba et al., 2010; Adam et al., 2012). In fact, the RH dependence is taken into account by the model itself and it is accounted for in the model results variability. Indeed, errors can be much larger in the retrieval of PM loads, where a further unknown (particle density) is involved. In fact, we propose to retrieve volume not mass, and reference aerosol volume measurements are rather complex to perform if not including the full size distribution (as in the case of optical instruments). A further missing information would concern hygroscopicity of

observed aerosols. We believe an extensive discussion (ALC vs other techniques) of volume comparisons would require a full paper itself. We believe it is better here to show some comparisons as in Figure 8 demonstrating the ALC volume estimates can well match the optical ones within the expected relevant variability. However, to provide more information about RH, we added a horizontal bar in the upper part of Figure 8 indicating the range (RH<60%, 60%<RH<90% and RH>90%, respectively) of the measured in-situ RH during the ALC-OPC volume comparison. In this respect, the following text has been added: This latter effect is confirmed by the large RH values (RH > 90%) measured after 18 UTC. The lower panel shows a good agreement between the ALC-derived and the Fidas OPC Va values, in particular until 04 UTC and after 16 UTC. Some differences emerge around 07 UTC and between 11 and 15 UTC, where the ALC volume is lower by a factor of 2 compared to the in situ Fidas Va values. The smaller minimum detectable size of the Fidas OPC instrument with respect to the OPS is likely the reason for the better accord between ALC and OPC Va values in this test date. For this case, the effect of RH seems to be less important, and indeed RH values keep lower than 90%. In general, high RH values (RH >= 90%) are known to markedly affect the aerosol mass estimation from remote sensing techniques and its relationship with 'reference' PM2.5 or PM10 measurements methods, usually performed in dried conditions (e. g. Barnaba et al., 2010; Adam et al., 2012, Li et al., 2016, Li et al., 2017). This theme is also discussed in Diemoz et al. 2018a for the ALC measurement site of Figure'.

---

## Author Comment (AC2) · 9 Aug 2018

**General comment: The approach described in this manuscript can provide a helpful expansion of the data analysis of lidar-ceilometers. How much information can be added to the data however depends on the input to the model. Here, the model is trained with observations representing a continental European aerosol. Thus the results are representative for this type of aerosol and regions/times where/when it occurs.**

[Figure]

The approach in general is able to add significant benefit to climatologies derived from lidar-ceilometer networks and should therefore be published. The presentation of methods and results is sound. Limitation of the applicability to specific situations have mostly clearly been addressed by the authors. The manuscript deals with as-far-as-possible exploitation of ceilometer data – that is good. But at the point of estimating mass concentrations (see below) I have concerns, because the results you show suggest that PM10 can be estimated within 10-20% accuracy by ceilometers, which I don't think is generally true. Given the uncertainties and assumptions involved, the presented time series comparison may not even be representative for your sites at all times. Though inversion of optical data is often remarkably good-natured and your 'calibration' works for the related conditions and regional climate, this does not take into account the complexity of PM10 measurements which reflects in the +/- 25% measurement accuracy in the EC 2008/50 directive.

- We agree with the reviewer: the validation results obtained for volume and mass concentrations are not necessarily representative of other sites and times. We now clearly state this in the revised version (see specific modification to the text on this point reported below in the reply to general comment, point #2). In fact, it is the validation exercise at this specific site that found " that PM10 was estimated within 10-20% accuracy by ceilometer observations", not the methodology proposed in this paper. In this respect, the paper clearly states that the model's standard uncertainties in the retrieval of aerosol extinction and volume are within 30-40% (line 514). The retrieval of mass (PM) requires adding to this uncertainty the one related to particle density and signal quality. So, on a more general basis, we expect an uncertainty of the order of 50% when inferring aerosol mass from lidar measurements. Still, our validation exercise returned results well within this range.

**1 The manuscript presents a model-based approach to infer extinction coefficients, particle surface- and volume/mass-concentrations from backscatter coefficients measured by lidar-ceilometers, based on statistical relations. Mie-calculations are per-**

[Figure]

formed for empirical ranges of particle sizes and refractive indices, yielding conversion factors which are stored into a look-up table. As the influence of unsperical particles is shortly discussed, the Mie approach seems sufficient. The aerosol modal representation and refractive indices for model input are based on a comprehensive literature survey. Owing to the size distributions and the range of refractive indices used for the ensemble calculations, it is valid for 'continental aerosol'. The inversion of ALC profiles uses state-of-the-art absolute calibration and the Rayleigh method according to Wiegner et al. 2012. Evaluations are the most important part of the manuscript: The simulations (not including measurements) are evaluated against measurements: First the backscatter coefficient (BSC) vs Lidar ratio (LR) relation is compared against climatologies from EARLINET, CALIPSO and other networks. It is shown that average climatological LR are reproduced and that the frequency distribution of simulated BSC-LR pairs is roughly consistent with the corresponding distribution of EARLINET observations. There are, however, deviations, which are attributed to particle sizes and compositions which are, by design, not captured by the model. Then, AOD inferred from CHM15k lidar-ceilometers are compared against each 1-2 years of data from 3 Italian stations with radiometers. Frequency distributions of the bias between inferred and measured AOD are shown and one example for illustration. The usual extrapolations of radiometers to 1064 nm and of the profile below 400m towards the ground (overlap) are done.

**2 Thirdly, volume and mass concentrations are estimated from lidar-ceilometers, based on the proposed model and compared to in-situ measurements with optical particle spectrometers. Given your limitation to 'continental type aerosol', the large variability in essential conversion factors showing up in the statistical evaluations, uncertainties due to the overlap extrapolation, I wonder how representative these results are. As OPS measure dry aerosol while lidar/ALC measure optical parameters under ambient conditions: I can hardly believe that the parametrisation in your model and the information about atmospheric humidity as such is accurate enough to allow proper humidity correction in the range of the uncertainties given here. I think that these results can only be achieved under very specific conditions - aerosol type, stratification, homogeneity etc. This should be discussed in more detail.**

- We agree with many of the referee considerations here. As mentioned in the text (#lines 455-459), the comparison between OPC and ALC-retrieved volumes suffers from intrinsic factors where the different sampling conditions play a major role. Regarding the RH impact, we report here our reply to reviewer #3 who also had a comment about the impact of RH on the ALC volume retrieval. We know this effect is important in our volume estimates and relevant errors (e.g. also Barnaba et al., 2010; Adam et al., 2012). In fact, the RH dependence is taken into account by the model itself and it is accounted for in the model results variability. Indeed, errors can be much larger in the retrieval of PM loads, where a further unknown (particle density) is involved. In fact, we propose to retrieve volume not mass, and reference aerosol volume measurements are rather complex to perform if not including the full size distribution (as optical instruments do). A further missing information would concern hygroscopicity of observed aerosols. We believe a full discussion (ALC vs other techniques) of volume comparisons would require a full paper itself. We believe it is better here to show some comparisons as in Figure 8, demonstrating the ALC volume estimates to well match the optical ones within the expected relevant variability. However, to provide more information about RH, we added a horizontal bar in the upper part of Figure 8 indicating the range (RH<60%, 60%<RH<90% and RH>90%, respectively) of the measured in-situ RH during the ALC-OPC volume comparison. The following text has been added: 'This latter effect is confirmed by the large RH values (RH > 90%) measured after 18 UTC. The lower panel shows a good agreement between the ALC-derived and the Fidas OPC Va values, in particular until 04 UTC and after 16 UTC. Some differences emerge around 07 UTC and between 11 and 15 UTC, where the ALC volume is lower by a factor of 2 compared to the in situ Fidas Va values. The smaller minimum detectable size of the Fidas OPC instrument with respect to the OPS is likely the reason for the better accord between ALC and OPC Va values in this test date. For this case, the effect of RH seems to be less important, and indeed RH values keep lower than 90%. In

general, high RH values (RH >= 90%) are known to markedly affect the aerosol mass estimation from remote sensing techniques and its relationship with 'reference' PM2.5 or PM10 measurements methods, usually performed in dried conditions (e. g. Barnaba et al., 2010; Adam et al., 2012, Li et al., 2016, Li et al., 2017). This theme is also discussed in Diemoz et al. 2018a for the ALC measurement site of Figure' Concerning the PM10 comparison, to specify the limitation of the obtained results, we added the following sentence in section 4.2.2: 'This agreement attests that SPC site can indeed be considered an 'average' continental site and suggests the potential of this approach to derive information on aerosol volume and mass. Still, due to the specificity of each site and to the limited period considered here, these results cannot be taken as representative of all continental sites at all times. Further studies at different places and over longer time periods would be necessary to better assess the uncertainty of the proposed retrieval, including uncertainties due to the variability of 'continental' conditions (in terms of particle size distribution, compositions, hygroscopic effects, etc..), but also of the instrument-dependent performances (e.g. overlap corrections, etc...). '. We also added the two following sentences to the manuscript conclusions: 1) 'Overall, the good results obtained in our validation efforts are encouraging but necessarily related to the specific conditions at the sites and to the instrument characteristics considered. They are therefore not necessarily representative of results obtainable in all European continental sites, at all times. Further tests using wider datasets covering a variety of sites and ALC/lidar instrumentation would be desirable to better understand potential and limits of the applicability of the proposed method over the larger scale.' 2) Additionally, although our validation exercise returned results well within the uncertainties related to the model statistical variability alone (i.e., the relative errors associated to the mean functional relationships), the expected total uncertainty to be associated to the method should include terms that have not been specifically addressed in this work, as for example the instrumental error itself. And modified the following sentence in the abstract: 'Although limited in time, our comparison showed rather good agreement too. In particular, the ALC-derived daily-mean mass concentration for the considered site

and specific period was found to well reproduce corresponding (EU regulated) PM10 values measured by the local Air Quality agency in terms of both temporal variability and absolute values'.

Specific comments:

**1 Line 80: : : :and affordable for aerosol applications, : : :**

- Done, thank you for this remark

**2 Line 108: this is at best true in a climatological sense, but not on shorter time scales. But even on the long-term, Putaud et al. 2010 report large differences in the aerosol distribution over Europe**

- We reformulated the sentence in the following way: 'we address here an 'average-continental' aerosol type (i.e. clean to moderately polluted continental aerosol conditions, e.g., section 2.1), expected to climatologically occur over most of Europe, despite the known differences that can be encountered across the continent both in the short and the long-term (e.g., Putaud et al. 2010).'

**3 Line 181: In this formula mRH converges to 2m0 for rmi_RH $\gg$ rmi_0, i.e. for a large aequous droplet. Replace by mRh = mW + : : :**

- Corrected, thank you for this remark

**4 Line 187: In eq. 4 and 5, rim_RH and miRH are the: : : should be : : :rmi_RH: : :**

- Corrected, thank you for this remark

**5 Line 225: It is unclear to me what that means – what is 1%?**

- We meant the $\beta$a region where each of the 10 equally-spaced bins per decade of $\beta$a contains at least 1% of the simulated points for the various aerosol parameters (i.e. $\alpha$a, Sa and Va). We then reformulated the sentence: 'The red vertical bars of Figure 2 also highlight the ranges of $\alpha$a, Sa and Va which are statistically significant,

i.e. those in which, at $\lambda$ = 1064 nm, the model provides at least 1% of the total points per corresponding bin of ßa.'

**6 L 235: weighted-LR' _ 1 s. d. -> write: standard deviation**

- Corrected, thank you for this remark

**7 Line 305 and Fig 4: but only as a statistical ensemble average over all data without evaluating the temporal correlation**

- That is correct. We reformulated the sentence as follows: 'Statistically, the highest number density of simulated data well fits the observations. . . .'

**8 Line 319ff: Hamburg (and the others as well) is not really a continental site but considerable sea-salt contribution can be expected in the coarse mode, (at least for Hamburg) likely not less on average than from dust. So does the statistical agreement with your model results confirm the significance of your model? Are you sure, you'd get a worse agreement e.g. for Mace Head and would you expect to be able to draw significant conclusions about the aerosol type?**

- As presented in the text (#lines 320-322), for Hamburg, the distribution of LR values towards large values of $\beta$a ( Fig. 4) could be due to the presence of sea-salt aerosols. This contribution does not appear using the relative LR difference (LRdif=0.05) at 355 nm as an indicator of the agreement between model and measurements. As suggested by the reviewer, to verify the statistical significance of LRdif, we computed the LRdif for the EARLINET station of Cork in Ireland (Mace Head was not available). In this case, the value of LRdif (=0.25) at 532 nm attests the presence of a significant difference between the model and measurements. This is correct because at this station the sea-salt contribution is predominant. Conversely, the Hamburg site (some 60 km from the sea) is mainly continental and affected by sea-salt aerosols mainly in summer and for a specific wind direction (Matthias and Bösenberg, 2002).

The LRdif values for these two stations show that we are able to draw some conclusions

about the aerosol type, at least discriminating between mainly continental and non-continental sites. From our results, this was already clear for the Potenza and Lecce sites, whose results indicate that these sites were not purely continental (see appendix D).

**9 Line 420: why do you exclude desert dust days?**

- This is because both optical and physical properties of desert dust are very different from those of 'continental aerosol' (and there is an important issue of aerosol non-sphericity in that case, e.g., Barnaba and Gobbi 2001). We remind that, for the same reason, we also removed desert dust affected measurements from the EARLINET dataset when comparing modelled and measured LR in this work.

**10 Line 434: you should note that these conclusions are valid only in a statistical sense**

- Done, thank you for this remark.

**11 Line 452: the data from 0-75m are those from 300 m a.g. (where overlap correction is feasible) extrapolated to the ground?**

- No, for this system we didn't use the extrapolation to the ground but the original RCS data corrected by the $O(z)$ function down to the lowermost atmospheric layers. In fact, as explained in section 4.2.1 #lines 394-395, the $O(z)$ of ALC system at ASC is optimally characterized down to the ground.

**12 Line 444ff: You should specify that these OPC channel data are given as diameters, while above you mostly discuss sizes in terms of radii**

- This has been specified, thank you.

**13 Line 480: what is the meaning of a particle density of 2 _g/m3? Typical densities are of the order 1000 kg/m3. There seems to be a conversion factor included.**

- We apologize for this typo, we corrected to g/cm3.

**14 Line 486ff: Isn't the nocturnal boundary layer depth often in the range of few tens to hundreds of meters, even in summer? Strong vertical gradients in the lowest 200-300 m seem quite likely to me.**

- In fact, our statistical (3-year) ALC record shows the mixing layer height at SPC to descend below 250 m only 4-5 hours per day in July (usually between 22 and 3 UTC, i.e., when emissions are at a minimum). We believe this contributes to the good agreement between the ALC and the PM10 measurements. We reformulated the sentence: The comparison to ground-level PM10 at SPC is expected to be only slightly affected by the height difference during the considered period of the year (i.e. June and July), particularly in daytime due to the strong convection in the boundary layer. Possible exception could be in nocturnal conditions when vertical gradients in the lowermost hundreds of meters can occur. However, our statistical (3-year) ALC records show the mixing layer height at SPC to descend below 250 m only 4-5 hours per day in July (usually between 22 and 3 UTC, i.e., when emissions are at a minimum).

––––––––––––––––––––––––

---

## Author Comment (AC3) · 10 Aug 2018

Reply to referee#3

General comments: The authors present an interesting study about retrieving aerosol properties (extinction coefficient (E), surface area (S) and volume (V)) from lidar and/or automated lidar-ceilometer (ALC) backscatter measurements. The key of the method is using a "Monte-Carlo" model to simulate the relationship between E, S, V and backscat-

ter for different continental aerosol microphysical properties which could occur in real life and then implementing the relationships in the retrievals. Based on the 20000 model simulations, the relationship between lidar backscatter and aerosol E, S, V were investigated and dependence of lidar ratio (LR) to the backscatter at three lidar wavelengths (355 nm, 532 nm, 1064 nm) were fitted. The model-based LR were tested by comparing model simulations with raman lidar observations at 355 nm and found agree well with observations. Then the method was implemented to retrieve AOT and aerosol volume, PM10 and the results were compared with in-situ measurements. Although this method has some limitations in retrieving aerosol volume, mass, it shows the potential of using ALC for aerosol properties retrieval. The paper is well written and structured. The method was explained clearly, and main assumptions and limitations of the method were discussed. The topic is well suited for the AMT. I have a few comments and recommendations before the paper can be published.

- We thank the reviewer for dedicating time to check and improve our manuscript. Following the constructive comments of the referee's, several corrections have been made on the paper.

**1 Line 128: Both the step 1 and step 2 are about the aerosol model, why didn't authors put them both in the same section (section 2)?**

- This choice was intended to separate methodology from results. Given the Reviewer's objection we re-structured the text so that the old section 3 ('Model simulation results') is new section 2.2.

**2 Line 140: Is the r_mi at here same as the r_i in the equation 1 or it is another parameter? What are the m_(r i) and m_(im i)?**

- We thank the referee for pointing out this inconsistency. ll the notations indicated as misleading by the reviewer have now been corrected accordingly. In particular: - r_mi (indicating the modal radius) was replaced by ri as in eq.1 - mr and mi (real and imaginary refractive indices, respectively) are now mr_i and mim_i.

**3 Line 143: what are the specific rules?**

- The "specific rules" concern the 'variability ranges for the number mixing ratio xi (Ni/Ntot) of each component to this total'. This latter definition replaces now the original one.

**4 Line 147: The description of m_(r i) and m_(im i) should be given at the first time when they appeared in the paper, see the related comment above. Secondly, more explanations about the real and imaginary refractive indices and how they are used in the aerosol optical properties calculation should be provided.**

- The real and imaginary refractive indices are now introduced at this line (147), which is where these variables are used for the first time. Their usage in the calculation of the aerosol optical properties is specified later in the text (see equation 7-8). To clarify this, the following sentence has been added (lines 147-148): 'Being the result of different sources/processes, the three modes are also assumed to have a different composition, this impacting the optical computations through the relevant particle refractive index (mi), with both its real and imaginary component (mi = mr_i - i × mim_i). The Mie theory for spherical particles of radius ri and refractive index mi are then used to compute the extinction and backscatter coefficients (see equations below ).The Mie theory for spherical particles of radius ri and refractive index mi (= mr_i - i × mim_i) is then used to compute the extinction and backscatter coefficients of mode i (see equations 7-8)'.

**5 Line 151: What is the exact size range? The authors should indicate the range or refer the tables which shows the range of the parameters at here. Same as the mode 1, 2 and 3.**

- Thank you. We added the size ranges used for the three modes (Lines 151, 156, 160, respectively)

**6 Line 154: Why did the authors only use those values at 355 nm?**

- In fact, the values at the three wavelengths addressed in the paper are provided in

Table 2. In the text we only mentioned those at 355 nm as a quick reference. To clarify this, we now added the following sentence (line 154): 'A description of the assumptions made for each mode and relevant parameter, mostly based on literature data (Table 1), is given hereafter, the summary of the relevant variability chosen for each parameter being provided in Table 2.Hereafter, for the sake of simplicity, we will report, as an example of the refractive index variability of each mode, only the values obtained at $\lambda$=355 nm. The values of mi at $\lambda$=532 and 1064 nm are reported in Table 2.'

**7 Line 175-182: How did the authors decide to use those equations to stand the altitude dependence? Some references should be added here. What is the BG1? Is it the BG01 mentioned before?**

- We added the relevant references (Patterson et al., 1980 and BG01). Yes, we erroneously used BG1 instead of BG01. This was corrected in the revised text.

**8 Line 197: The reference of the Mie theory or code should be added here.**

- The reference to Bohren and Huffman (1983) has been added.

**9 Line 203: For mode 1 and mode 2, only the values of m_(r i) and m_(im i) at wavelength at 355 nm were introduced. How did the authors get the value at 1064 nm?**

- Definition of the wavelength dependence of the refractive indexes employed for the different modes is specified at lines #164-168. Basically, we used the wavelength dependence reported in d'Almeida et al. (1991) and in Gasteiger et al. 2011, and Wagner et al., 2012.

**10 Line 215: The "(A)" should be after the "average".**

- Corrected, thanks.

**11 Line 242: The maxima is the maxima of the fitting curve but not the maxima of the all samples. Right?**

-Correct. We added 'maxima of the fitting curve' in the sentence.

**12Line 245: For the wavelength 1064 nm, there are some samples with LR larger than 80 based on the figure 3c.**

- Right. We corrected the sentence in the following way:' (LR in the range 18 – 80 sr, except for a minor number of outliers)'.

**13 Line 292: Are the relative errors the errors of lidar measurements? What are the standard measurements (truth)?**

- Yes, these are the errors associated to the EARLINET measurement used as reference in our study (e.g., Introduction)

**14 Line 298: 5 sites were chosen, but why there are only 4 sites depicted in the figure 4.**

- This is because Figure 4 depicts the results of the model vs. measurements comparison in terms of LR vs ßa at $\lambda$=355 nm. Unfortunately, the Madrid lidar system does not have the 355 nm emission wavelength. Still, we reported in Figure C1 (Appendix C) the corresponding results at $\lambda$=532 nm including Madrid (the Hamburg lidar system is missing in this case as it does not have the 532 nm emission wavelength). Following this comment, we now added the following sentence (line299): 'For the EARLINET Raman stations fulfilling these requirements, Figure 4 depicts the results of the model-measurements comparison in terms of LR vs ßa at $\lambda$=355 nm (the corresponding results at $\lambda$=532 nm, including Madrid in place of Hamburg, are given in Appendix C, Figure C1)'The corresponding results at $\lambda$=532 nm for those system having this green channel are given in Appendix C, Figure C1'.

**15 Line 313: ãËŸA Ëİ ULRãËŸA ° U_mod,ãËŸA Ëİ ULRãËŸAU° _meas should be explained at here.**

- The definition of LRmod and LRmeas has been added, (we guess this was the objection).

**16 Line 329: The table 6 should be referred at here.**

- Thank you, the reference to Table 6 has been added (line 329).

**17 Line 366-371: Although the retrieval method was introduced by other scientists before, it is better to discuss more about how to derive aerosol extinction from the ALC e.g. show the key equations. Audiences may have questions like what are the raw data of the ALC? Are the raw data the range corrected backscatter? Does the raw data already consider the attenuation of signal from height z to surface due to aerosol and molecular extinction?**

- The raw data of the ALC considered in this study is the range corrected signal $z2*P(z)$ where z is the range, and (P) the raw signal stored in Netcdf format. To provide the information requested, we added the key equations of the algorithm at the end of section 3.2. Now the new text with the equations are in the pdf supplement file.

**18 Line 389: What made the authors to choose this threshold of AOT for cases screen?**

- Thank for noticing this omission. We now added the following sentence to explain this choice: 'This range allows for excluding the data points with 1064nm AOT lower than the sunphotometer accuracy (dAOT=0.01) and those where we found aerosol extinction to cause significant deterioration in our ALC signal.

**19 Line 408-421: It is suggested to give a AOT VS AOT scatter plot at here. Then it will help the audiences to have sense of both absolute and relative errors of AOT.**

- Following this suggestion, we added in Appendix E, the AOT vs AOT scatter plots for the three considered sites. The new text reads as follows: 'To have sense of both absolute and relative errors of AOT, we reported in this section the scatter plots between the hourly-mean coincident AOTs at 1064 nm as derived by ALC model-based approach and those measured at 1020 nm by the sun-photometers at 1020 nm installed at RTV, SPC and ASC, respectively (Figure E1, E2 and E3). The corresponding linear fit y = bx

[Figure]

(red line), where x = sun-photometer AOT, y = Nimbus CHM15k AOT are also shown in the plots. The values of the correlation coefficients for the three sites (R = 0.77, R=0.72 and R=0.73 for RTV, SPC and ASC, respectively) attest a relatively good agreement between the two AOT measurements.' The three added scatter plots are attached at the end of the file and their captions are, respectively:

Figure E1. Scatter plot between the hourly-mean coincident AOTs at 1064 nm as derived by the ALC model-based approach and measured at 1020 nm by the AERONET AERONET sunphotometer at RTV. The red line represents the linear fit y = bx between the two datasets, where x = sun-photometer AOT; y = Nimbus CHM15k AOT.

Figure E2. Scatter plot between the hourly-mean coincident AOTs at 1064 nm as derived by the ALC model-based approach and measured at 1020 nm by the AERONET SKYRAD photometer at SPC. The red line represents the linear fit y = bx between the two datasets, where x = sun-photometer AOT; y = Nimbus CHM15k AOT.

Figure E3. Scatter plot between the hourly-mean coincident AOTs at 1064 nm as derived by the ALC model-based approach and measured at 1020 nm by the AERONET SKYRAD photometer at ASC. The red line represents the linear fit y = bx between the two datasets, where x = sun-photometer AOT; y = Nimbus CHM15k AOT.

**20 Line 431: With the fixed LR=52, the bias (<|dAOT|>=0.021 and 0.006) are smaller than the model-based bias (<|dAOT|> =0.11, 0.13) shown in line 428, right? authors said it is larger?**

- Indeed it is larger. The problem was that the numbers reported in the text did not correspond to the values in Table 8. This was an error of us. Thank you for noting that. Actually, with fixed LR=52 sr, the bias at SPC and RTV is equal to 0.021 and 0.026, respectively, whereas the model-based bias is 0.011 and 0.013. The correct values have been inserted in the revised text.

**21 Line 440-444: How did the authors calculate the aerosol volume? Was the retrieval**

based on equation 7 and 10? Authors should explain more about the retrieval at here.

- We understand this point was not clear enough, we thus reformulated the relevant sentence as follows (line 441): 'In particular, we use the model-estimated 7th-order polynomial fit equation linking Va and ßa at $\lambda$ = 1064 nm (see Table 3 and Figure 2c) to retrieve aerosol volume profiles from ALC-derived- ßa measurements. These results were compared to . . .'.

**22 Line 460-470: The hygroscopic growth could induce large differences between the insitu measured and the ALC retrieved aerosol volume. In the work of Siwei Li et al. (2016, 2017), they discussed the impacts of aerosol size distribution in the retrieval of PM2.5 using ceilometers (Li et al., 2016) and relationship between relative humidity and PM2.5/ceilometer-backscatter ratio (Li et al., 2017). More discussion about volume, PM retrieval and comparisons of model-based retrieval with in-situ measurements e.g. model vs in-situ scatter plot should be added here. Adding aerosol size (can compare the in-situ measurements and angstrom exponent) and relative humidity information and analysis at here may help the authors to support their conclusion.**

- The referee is right and we know this effect is important in our volume estimates and relevant errors (e.g. also Barnaba et al., 2010; Adam et al., 2012). In fact, the RH dependence is taken into account by the model itself and it is accounted for in the model results variability. Indeed, errors can be much larger in the retrieval of PM loads, where a further unknown (particle density) is involved. In fact, we propose to retrieve volume not mass, and reference aerosol volume measurements are rather complex to perform if not including the full size distribution (as in the case of optical instruments). A further missing information would concern hygroscopicity of observed aerosols. We believe an extensive discussion (ALC vs other techniques) of volume comparisons would require a full paper itself. We believe it is better here to show some comparisons as in Figure 8 demonstrating the ALC volume estimates can well match the optical ones within the expected relevant variability. However, to provide more information about RH, we added a horizontal bar in the upper part of Figure 8 indicating the range (RH<60%,

60%<RH<90% and RH>90%, respectively) of the measured in-situ RH during the ALC-OPC volume comparison. In this respect, the following text has been added: 'This latter effect is confirmed by the large RH values (RH > 90%) measured after 18 UTC. The lower panel shows a good agreement between the ALC-derived and the Fidas OPC Va values, in particular until 04 UTC and after 16 UTC. Some differences emerge around 07 UTC and between 11 and 15 UTC, where the ALC volume is lower by a factor of 2 compared to the in situ Fidas Va values. The smaller minimum detectable size of the Fidas OPC instrument with respect to the OPS is likely the reason for the better accord between ALC and OPC Va values in this test date. For this case, the effect of RH seems to be less important, and indeed RH values keep lower than 90%. In general, high RH values (RH >= 90%) are known to markedly affect the aerosol mass estimation from remote sensing techniques and its relationship with 'reference' PM2.5 or PM10 measurements methods, usually performed in dried conditions (e. g. Barnaba et al., 2010; Adam et al., 2012, Li et al., 2016, Li et al., 2017). This theme is partially discussed in Diemoz et al. 2018a for the ALC measurement site of Figure 8'.The impacts of aerosol size distribution in the retrieval of PM2.5 using ceilometers and the relationship between relative humidity and PM2.5/ceilometer-backscatter ratio have been discussed in several studies (e.g. Li et al., 2016, Li et al., 2017). These impacts are important in our volume estimates and relevant errors (e.g. also Barnaba et al., 2010; Adam et al., 2012). However, the quantitative characterization of the impact of RH and of particle number concentration to the ALC retrieved volume is beyond the aims of this paper and would require a full paper itself. This theme is also discussed in Diemoz et al. 2018 (being submitted to this same issue).

**23 Line 477: What specific aerosol densities did the authors use in the retrieval and why?**

- Actually, values of aerosol densities were already mentioned in the text (#line 480: a = 2 g/cm3 , with a range between 1.5-2.5 g/cm3). We took the opportunity of this comment to specify that: 'This range covers approximately the mean a values of the

SPC site.'

**24 Line 485: Why did the authors use different heights in estimation of surface aerosol volume (0-75 m) and mass (at 225 m)?**

- This is because the two estimates come from different systems. As explained at #lines 394-395, the ALC overlap function of the ASC site has been optimally characterized and therefore for this system we used the lowest altitudes to estimate the surface aerosol volume. Conversely, the ALC system at SPC has an old firmware and its overlap function is not optimally characterized, we therefore used the 225 m level as more trustworthy. This was highlighted in line 485.

**25 Line 488-490: Were the mean and relative difference between the two-series based on hourly average PM10 or daily average PM10? What is the absolute difference? What is the R between them?**

- As reported at #lines 474 and 481, the two series are the daily average PM10. We added in the text the values of R and of the absolute and relative differences (line 488): 'Overall, Figure 9 confirms a good agreement between the ALC-derived and the ARPA reference PM10 values, with a correlation coefficient (R) of 0.73. In fact, mean, absolute mean and relative differences, between the two series are: <dPM10> = 2.8 $\pm$ 6.5 g/cm3, <|dPM10|> = 5.2 $\pm$ 4.7 g/cm3 and <(dPM10/PM10)> = 0.15 $\pm$ 0.27.This is confirmed by the good agreement between the ALC-derived and the ARPA PM10 values (Fig. 9) with a correlation coefficient (R) of 0.73. In fact, mean, absolute mean and relative differences, between the two series are: <dPM10> = 2.8 $\pm$ 6.5 $\mu$g/m3, <|dPM10|> = 5.2 $\pm$ 4.7 $\mu$g/m3 and <(dPM10/PM10)> = 0.15 $\pm$ 0.27'.

**26 Line 496: Where is the close bracket?**

- It was missing, sorry. Corrected.

**27 Line 549: Surface area and volume are not the optical properties. The results from this work showed that ancillary data information are needed to get accurate aerosol**

[Figure]

properties e.g. volume, mass.

-The sentence has been reformulated in the following way: 'On the other hand, the proposed approach has the main advantage of allowing the operational (i.e. 24/7) retrieval of fairly reliable, remote sensing profiles of aerosol optical (ßa, $\alpha$a) and physical (Sa, Va) properties (with associated uncertainties and limitations) by means of relatively simple and robust instruments.Overall, the main advantage of the proposed approach is the possibility to operationally (i.e. h24/7) retrieve fairly reliable, remote sensing profiles of aerosol optical (ßa, $\alpha$a ) and physical (Sa, Va) properties (with associated uncertainties and limitations) by means of relatively simple and robust instruments. Conversely, we know accurate measuring of aerosol optical properties to require rather expensive, in situ instruments. Furthermore, measurements of aerosol volume and surface area represent a rather difficult task even for in-situ, ground alone observations.

**28 Line 551: What kind of meteorological monitoring can be provided by the method?**

- Potentially, the aerosol vertical characterization in terms of aerosol backscatter, extinction, surface and volume derived by the proposed-method together with the ALC 'standard' information on cloud base and on the boundary layer can provide interesting information on the aerosol-cloud interaction and the involved meteorological processes. We have integrated the sentence: 'This could temporally and spatially complement the information coming from more advanced lidar networks (for example, the Raman channel of multi-wavelength system cannot be used in daylight conditions) and, more in general, could represent a valid option to deliver, in quasi real time, the 3D aerosol fields useful for operational air quality (e.g. integration of the in situ surface measurements) and for meteorological and climate monitoring (e.g. aerosol-cloud interaction and aerosol transport and dispersion processes).This approach can represent a valid option to extend the capabilities of ALCs at characterizing the aerosol vertical distribution, providing important information for operational air quality (e.g. integration of the in situ surface measurements) and for meteorological and

climate monitoring (e.g. aerosol-cloud interaction and aerosol transport and dispersion processes).

Please also note the supplement to this comment:
https://www.atmos-meas-tech-discuss.net/amt-2018-79/amt-2018-79-AC3-supplement.pdf

––––––––––––––––––––––––––

[Figure]

**Rome Tor Vergata**

R = 0.77
y=bx;  b = 0.99

Nimbus CHM15k AOT

sun-photometer AOT

**Fig. 1.**

[Figure]

[Figure]

**San Pietro Capofiume**

R = 0.72
y=bx;  b = 0.97

Nimbus CHM15k AOT

sun-photometer AOT

**Fig. 2.**

[Figure]

**Aosta Saint Christophe**

R = 0.73
y=bx;  b = 0.89

Nimbus CHM15k AOT (y-axis: 0, 0.02, 0.04, 0.06, 0.08, 0.1, 0.12, 0.14, 0.16, 0.18, 0.2)

sun-photometer AOT (x-axis: 0, 0.02, 0.04, 0.06, 0.08, 0.1, 0.12, 0.14, 0.16, 0.18, 0.2)

**Fig. 3.**

[Figure]